

# Measurements and modeling of snow albedo at Alerce Glacier, Argentina: effects of volcanic ash, snow grain size and cloudiness

Julián Gelman Constantin[1,2], Lucas Ruiz[3], Gustavo Villarosa[4,5], Valeria Outes[4], Facundo Bajano[1], Cenlin He[6], Hector Bajano[1], and Laura Dawidowski[1]

[1]División de Química Atmosférica, Gerencia de Química, Comisión Nacional de Energía Atómica, Av General Paz 1499, San Martin, B1650KNA Buenos Aires, Argentina
[2]Consejo Nacional de Investigaciones Científicas y Técnicas (CONICET), Argentina
[3]IANIGLA, Gobierno de Mendoza, Universidad de Cuyo, CONICET, CCT-Mendoza, Mendoza, Argentina
[4]Instituto Andino Patagónico de Tecnologías Biológicas y Geoambientales (IPATEC), CONICET-UNCo, Bariloche. Argentina
[5]Centro Regional Universitario Bariloche, Universidad Nacional del Comahue, Bariloche. Argentina
[6]Research Applications Laboratory, National Center for Atmospheric Research, Boulder, CO, USA

**Correspondence:** Julian Gelman Constantin (juliangelman@cnea.gov.ar)

**Abstract.** The relevance of light absorbing impurities in snow albedo (and its effects in seasonal snow or glacier mass balance) have been under study for several decades. However, the effect of volcanic ash has been much less studied, and most articles studied only the effect of thick layers after direct deposition. There is also a knowledge gap in field measurements of seasonal snow and glaciers of the southern Andes, that only recently has started to be filled. We present here the first field measurements

on Argentinian Andes, combined with albedo and mass balance modeling activities. Measured impurities content ($1.1 \, \mathrm{mg \, kg^{-1}}$ to $30000 \, \mathrm{mg \, kg^{-1}}$) varied abruptly in snow pits and snow/firn cores, due to high surface enrichment during ablation season and possibly local/regional wind driven resuspension and redeposition of dust and volcanic ash. In addition, we observed a high spatial hetereogeneity, due to seasonality, glacier topography and prevailing wind direction. Microscopical characterization showed that the major component was ash from recent Calbuco (2015) and Cordón Caulle (2011) volcanic eruption, with

minor presence of mineral dust and Black Carbon. We also found a wide range of measured snow albedo (0.26 to 0.81), which reflected mainly the impurities content and the snow/firn grain size (due to aging). SNICAR model has been updated to model snow albedo taking into account the effect of cloudiness on incident radiation spectra, improving the match of modeled and measured values. We also ran sensitivity studies on the main measured parameters (impurities content and composition, snow grain size, layer thickness, etc) to assess which field measurements precision can improve the uncertainty of albedo modeling.

Finally, we studied the impact of these albedo reductions in Alerce glacier using a spatially distributed surface mass-balance model. We found a large impact of albedo changes in glacier mass balance, and we estimated that the effect of observed ash concentrations can be as high as a $1.25 \, \mathrm{m \, w \, e}$ decrease in the glacier-wide annual mass balance (due to a 34 % of increase in the melt during the ablation season).



## 1  Introduction

Glaciers are highly sensitive to climate fluctuations, their unprecedented retreating rates observed during the last decades represent one of the most unambiguous signals of climate change (Zemp et al., 2015; IPCC, 2019). Along the Southern Andes, both precipitation decrease and air surface temperature increase have been pointed out as the drivers of the shrinkage of glaciers in the last decades (Dussaillant et al., 2019). Although some processes, like sublimation at the high and cold Dry Andes (37° S to 20° S) or the calving at the outlet glaciers of the Patagonian Ice fields (south of 45° S), could contribute, or

be even more critical than melt for the shrinkage of glaciers in some particular cases, ablation is mainly ruled by melt. Along the Southern Andes, melt is driven by shortwave radiation and sensible turbulent flux (Schaefer et al., 2019). The effect of incoming shortwave radiation is enhanced during spring and summer, due to the exposure of low albedo areas in their ablation zones, which causes strong, positive feedback that enhances surface melt significantly and shapes the spatial ablation pattern (Brock et al., 2000). Furthermore, deposition of light-absorbing impurities (LAI; mineral dust, volcanic ash, and black carbon)

have a fundamental impact on the melting of glacier and snow-covered areas by increasing the absorption of solar radiation and produces a regional land-atmosphere feedback (Warren and Wiscombe, 1980; Bond et al., 2013; Molina et al., 2015). Along with the enhanced melting due to the darkening of the snow or ice surface, the growth of snow grains is accelerated, which further reinforces snowmelt rates due to further albedo decrease (Bond et al., 2013; Flanner et al., 2007). While LAI control the snow albedo mainly in the visible wavelengths (since ice is relatively transparent in the visible band), the snow grain

size affects the albedo in the near-infrared (e.g., Hadley and Kirchstetter, 2012; Pirazzini et al., 2015; He and Flanner, 2020). Recently it has been highlighted that the growth of glacier algae could also decrease the albedo (Williamson et al., 2019).

Different snow albedo models have been developed to include the direct effect of Black Carbon (BC) and other atmospheric particulate matter (PM) as well as several positive feedbacks (Flanner et al., 2007; Koch et al., 2009; Krinner et al., 2006), the effects of non-spherical snow grains (Libois et al., 2013; He et al., 2017), and external/internal mixing of impurities with

snow grains (He et al., 2018). Although some snow albedo models have been successfully validated for laboratory conditions (Brandt et al., 2011; Hadley and Kirchstetter, 2012), the prediction of snow spectral albedo in environmental conditions is still challenging. More than just one particle metric distribution is necessary to reproduce the spectral snow albedo at all optical wavelengths, especially when the snow has been undergoing heavy metamorphosis processes (Carmagnola et al., 2013; Pirazzini et al., 2015). Notably, there has been found that taking into account the amount of LAI in the snow reduces the

difference between simulated and measured broadband albedos (Zhang et al., 2018).

Different studies have considered the effect of LAI in snow and ice albedo and its impact on glaciers mass balance or seasonal snow cover, and estimated its radiative forcing (Qian et al., 2015; Skiles et al., 2018). Some studies used point measurements of LAI content (ice cores) together with a snow albedo model to estimate potential melting, using a radiative transfer model to calculate the additional absorbed energy by BC and mineral dust (Ginot et al., 2014; Zhang et al., 2018) or perturbing a

glacier mass balance model to include BC forcing (Painter et al., 2013). Online coupling of snow albedo models in global or regional atmospheric chemistry models have been applied to study snow and glaciers interaction with the climate around the globe (Hansen et al., 2005; Flanner, 2013; Ménégoz et al., 2014). Although these global or regional atmospheric studies are



beneficial to identify LAI sources and dispersion patterns and to compare snow-atmosphere feedback in different regions, the spatial resolution can be inadequate to obtain accurate results in mountain regions (Ménégoz et al., 2014; Qian et al., 2015).

Even though most studies focus on the effect of BC, some include the effect of mineral dust (e.g., Ginot et al., 2014; Skiles and Painter, 2017; Zhang et al., 2018) or even concentrate on mineral dust due to local/regional relevance (e.g., Krinner et al., 2006; Painter et al., 2012; Wittmann et al., 2017). Studies on the effect of volcanic ash concentration on snow albedo are scarcer (e.g., Conway et al., 1996; Brock et al., 2007; Young et al., 2014).

In recent years there has been an increase of measurement and modeling of albedo along the Southern Andes (Rowe et al.,
2019). A three-year study (Schmitt et al., 2015) showed that glaciers closer to population centers in the Cordillera Blanca, Peru, have higher surface content of equivalent black carbon (EBC, BC plus other LAI, especially dust in this case), up to $70\,\mathrm{ng\,g^{-1}}$ EBC, as compared with remote glaciers (with surface content as low as $2.0\,\mathrm{ng\,g^{-1}}$ EBC). A one-week study successfully connected the decreases in snow broadband albedo with heavy traffic days in the nearby road that connects Argentina and Chile (Cereceda-Balic et al., 2018). A more recent study along the Southern Andes of Chile found a mean albedo reduction
due to light-absorbing impurities in the snow, with its corresponding mean radiative forcing increase (Rowe et al., 2019). They conclude that in the north (dusty, vegetation-sparse Atacama Desert), BC plays a smaller role than non-BC, whereas near Santiago and in the south (vegetation-rich), the BC contribution is higher. For example, the albedo reduction for spherical snow grains radii of $100\,\mu\mathrm{m}$ due to BC alone in the north is only about 43 % of that for all light-absorbing impurities. By comparison, these albedo reductions are 53% and 82% near Santiago and in southern Chile, where a greater share of light absorption is due
to BC. In the Southern Andes of Argentina, the only available information on snow albedo is due to remote sensing (Malmros et al., 2018), and up to now, the impact of volcanic ash and other LAI on Argentinian glaciers mass balance has not been evaluated either.

Here we present the results from two field campaigns developed in the Alerce glacier during April 2016 and April 2017 to assess the bounds of PM deposition impact in the Alerce glacier mass balance. We show in situ albedo measurements and PM
concentration values measured on surface and sub-surface snow and firn samples in accumulation and ablation zones of the glacier. Albedo in situ measurements are compared with results from SNICAR snow albedo model (Flanner et al., 2007; He et al., 2018), using measured snow properties and PM content as input data. We present here an improvement of SNICAR's incident radiation spectra (presented as SNICARv2.1), to take into account changes in direct and diffuse solar radiation for partly cloudy skies. We study the effect of nearby volcanic events that occurred in recent years (Puyehue-Cordón Caulle and
Calbuco). Finally, the influence of PM on snow/ice albedo on the annual surface mass balance of Alerce glacier is assessed using an enhanced temperature index melt model (Oerlemans, 2001). This study is not only the first field study of the impact of PM in Argentinian glaciers, but also one of the few studies of the long-term impact of volcanic ash on snow albedo.

## 2   Site Description and Experimental Methods

Alerce is a small ($2.2\,\mathrm{km^2}$), debris-free, mountain glacier located at Monte Tronador ($41.15^\circ$ S, $71.88^\circ$ W), in the Northern
Patagonian Andes. The climate on this region is primarily modulated by the weather disturbance embedded in the mid-latitude



westerlies (Garreaud et al., 2009). Weather disturbances and prevailing winds coming from the Pacific Ocean are more frequent and stronger in winter. However, associated frontal precipitation system move over the Patagonian Andes all year round. In this region, the hydrological year is defined from the 1-April to the 31-March of the next year. The accumulation season last from 1-April to 31-October and the ablation season from 31-October to the 31-March of the next year.

Alerce glacier has an elevation range between $1650\,\mathrm{m}$ to $2400\,\mathrm{m}$ a.s.l. (above sea level), a gentle slope (mean of $10°$), and is exposed to the southeast. Since 2013 it has been the focus of a glacier mass balance monitoring program by the IANIGLA (Instituto Argentino de Nivología, Glaciología y Ciencias Ambientales). Seasonal mass balance has been studied every year using the traditional glaciological method of stakes, and snow pits. An enhanced temperature index mass balance model has been developed (Ruiz et al., 2015, 2017) to study the surface mass balance of the glacier. This model is used here to analyze

the influence of PM, through glacier albedo changes, over the mass balance of Alerce glacier.

In recent years Monte Tronador glaciers have been reached by volcanic ash derived from two volcanic events: (i) Puyehue-Cordón Caulle volcanic complex, which had a long eruption between June 2011 and January 2012 and (ii) Volcán Calbuco, which commenced on April 23rd 2015.

## 2.1    Fieldwork

In April 2016 and April 2017, besides mass balance measuring, we took snow and firn samples and we measured surface albedo at Alerce glacier. Figure 1 shown the sampling sites at Alerce glacier. We sampled both accumulation and ablation zones, and we looked for similar sampling sites in both campaigns. Otto Meiling mountain hut served both as a base camp for field trips and as a field laboratory for initial processing of the snow samples. April 2016 served as a exploratory campaign. Albedo measurements were improved for the 2017 campaign. We lowered instrumental uncertainty and we used an improved support for the pyranometer, which allowed us to evaluate the variability/uncertainty of albedo measurements by repeatedly measuring

in the same site. We also improved the measurement of snow grain size distribution. More details are given below. However, the second campaign duration and number of sampling sites were shortened due to poor weather conditions. Nevertheless, relevant results of PM concentration and albedo measurements are presented for the first time for Monte Tronador glaciers.

### 2.1.1    Snow samples. Filters treatment

Before collecting snow/firn/ice samples, we performed a stratigraphy at each site to identify and date layers. Many of the sampling sites corresponded to the accumulation zone of Alerce glacier or accumulation pockets in the ablation zone. In those sites, we dated seasonal layers of snow/firn. The main elements to attribute layers were PM content and hardness of the layers. Figure 2 shows the results of the stratigraphy and PM gravimetry, which are described in detail in Section 3.1. In sampling sites located on the ablation zone, we distinguish glacier ice from recent snow covering the glacier ice.

Most of the samples were taken from snow/firn pits. In the 2016 campaign we also used a snow/firn hand auger to sample a $2.5\,\mathrm{m}$ snow/firn core (site *Acc2-2016*, Fig. 2). Samples were melted and filtered in the base camp, and filters were taken to the laboratory for gravimetric determination of PM content and further analysis. Further details are given in Section S1.1 of Supplement.



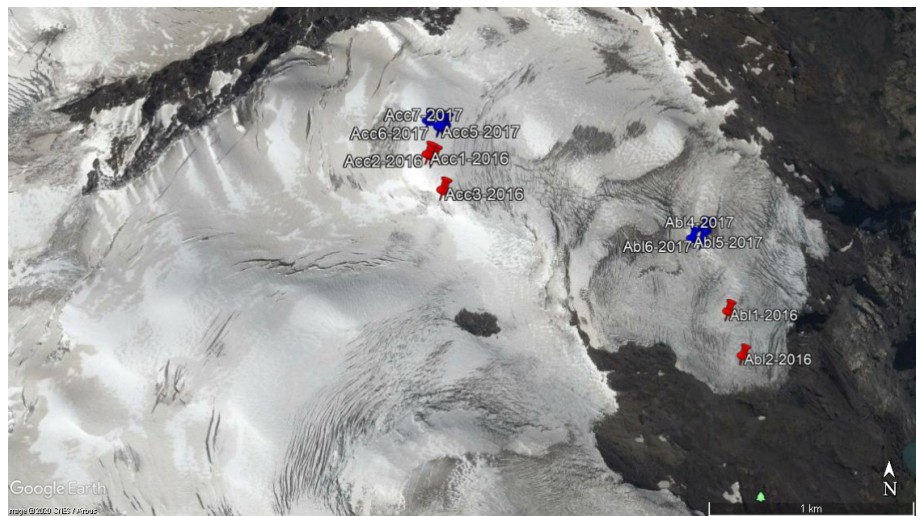

**Figure 1.** Satellite image of Alerce Glacier. Sampling points are represented as blue markers (2017 campaign) and red markers (2016 campaign). Green marker represents Otto Meiling mountain hut. Copyright: © Google Earth, 2020, CNES/Airbus

PM in the filters was described and photographed using a Leica S8APO stereo microscope equipped with a DFC 295 camera.
Some samples were also studied by Scanning Electron Microscopy (FEI Quanta 200, equipped with an Edax accessory for energy dispersive X-ray analysis).

## 2.2 Albedo: measurements and corrections

We performed in-situ albedo measurements in some of the snow sampling sites in both field campaigns. Upwelling (reflected) and downwelling (direct + diffuse) radiation were measured with a CM5 Kipp & Zonen pyranometer (wavelength range $0.3\,\mu m$
to $2.8\,\mu m$), using two different in-house developed supports in 2016 and 2017 campaigns, logged with a handheld voltmeter. The voltmeter used in the 2016 had a reduced precision (resolution of $0.1\,mV$) that limited the overall accuracy of the albedo measurement (first two rows of Table 1). In the 2017 a new, more accurate voltmeter was used (resolution of $0.001\,mV$, accuracy of $0.010\,mV$), reducing significantly instrumental uncertainty. Further details are given in Section 3.3.

Raw albedo values were corrected to account for the diffuse or reflected light blocked by the operator or the support and, for
upwelling radiation, the effect of shadows of the sensor and the support in the snow surface (Wright et al., 2014; Carmagnola et al., 2013). Further details are given in Section S1.2 of Supplement.

**Pyranometer supports and cloudiness effect**

In the 2016 campaign, we used a fixed support with three stainless steel legs (Fig. 3 (a)). It was designed to provide a stable irradiation measurement, with a precise tilt angle (parallel to the snow surface), and to minimize the blocking of incident light.
When measuring clear-sky downwelling radiation, this support does not block light at all (operators stand $4\,m$ away from the



sensor, blocking less than 0.1% of incoming diffuse radiation). For clear-sky upwelling radiation, the percentage of blocked light is below 0.8 %, and shadows from the equipment represent another 0.4 %. Hence, total correction for upwelling radiation sum up around 1.2 %, affecting around 1% measured albedo. For cloudy or overcast conditions, due to the sharp changes in cloud cover, incoming radiation varies more quickly than the time needed for assembling/disassembling the pyranometer sup-

port. To proceed faster under these conditions, the measurements were made differently: the sensor was held by two operators, each $0.45\,\mathrm{m}$ away from the sensor, without using the support legs. Under these conditions 12 % of diffuse downwelling and 9% of upwelling radiation is blocked by the operators, resulting in an albedo correction of 3.5 %, significantly higher than those obtained for clear-sky conditions.

  To overcome the difficulties due to cloudiness, for the 2017 campaign a new support was designed. The new lighter design

has only one arm and one leg and is carried by one operator, located $1.25\,\mathrm{m}$ away from the sensor, and leveled manually with the help of a bubble level (Fig. 3 (b)). This design allows fast and easy alternate downwelling/upwelling radiation measurements, making it possible to assess the variability of albedo under the same sky conditions. For downwelling radiation the operator blocks around 1.1 % of diffuse light. For upwelling radiation, the operator blocks around 1.9 % of light, which, together with shadows of the equipment, brings corrections to a maximum of 2.4 %. Overall albedo corrections vary between 0.8 % and 2.0

150 %.

**Diffuse and direct radiation fraction**

For albedo calculation, the upwelling radiation measurement is used directly from measurements. But for downwelling radiation, direct and diffuse fraction must be distinguished (see Eq. (S1) in Supplement).

  The calculation of the diffuse fraction of downwelling radiation requires to add another measurement with the pyranometer

(total downwelling, diffuse downwelling, and total upwelling radiation), and the operation of the accessory to block direct radiation. Fast changes in cloudiness during measurements made it very difficult to assure that all three measurements were performed under the same sky conditions. Therefore, we decided to prioritize that measurements required for albedo calculation (total downwelling and total upwelling radiation) were performed under the same conditions, and thus we dropped the diffuse downwelling radiation measurement. Hence, the diffuse to global radiation ratio $I_{diff\downarrow}/I_{glob\downarrow}$ (needed for albedo mea-

surements corrections and comparison with modeled albedo) had to be estimated differently. We used in-situ observations of cloudiness (or pictures of the sky taken before and after albedo measurements) together with the relations found by Kasten and Czeplak (1980)[eq. 4] to estimate the diffuse radiation ratios, which are presented in Table 1.

**Snow/firn grain size**

In the 2016 campaign, snow was placed in a crystal grid (with three different scales: $2\,\mathrm{mm}$, $1.2\,\mathrm{mm}$, and $0.6\,\mathrm{mm}$) and average

size was determined with a magnifying lens. In the 2017 campaign, a similar in-house developed grid was used (with two scales: $1\,\mathrm{mm}$ and $0.5\,\mathrm{mm}$) in combination with a macro lens and a mobile phone digital camera. High-resolution pictures where analyzed later with ImageJ software (Schneider et al., 2012). The new equipment and methodology introduced in the 2017 campaign allows a more detailed description of the snow samples and a more precise average radius value.



## 2.3 Albedo: modeling

To analyze the different factors affecting measured albedo at each sampling site, we modeled albedo for the same conditions using SNICAR (Flanner et al., 2007; He et al., 2017, 2018). Snow density and layer thicknesses were taken as parameters from in-situ stratigraphies. Average snow grain size and shape were obtained from in-situ measurements. PM content was obtained from filters gravimetry. Based on in-situ observations and the analysis of microscopy images, are described in detail in Section 3.2, we assigned all recollected PM mass to volcanic ashes (in a similar way as previously done in sites where mineral

dust represents most of LAI (Krinner et al., 2006; Painter et al., 2012; Wittmann et al., 2017)). Albedo of the underlying layers was calculated explicitly within the same model, using the properties of those layers.

SNICARv2 (He et al., 2017, 2018) supported only four incident solar spectra: two clear-sky direct solar spectra (one for Summit, Greenland, and one for mid-latitude), and two overcast diffuse spectra (for the same locations). These spectra are used to calculate direct radiation albedo and diffuse radiation albedo, respectively. These are good approximations for clear-

sky albedo (where most for the incident radiation is direct, clear-sky solar radiation) or for overcast sky albedo (where most of the radiation is diffuse). In this updated version of SNICAR (referred as SNICARv2.1 throughout the article) we provided an alternative for these spectra for cases when latitude, longitude or altitude differ significantly from those of the provided spectra, or where the sky is partly cloudy.

First, we calculated the clear-sky spectra for the site location and time using SMARTS model (Gueymard, 2001). Then, we

calculated the direct and diffuse spectra for overcast or partly cloudy sky following Gueymard (1986, 1987) and Ernst et al. (2016):

$$F_{dir,norm}(\lambda) = \frac{F_{dir,S}(\lambda)}{I_{dir,S}} \tag{1}$$

$$F_{diff,norm}(\lambda) = [1 - N_{pt}]\frac{F_{diff,S}(\lambda)}{I_{diff,S}} + N_{pt}\frac{F_{dir,S}(\lambda) + F_{diff,S}(\lambda)}{I_{glob,S}} \tag{2}$$

$I_{dir}$, $I_{diff}$, and $I_{glob}$ are clear-sky direct, diffuse and global solar irradiance as calculated from SMARTS model. $F_{dir,S}(\lambda)$

and $F_{diff,S}(\lambda)$ are the spectral distributions of clear sky direct and diffuse solar irradiance, also from SMARTS model. $F_{dir,norm}(\lambda)$ and $F_{diff,norm}(\lambda)$ are the normalized spectral distributions of direct and diffuse solar irradiance thus calculated for our sites. The cloud opacity factor $N_{pt}$ is calculated following Ernst et al.:

$$N_{pt} = \frac{\rho - \rho_S}{1 - \rho_S} \tag{3}$$

where $\rho$ and $\rho_S$ are the diffuse to global irradiance ratio for the site and from SMARTS model, respectively.

The clear-sky direct radiation spectra available in SNICARv2 matches reasonably well SMARTS clear sky direct radiation spectra. In the other hand, SMARTS clear sky diffuse radiation spectra is very different from diffuse radiation spectra available



in SNICARv2 (Fig. 4). The spectral distribution obtained for 95 % cloud fraction for SNICARv2.1 closely matches the diffuse radiation spectra available in SNICARv2, which confirms that the latter was prepared to represent an overcast sky condition. In the other hand, the spectral distribution obtained for a 50 % cloud fraction differs significantly from both spectra available in SNICARv2, showing a larger contribution from clear sky diffuse radiation (Fig. 4).

Hence, we expect to find a larger impact of our improved incident sun spectra for intermediate cloud cover fractions. For clear sky conditions, direct radiation spectra were already well represented. Even though diffuse radiation spectra were not accounted for, this fact has little impact on the calculated albedo, due to the low diffuse radiation fraction for clear sky conditions. Conversely, for overcast conditions, diffuse radiation spectra were already well represented and neglecting direct radiation fraction has a low impact on albedo calculations.

Using different incident radiation spectral distributions, we obtained the pure direct and diffuse albedo with SNICARv2 and SNICARv2.1 ($\alpha_{dir}$ and $\alpha_{diff}$). For SNICARv2.1 we also calculated the weighted average albedo, which should be compared to the net measured albedo:

$$\alpha = \rho\alpha_{diff} + (1 - \rho)\alpha_{dir} \tag{4}$$

## 2.4 Alerce glacier surface mass balance model

To analyze the role of albedo decrease over the surface mass balance of Alerce glacier, we use a spatially distributed surface mass-balance model (spatial resolution 20 m) driven by daily temperature, precipitation, and potential direct solar radiation (Huss et al., 2008). The model was calibrated by surface mass balance measurements performed on a seasonal to annual basis through the year 2016 over Alerce glacier.

Here we summarize the most relevant model components. Snow accumulation $C_{(x,y,t)}$ for all grid cells $(x, y)$ and all time steps $(t)$ was calculated based on precipitation $P_{(t)}$ occurring below a threshold air temperature of $1.5\,°\mathrm{C}$ (Hock, 1999). Accumulation distribution $D_{s(x,y)}$ was inferred based on a spatial distribution pattern derived from winter snow measurements and topographic parameters (slope, curvature) to account for small-scale snow redistribution (Huss et al., 2008; Sold et al., 2016).

$$C_{(x,y,t)} = P_{(t)}C_{pre}D_{s(x,y)} \tag{5}$$

$P_{(t)}$ was the daily precipitation at Tepuel weather station (ID = 857990; http://www7.ncdc.noaa.gov/). The factor $C_{pre}$ allows adjusting precipitation measured at the weather station to the conditions on the glacier.

Snow and ice melt were calculated based on a simplified energy-balance formulation proposed by Oerlemans (2001), where the energy available for melt $\Psi_{d(x,y,t)}$ was defined as follows:

$$\Psi_{d(x,y,t)} = \tau(1 - \alpha_{(x,y,t)})I_{(x,y,t)} + (c_0 + c_1 T_{(t)}) \tag{6}$$

where $I_{(x,y,t)}$ is the potential direct solar radiation in $\mathrm{W\,m^{-2}}$, $\tau$ is the atmospheric transmission to solar irradiance, $T_{(x,y,t)}$ the air temperature and $c_0$ and $c_1$ represent parameters. $T_{(t)}$ was taken from the air surface temperature at Bariloche airport weather station (ID = 877650; http://www7.ncdc.noaa.gov/). After calibration of the model, $c_0 = -50\,\mathrm{W\,m^{-2}}$ and $c_1 =$



$12\,\mathrm{W\,m^{-2}\,{}^\circ C^{-1}}$. Potential direct solar radiation for all grid cells and days was calculated following Hock (1999). The local

surface albedo $\alpha_{(x,y,t)}$ was taken to be constant for bare-ice surfaces ($\alpha_{ice} = 0.34$), using most commonly applied literature value (Oerlemans and Knap, 1998; Cuffey and Paterson, 2010), for snow surfaces, $\alpha_{snow}$ was calculated based on the snow aging function proposed by Oerlemans and Knap (1998) with a maximum snow albedo ($\alpha_{max}$) of 0.8 and a variable minimum snow albedo ($\alpha_{firn}$, table 2). Glacier-wide mass balance changes between different values of $\alpha_{firn}$ are indicative of the sensitivity of glacier mass balance to a change in albedo that might occur in response to the darkening of the glacier surface.

## 235    3    Results and Discussion

### 3.1    PM concentration on Alerce glacier

PM concentrations in samples obtained in both field campaigns in the accumulation and the ablation zones are depicted in Fig. 2 as a function of pit or core depth. Alternating thin, high PM concentration layers and thick, low PM concentration layers are indicative of the seasonal glacier mass balance of more than one hydrological years, combined with the impact of

long-range transported aerosols and the re-suspension and re-deposition of local particles.

Thick and low PM concentration layers ($4.9\,\mathrm{mg\,kg^{-1}}$ to $51\,\mathrm{mg\,kg^{-1}}$, excluding two samples from ablation zone of higher concentration, $(128 \pm 2)\,\mathrm{mg\,kg^{-1}}$ and $(667 \pm 17)\,\mathrm{mg\,kg^{-1}}$) correspond to snow accumulated during autumn and winter (accumulation season). Meanwhile, thin and high PM concentration layers (with a wide range of concentration, between $(339 \pm 26)\,\mathrm{mg\,kg^{-1}}$ and $(9040 \pm 950)\,\mathrm{mg\,kg^{-1}}$), are related to the surface enrichment of PM content due to the melt of snow during spring and sum-

mer (ablation season) or fair-weather melt events during the accumulation season. In the longest snow/firn core (*Acc1-2016*), four high PM concentration layers were recognized. The first one at 3-5 cm deep represent the end of the ablation season of the hydrological year 2015-16, with a concentration of $(339 \pm 26)\,\mathrm{mg\,kg^{-1}}$. The next two thin layers with relative high PM concentration at $118\,\mathrm{cm}$ to $120\,\mathrm{cm}$ and $187\,\mathrm{cm}$ to $191\,\mathrm{cm}$ deep ($(365 \pm 26)\,\mathrm{mg\,kg^{-1}}$ and $(410 \pm 20)\,\mathrm{mg\,kg^{-1}}$, respectively), were, on the basis of microscopy analysis (see section 3.2), attributed to the resuspension and redeposition of dust and volcanic

ash, and also, possible melt events, related to fair-weather events during the accumulation season of the hydrological year 2015-2016. The deepest ($242\,\mathrm{cm}$ to $247\,\mathrm{cm}$ deep) thin, high PM concentration layer ($(1970 \pm 200)\,\mathrm{mg\,kg^{-1}}$) was interpreted as the surface at end of the ablation season of the hydrological year 2014-15. In addition to PM enrichment due to melting, this last layer suffered a direct ash fall event from Calbuco volcano, which erupted on 22-23 April 2015 (Reckziegel et al., 2016).

The same alternating pattern of low and high PM concentration layers was observed at other snow pits in the accumulation

zone (*Acc2-2016*, *Acc4-2017* to *Acc7-2017*). At the snow pit *Acc4-2017*, roughly the same location as *Acc1-2016*, the low PM concentration layer between the high concentration layers, is less than $30\,\mathrm{cm}$ thick, which illustrates the strong decrease in direct snow-fall during the accumulation season of the hydrological year 2016-2017. At site *Acc3-2016*, due to the slope of the site, there was no fresh snow accumulation, so it is interpreted as representative of the surface of the accumulation area at the end-of-ablation season.

In the ablation zones we collected samples in two different environments: accumulation pockets (*Abl1-2016*, *Abl3-2017*, *Abl4-2017*) and glacier ice with or without fresh snow on top of it (*Abl2-2016*, *Abl5-2017*, *Abl6-2017*).



The net accumulation layer of *Abl2-2016* goes only from $3\,\mathrm{cm}$ to $26\,\mathrm{cm}$ deep. This accumulation pocket completely disappeared in the summer 2016-17. In the 2017 campaign we took two samples in a different accumulation pocket (*Abl3-2017* and *Abl4-2017*). These sites had a negative net balance during hydrological year 2016-17, consequently the surface layer presented

the highest PM content observed in both campaigns ($(30\,000 \pm 5000)\,\mathrm{mg\,kg^{-1}}$ and $(12\,000 \pm 2000)\,\mathrm{mg\,kg^{-1}}$ respectively), due to the accumulation of PM depositions from several hydrological years (together with the impact of volcanic eruptions). In-situ stratigraphy revealed that in *Abl4-2017* site, the high concentration layer was on top of relatively low concentration, firn layer from 2015 winter, which means that, during the 2016-2017 ablation season, all the snow accumulated during 2016 winter was melted. Site *Abl3-2017* presented an even lower net balance, revealing older firn (winter 2014) below the surface

high concentration layer.

The fresh snow at the top of *Abl2-2016* shows slightly higher content of PM than fresh snow sampled on the accumulation zone ($(21.9 \pm 0.6)\,\mathrm{mg\,kg^{-1}}$). In the case of fresh snow on site *Abl5-2017* (with a higher PM content of $(1410 \pm 30)\,\mathrm{mg\,kg^{-1}}$) we could not discard, due to its thin thickness, some contamination with PM from the glacier ice. Glacier ice was highly heterogeneous (relatively pure ice mixed with debris and cryoconite holes), in consequence a substantial variability of PM

content over the ice surface was retrieved ($(200 \pm 20)\,\mathrm{mg\,kg^{-1}}$ to $(4300 \pm 900)\,\mathrm{mg\,kg^{-1}}$).

Figure 5 combines data from both field campaigns and groups PM concentrations according to the attributed date of the layers, but excludes glacier ice samples, which cannot be assigned to an specific year/season. It must be noted that PM content varies over several orders of magnitude ($1.3\,\mathrm{mg\,kg^{-1}}$ to $21.9\,\mathrm{mg\,kg^{-1}}$ on fresh snow, to up to $(30\,000 \pm 5000)\,\mathrm{mg\,kg^{-1}}$ in end-of-summer layers of the ablation zones). As discussed in section 3.3, this is one of the main causes of the albedo values

variation.

The alternation of thin and high PM concentration with thick and low PM concentration is partially due to seasonality, as explained above. But in addition to seasonality, there is a large spatial heterogeneity, specially during spring/summer (in winter, abundant fresh snow covers the glacier and gives a more homogeneous PM content and albedo distribution, as observed in other glaciers, Brock et al., 2000). The spatial variation is not only between the ablation and accumulation zones of the glacier. The

interaction between glacier topography and prevailing winds produce accumulation pockets and windswept ridges, which have contrasted snow accumulation values. These areas of higher and lower accumulation lead to a wide range of spectral albedos. The detailed variations in PM concentrations, and therefore in the albedo, need to be accounted for in a detailed mass balance of the glacier (see section 3.4).

Field observations on Monte Tronador in 2013 and 2014 confirmed the presence of volcanic ash in the atmosphere, derived

from re-suspension of volcanic ash. The magnitude of resuspension events in Andean Patagonia, a region with strong, persistent westerlies and low seasonal humidity, is well known. These events produce huge ash clouds that may be confused with true volcanic plumes, they remobilize ash tenths of kilometres away (Toyos et al., 2017). In particular, the deposits of volcanic ash that are covered by snow during the winter in the high mountain usually become exposed to remobilization during the summer, travelling through the atmosphere and redepositing at considerably high altitudes.

The 2011 Puyehue-Cordón Caulle eruption produced several ashfall events during the second semester of 2011, by January 2012 explosive activity had declined. As a consequence, thick deposits of tephra with different grain size covered an extended





area in Argentina (see Fig. 2, Alloway et al., 2015). Calbuco eruption (April 2015) was active during a shorter period, but due to its location and predominant wind direction also affected Monte Tronador (Romero et al., 2016; Reckziegel et al., 2016).

Direct ash deposition and re-suspension events can affect the glacier surface in different ways. Continuous, thick layers of
ash (few millimeters to few centimeters) have shown to behave as an isolating layer when deposited over snow, in a similar way as in debris-covered glaciers (Brock et al., 2007), which reduces ablation. But on the other hand, a thinner or disperse deposit may have the opposite effect, lowering the surface albedo of the glacier and increasing its melting. The effect of ash (or other PM, for instances from biomass burning events) deposition during autumn or winter can extend a few days until the next snow event, which covers the dark surface with the highly reflecting surface of fresh snow (see Fig. 7, Córdoba et al., 2015).
But during spring and summer, warmer temperatures and fewer snow events result in an increase of ablation processes over accumulation. Snow melting can flush some of the smaller, hydrophilic PM, but larger particles (or less water-soluble small particles) are concentrated in the glacier surface (Conway et al., 1996; Xu et al., 2012; Doherty et al., 2013; Li et al., 2017; Skiles and Painter, 2017), producing up to two orders of magnitude of surface enrichment of PM content (Doherty et al., 2016). Resuspension and surface enrichment explain the observed alternating thin, high PM concentration layers and thick, low PM
concentration layers. They also impact the spatial variability of albedo on the glacier surface during summer (Fig. 1) (Brock et al., 2000).

## 3.2   PM characterization

Three main types of particles were identified in samples collected in the field: mineral dust, volcanic ash and crystals derived from ash-fall events, and carbonaceous particles.
Based on glass morphology, SEM images, and energy dispersive spectroscopy (EDS) microanalysis performed on selected fragments, we were able to identify the presence of volcanic glass derived from Cordón Caulle 2011 (CC) and Calbuco 2015 (Cal) eruptions. Isopach maps for both eruptions (Alloway et al., 2015; Villarosa et al., 2016; Reckziegel et al., 2016) show that Monte Tronador was reached by different plumes from ash fall events marginally, further confirming that most of the volcanic ash identified in the filters derive from these two recent eruptions. Though both eruptions deposited pumiceous ash east of the
Andes in Patagonia, they can be distinguished by petrographic and morphological characteristics of the glass fragments (Fig. 6). CC glass is very fine-grained colourless glass (rhyolitic) while Cal pumice is light to pale brown, clear glass (dacitic to andesitic). SEM images show the presence of irregular glass fragments, with evidence of bubble coalescence, flat or slightly curved platy glass shards that are most probably pieces of broken thin vesicle walls and triangular (in cross section) to Y-shaped particles, which are vesicle walls from the junction of three adjacent vesicles (Fig. 7). EDS analysis of individual fragments of
glass from these samples were performed, and were compared with the composition of volcanic glass from samples collected in nearby locations during direct ash-fall events. Results confirm the presence of glass shards from 2015 Cal and 2011 CC eruptions (Fig. 8). One of the samples described under microscope, corresponds to a sub-surface sample from site *Abl3-2017*, it was dated as winter snow from 2014, previous to 2015 Calbuco eruption, and approximately 75 % of the observed particles correspond to fine-grained colourless pumiceous ash. EDS of individual fragments confirmed that ash on that sample
corresponded to CC eruption, as expected.



Another evidence of the presence of volcanic material within the PM collected in the study area are crystals from pyroclastic origin. They are clearly identified as they are partially surrounded by or associated with patches of glass and they are irregular in shape. Crystals that are not directly derived from CC 2011 or Cal 2015 are more or less rounded due to erosion and transport and they exhibit a dull lustre, and they are identified as mineral dust.

Another identified PM component is charcoal, present as black, elongated, brittle fragments. In addition, some of the samples showed evidence of the presence of BC particles, identified by their characteristic shape (carbon spherules of $100\,\mathrm{nm}$ to $200\,\mathrm{nm}$ in aggregates of different morphology). Carbon content by EDS could not be used to confirm the identity of BC particles due to the usage of carbon tape to fix the particles for SEM imaging.

The predominance of volcanic glass in the collected PM indicates the need to take into account the effect of volcanic ash in the albedo of seasonal snow and glaciers of the region, which can be frequently affected by volcanic eruptions. It must be emphasized that ash from CC and Cal eruptions was observed in most of the samples, not only in layers dated immediately after the eruptions, but also many years after direct deposition.

### 3.3 Albedo: measurements and models

Table 1 shows measured and modeled albedo values for six sites (two from the first field campaign, 2016, and four from the latter, 2017), together with different measured properties of the snow topmost layer and site.

Reported values of measured albedo include shadow corrections, although these corrections were quite small in all cases (below 3.5 % for worst conditions in the 2016 campaign and below 2 % for the 2017 campaign). In some cases (site *Acc3-2016*) the corrections in the measured incoming and reflected radiation are higher (10 to 14 %), but they largely balance out. For the 2016 campaign, the reported measured albedo is a single measurement and is informed together with its instrumental uncertainty. It must be noted that for this campaign the reported uncertainty reached values as high as 15 % for worst conditions (low incident radiation and low albedo, as in *Acc3-2016*) or around 2 % for best conditions (clear sky, high albedo). For the 2017 campaign the instrumental uncertainty was lowered by improving the accuracy of the digital multimeter used with the pyranometer, achieving uncertainties lower than 3.5 % (worst conditions) or lower than 1.2 % (best conditions).

Results from the 2017 campaign, obtained using the improved support, shed light on the reproducibility of albedo measurements. For this campaign, we found that repeated albedo measurements in the same site have a standard deviation corresponding around 5 to 10 % of the average values. This range could be partly due to the leveling of the support, or to inherent variability of the measurement at these sites (specially to differences in the solar irradiance for situations with rapid changes in cloudiness).

Regarding snow grain sizes, it is relevant to notice the range of observed average radius. In fresh snow samples from the accumulation zone (sites *Acc5-2017* and *Acc6-2017*) we found an average radius of $(151 \pm 41)\,\mu\mathrm{m}$, whereas in samples of older firn in the ablation zone (or sub-surface snow/firn in the accumulation zone) we measured values usually around $(1000 \pm 200)\,\mu\mathrm{m}$. It must be emphasized here that the developed method for characterizing snow grains for the 2017 campaign allows us to measure the size distribution and to assess the relevance of different grain shapes (when necessary). It has been shown that the shape of the snow grains can significantly affect snow albedo (Libois et al., 2013; He et al., 2017). Except for fresh snow (snow less than one day old), where it is possible to still distinguish crystal fragments, in both campaigns the

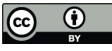


**Table 1.** Measured and modeled snow albedo for six sites (two in 2016 campaign and four in 2017 campaign). For 2016 campaign the measured albedo is a single measurement and is informed together with its instrumental uncertainty. For 2017 campaign, we report the average and the standard error of the average for several repetitions. For modeled albedo, sensitivity to different input parameters is reported, as an estimation of albedo uncertainty.

| Site | Surface | $\alpha_{meas}$ | $\alpha SNICARv2$ | $\alpha SNICARv2.1$ | $\alpha SNICARv2.1$ sens. |
|---|---|---|---|---|---|
| Acc2-2016 (accum. zone)<br>April 12th 2016<br>14:47 (UTC - 3)<br>Zenith: 51.98°<br>Effective angle: 51.98°<br>Clear Sky | Recent snow<br>Layer: (6±1) cm<br>Grain size: (1000±200) µm<br>Snow density: 300 kg m⁻³<br>PM: (22.0±0.6) mg kg⁻¹<br>Slope: 0° | 0.626±0.011 | Direct: 0.583<br>Diffuse: 0.655 | Direct: 0.573<br>Diffuse: 0.748<br>W.Aver.: 0.590 | Grain size: ±0.028<br>PM content: ±0.001<br>Layer thickness: ±0.013<br>100 µg kg⁻¹ BC: −0.017 |
| Acc3-2016 (accum. zone)<br>April 12th 2016<br>16:55 (UTC - 3)<br>Zenith: 66.04°<br>Effective angle: 68.7° to 73.6°<br>Overcast sky (approx. 100 % diffuse rad.) | Dirty summer snow<br>Layer: (0.3±0.1) cm<br>Grain size: (1000±200) µm<br>Snow density: 500 kg m⁻³<br>PM: (7800±1500) mg kg⁻¹<br>Slope: 11° | 0.257±0.041 | Direct: 0.435<br>Diffuse: 0.364 | Direct: 0.445<br>Diffuse: 0.359<br>W.Aver.: 0.359 | Grain size: ±0.002<br>PM content: ±0.002<br>Layer thickness: +0.007 (0.2 cm), −0.001 (0.4 cm)<br>Snow density: ±0.002<br>% Diff. Rad.: +0.001 (89 % diff. rad.)<br>20 mg kg⁻¹ BC: −0.049 |
| Abl3-2017 (accum. pocket on abl. zone)<br>April 3rd 2017<br>13:11 (UTC - 3)<br>Zenith: 47.57°<br>Effective angle: 56.5° to 59.9°<br>Overcast sky (89 % to 95 % diffuse rad.) | Dirty snow<br>Layer: (0.3±0.1) cm<br>Grain size: (1020±160) µm<br>Snow density: 500 kg m⁻³<br>PM: (30000±5000) mg kg⁻¹<br>Slope: 15° | 0.371±0.011 | Direct: 0.374<br>Diffuse: 0.360 | Direct: 0.381<br>Diffuse: 0.354<br>W.Aver.: 0.356 | Grain size: ±0.001<br>PM content: ±0.001<br>Layer thickness: ±0.00001<br>% Diff. Rad.: ±0.003<br>Effective angle: ±0.001<br>20 mg kg⁻¹ BC: −0.015 |
| Abl4-2017 (accum. pocket on abl. zone)<br>April 3rd 2017<br>13:30 (UTC - 3)<br>Zenith: 46.95°<br>Effective angle: 57.1° to 60.1°<br>Overcast sky (approx. 100 % diffuse rad.) | Dirty snow<br>Layer: (0.10±0.05) cm<br>Grain size: (740±170) µm<br>Snow density: 500 kg m⁻³<br>PM: (12250±2050) mg kg⁻¹<br>Slope: 15° | 0.266±0.008 | Direct: 0.379<br>Diffuse:0.375 | Direct: 0.384<br>Diffuse: 0.368<br>W.Aver.: 0.368 | Grain size: ±0.001<br>PM content: ±0.006<br>Layer thickness: +0.030 (0.05 cm), −0.008 (0.15 cm)<br>Snow density: ±0.008<br>% Diff. Rad.: −0.007 (89 % diff. rad.)<br>20 mg kg⁻¹ BC: −0.050 |
| Acc5-2017 (accum. zone)<br>April 5th 2017<br>14:26 (UTC - 3)<br>Zenith: 48.20°<br>Effective angle: 49.4° to 52.3°<br>Cloudy sky (34 % to 48 % diffuse rad.) | Recent snow<br>Layer: (9±1) cm<br>Grain size: (150±40) µm<br>Snow density: 300 kg m⁻³<br>PM: (1.28±0.03) mg kg⁻¹<br>Slope: 5° | 0.814±0.013 | Direct: 0.788<br>Diffuse: 0.860 | Direct: 0.778<br>Diffuse: 0.910<br>W.Aver.:0.828 | Grain size: ±0.015<br>% Diff. Rad.: ±0.005<br>PM content: ±0.001<br>Effective angle: ±0.001<br>Layer thickness: ±0.002<br>100 µg kg⁻¹ BC: −0.022 |
| Acc6-2017 (accum. zone)<br>April 5th 2017<br>14:48 (UTC - 3)<br>Zenith: 49.35°<br>Effective angle: 51.0° to 53.9°<br>Cloudy sky (34 % to 48 % diffuse rad.) | Recent snow<br>Layer: (9±1) cm<br>Grain size: (150±40) µm<br>Snow density: 300 kg m⁻³<br>PM: (3.9±0.2) mg kg⁻¹<br>Slope: 5° | 0.757±0.026 | Direct: 0.786<br>Diffuse: 0.856 | Direct: 0.776<br>Diffuse: 0.905<br>W.Aver.: 0.825 | Grain size: ±0.015<br>% Diff. Rad.: ±0.004<br>PM content: ±0.001<br>Effective angle: ±0.001<br>Layer thickness: ±0.002<br>100 µg kg⁻¹ BC: −0.021 |





observed snow/firn grains were rounded. This is related with the temperate climate at Monte Tronador, where snow temperature is above $-5\,°C$ and the temperature gradient is low. Also, the presence of meltwater within the snow layers enhance the rate at which grains become rounded, because the grains melt first at their extremities. Finally, the average grain size increases because the smaller grains tend to melt before the larger ones (Cuffey and Paterson, 2010). Hence, we assumed spherical grains for all modeled albedo calculations.

Table 1 also reports modeled albedo results for each site. Results of the updated model (SNICARv2.1) were calculated with the direct and diffuse spectra estimated for the specific sky conditions, as detailed in section 2.3. The weighted average of pure direct and pure diffuse radiation albedos represents the net albedo of snow for total incident radiation. For comparison, results from SNICARv2 with the available standard spectra (mid-latitude clear-sky direct radiation spectrum or overcast sky diffuse radiation spectrum) are presented. As expected, for clear-sky conditions (site *Acc2-2016*) the pure direct albedo from SNI-

CARv2 is similar to the weighted average from SNICARv2.1. The pure diffuse albedo from both models differ significantly, but the fraction of diffuse radiation is very low, and hence its contribution to net albedo is also low. For overcast conditions (*Acc3-2016*, *Abl3-2017* and *Abl4-2017*), the pure diffuse albedo from both models is also similar, and weighted average albedo from SNICARv2.1 is coincident with the pure diffuse albedo. For both models, the diffuse radiation spectrum for overcast conditions is coincident with global solar radiation spectrum (see Fig. 4), which explains the similar results. Finally, partly cloudy

skies (sites *Acc5-2017* and *Acc6-2017*) are the main reason for the development of SNICARv2.1. For these cases, pure direct and pure diffuse albedo differ much more than the associated uncertainties, and pure diffuse albedo from SNICARv2.1 also differs from that from SNICARv2. These differences are also evident from the comparison between the diffuse radiation spectra for partly cloudy skies developed for SNICARv2.1 and the diffuse spectra for overcast skies used in SNICARv2 (Fig. 4). For these sites, SNICARv2 cannot give a good approximation. For *Acc5-2017* SNICARv2.1 weighted average albedo seems

a good approximation of the measured albedo. For *Acc6-2017*, measured albedo is lower than pure direct and pure diffuse albedo, so both models give higher estimates for this site. As discussed below in this section, the effect of the diffuse radiation fraction does not seem to be the main source of this disagreement.

  The updated model reproduces quite well the main features of the measured albedo (with a larger discrepancy for sampling site *Acc3-2016*). One of the most important parameters affecting albedo is PM content: the measurements with lower albedo

values ( $\alpha_{meas} < 0.4$ ) correspond to sites with the highest PM content (*Acc3-2016*, *Abl3-2017* and *Abl4-2017*), whereas the remaining sites have much lower PM content (fresh snow) and $\alpha_{meas} > 0.6$. It must be noted that for high PM content, a further increase in particle content does not significantly affect the albedo: our simulations for site *Acc3-2016*, with $(7800 \pm 1500)\,\mathrm{mg\,kg^{-1}}$ of PM match closely those for sites *Abl3-2017* and *Abl4-2017*, with $(30\,000 \pm 5000)\,\mathrm{mg\,kg^{-1}}$ and $(12\,250 \pm 2050)\,\mathrm{mg\,kg^{-1}}$ of PM, respectively. The same effect is noticed when simulating the impact of the possible presence

of BC on snow. For sites with low PM content, an increment of $100\,\mathrm{\mu g\,kg^{-1}}$ of aged BC has a relevant impact on modeled albedo (between $-0.017$ and $-0.022$ for the studied sites). However, for sites with higher PM content, much higher BC concentrations were needed in order to observe a relevant effect in modeled albedo (for a $20\,\mathrm{mg\,kg^{-1}}$ increment of BC, we calculated an effect of $-0.015$ to $-0.050$ in calculated albedo). Ginot et al. (2014) have already reported simulation results for Mera Glacier, Nepal, that showed that the effect of dust and BC content on albedo and potential melting of snow are non-additive.





Our results show that for site *Acc3-2016* $20\,\mathrm{mg\,kg^{-1}}$ of BC represent a lowering of $-0.049$ of albedo for snow containing $7800\,\mathrm{mg\,kg^{-1}}$ of volcanic ash, but the impact increases to $-0.057$ if the snow contains only $6300\,\mathrm{mg\,kg^{-1}}$ of volcanic ash (which is possible due to the uncertainty in gravimetric PM content).

In the other hand, comparison between sites with low PM content shows that snow grain size has a remarkable effect, as previously reported (Wiscombe and Warren, 1980; Hadley and Kirchstetter, 2012). Fresh snow with small grain size presents

$\alpha_{meas} \approx 0.8$ (sites *Acc5-2017* and *Acc6-2017*), but snow with similar PM content that has aged a few days presents $\alpha_{meas} \approx$ 0.6 (site *Acc2-2016*).

The last column in Table 1 reports the results of sensitivity studies to evaluate the impact on the calculated albedo of the uncertainty in key input parameters. The parameters have been modified in ranges allowed by the uncertainty of the input parameters. For each site, we studied PM content and grain size impact, together with other parameters that could be relevant

at each site.

Concerning grain size uncertainty, it is clear that the impact on albedo is much larger when PM content is low (sites *Acc2-2016*, *Acc5-2017* and *Acc6-2017*). For low PM content sites, the effect is comparable to experimental uncertainty, and is relevant both for sites with finer and coarser grain sizes snow. For sites with high content of PM the uncertainty of grain size do not have an appreciable effect. Volcanic ash content uncertainty does not have a relevant impact for any of the sites, although it

is larger for site *Abl4-2017*. However, as previously mentioned, the presence of BC (not yet quantified in these samples) could have a more relevant impact on albedo. For instance, it could explain the difference between measured and modeled albedo for site *Acc6-2017*, and the difference with site *Acc5-2017*.

Regarding the impact of the uncertainty of layer thickness, the results show that several factors determine the relevance of this parameter. The impact is maximum for very thin layers, especially when the underlying layer has a significantly different

albedo (i.e., PM content), and its minimum for the thicker layers, or for intermediate thicknesses with high PM content (i.e., low penetration of incident light). The impact of uncertainty of snow density was not studied in detail, but the impact is inverse to that of the thickness of the layer. Hence, we report only the moderate impact of snow density uncertainty for site *Abl4-2017*.

The impact of the uncertainty of the diffuse to global irradiance ratio is moderate but appreciable, which emphasizes the relevance of measuring the ratio on the field. Finally, the impact of the uncertainty of the incidence angle is low, and not

appreciable for this range of experimental albedo uncertainty.

Another possible reason for disagreement between modeled and measured albedo, especially for aged snow, is surface roughness. Millimeter scale surface roughness due to snow aging have shown to reduce albedo, especially in the infrared region, due to multiple reflections in the cavities (Pirazzini et al., 2015). Computer simulations have studied the parameters that determine the magnitude of the effect of sastrugi (centimeter-scale roughness) on albedo (Zhuravleva and Kokhanovsky,

2011). Quantification of the impact of surface roughness of snow in measured albedo is out of the scope of this work, but it must be remarked that sites with higher PM content, which has been under longer snow metamorphosis processes (*Acc3-2016*, *Abl3-2017* and *Abl4-2017*), presented higher surface roughness.

Literature values of snow albedo depend mainly on the PM content. Two other studies that found snow albedo ranges similar to our measurements are connected with local/regional transport of dust (Painter et al., 2012; Wittmann et al., 2017). Young





et al. (2014) modeled the direct deposition of volcanic ash from Redoubt volcano 2009 eruption on Arctic snow, finding similarly high albedo reductions. Sicart et al. (2001) also found a similar albedo range at Zongo glacier, but their lower values of albedo are not attributed to PM surface enrichment but to very thin snow layers over dirty ice.

Recent studies in Chilean Andes measured or modeled small reductions on snow albedo, due to traffic related BC (Cereceda-Balic et al., 2018) or to a combination of urban BC and dust from desert regions (Rowe et al., 2019). Similarly, studies on Mera 440 Glacier, Nepal (Ginot et al., 2014), and at several sites at Tibetan Plateau (Zhang et al., 2018) found small albedo reductions due to BC and dust, and almost negligible effects of impurities in Greenland (Carmagnola et al., 2013; Wright et al., 2014).

### 3.4 Albedo and glacier mass balance

Table 2 shows the glacier-wide annual and winter mass balance, Equilibrium Line Altitude (ELA) and Accumulation Area Ratio (AAR) for different values of old snow albedo ($\alpha_{firn}$). Figure 9 shows the change in cumulative glacier-wide surface 445 mass balance and ablation and the annual mass balance elevation gradient for the different values of $\alpha_{firn}$. The mass balance sensitivity to albedo change, defined as the change in glacier-wide mass balance per 0.1 of $\alpha_{firn}$ decrease is around of $-0.6\,\mathrm{m\,w\,e/yr}$ and $-0.07\,\mathrm{m\,w\,e/yr}$, for annual and winter mass balance, respectively (Table 2). Firn albedo or old snow albedo have a considerable effect on the surface mass balance of Alerce glacier (Fig. 9 A) increasing the amount of melt during the ablation period, from almost 2.4 m w.e. to more than 4.6 m w.e. when $\alpha_{firn}$ is decreased from 0.7 to 0.3 (Fig. 9 B). Although 450 the accumulation of the glacier does not change (the amount of precipitation for the different run test is the same) there is a decrease in the winter (accumulation) mass balance due to the albedo effect over ablation episodes at the begging of the accumulation season (Fig. 9, Table 2). The decrease in the old snow albedo had an impact all over the glacier, decreasing the surface mass balance at all elevation range. Other glaciological parameters related to the surface mass balance of the glacier, like the ELA or AAR also seems to be profoundly impacted with the decrease of albedo, with a total increase of ELA of $250\,\mathrm{m}$ 455 and a decrease of AAR of more than 50% when the old snow albedo changes from 0.7 to 0.3. Nevertheless, since both ELA and AAR depends on the hypsometry of the glacier the change do not increase constantly.

**Table 2.** Albedo values for ice ($\alpha_{ice}$), old snow (firn, $\alpha_{firn}$) and fresh snow ($\alpha_{snow,max}$) used for the sensitivity study of Alerce glacier-wide mass balance to change in the albedo. The winter and annual glacier-wide surface mass balance, ELA and AAR for each simulation is presented.

| $\alpha_{ice}$ | $\alpha_{firn}$ | $\alpha_{max}$ | Wint. MB (m w.e.) | Annu. MB (m w.e.) | ELA (m) | AAR (m) |
|---|---|---|---|---|---|---|
| 0.35 | 0.3 | 0.8 | 3.32 | -1.28 | 2165 | 22.30 |
| 0.35 | 0.4 | 0.8 | 3.4 | -0.69 | 2125 | 34.6 |
| 0.35 | 0.5 | 0.8 | 3.48 | -0.08 | 2055 | 50.3 |
| 0.35 | 0.6 | 0.8 | 3.55 | 0.56 | 1935 | 70.5 |
| 0.35 | 0.7 | 0.8 | 3.61 | 1.22 | 1915 | 78.8 |





To give physical meaning to the albedo values presented in Fig. 9 and Table 2, we can use as a reference the daily-averaged albedo values modeled with SNICARv2.1 for some of the sampling sites in Table 1.

The $\alpha_{max} = 0.8$ used in the mass balance model is equivalent to the daily average of 0.805 for clear-sky conditions, 0.803
for overcast sky, and 0.835 for 33% of cloudiness, modeled for fresh recent snow, with very low PM content at site *Acc5-2017*. The $\alpha_{firn} = 0.6$ scenario in Table 2 is similar to the daily average of 0.612 for clear-sky conditions, 0.605 for overcast sky, and 0.637 for 33% of cloudiness, modeled for recent aged snow with low PM content (*Acc2-2016*). Although it represents intermediately aged snow, it can serve as an example of a firn surface with low PM content, a situation where no ash fall occurred at Monte Tronador. The $\alpha_{firn} = 0.4$ scenario in Table 2, is similar to the modeled daily average of 0.407 for clear-
sky conditions, 0.368 for overcast sky, and 0.382 for 33% of cloudiness of the firn with very high PM content (*Abl4-2017*). These values are representatives of the firn albedo during summer for the years 2016 and 2017. The other scenarios are used to depict intermediate or more extreme situation and to analyze the role of albedo change in the surface mass balance of the glacier.

Our $\alpha_{firn}$ analysis allows us to estimate the impact of volcanic ash on the surface mass balance of Alerce glacier. In
absence of volcanic eruptions, if we assume that other local or regional PM sources (mineral dust, biomass burning, etc.) do not affect significantly fresh snow albedo, it is expected that the summer $\alpha_{firn}$ over glacier surface is similar to $\alpha_{firn} = 0.6$ scenario. Although we could not sample summer firn layers previous to 2015 Cal eruption to test this hypothesis, this first order assumption would mean, that volcanic ash are responsible for a $1.25\,\mathrm{mwe}$ decrease in the glacier-wide annual mass balance (or a 36 % increase in summer ablation), if we compared the $\alpha_{firn} = 0.6$ and $\alpha_{firn} = 0.4$ scenarios.

Although more sampling of firn/snow layer and further chemical analysis on the samples are needed to confirm that the decrease of albedo is only due to the effect of volcanic ash, we have shown that PM content (and hence $\alpha_{firn}$) varies largely over the glacier surface. Taking into account these spatiotemporal changes in albedo for glacier mass balance models is a defying task. Defining a low number of representative regions over the glacier surface is not an easy task, due to the already mentioned high heterogeneity. In addition, it would be difficult to regularly measure PM content (and/or albedo) on those
regions, due to the distances and path conditions on the glacier. Regional atmospheric models could be of help in predicting deposition of volcanic ashes, mineral dust, BC and other PM. But the spatial scale of those models ($\geq 1\,\mathrm{km}$) is too coarse to capture to reproduce the spatial variation of the albedo over the glacier.

These challenges have been acknowledged in literature, and several approaches have been followed to estimate snow/ice melting. The simplest approaches have used measured or modeled albedo changes together with measured or modeled solar
radiation to estimate melting, without taking into account spatial heterogeneity (in surface temperature, PM concentration, etc) (Ginot et al., 2014; Zhang et al., 2018). For Mera glacier, Ginot et al. (2014) calculate that BC and dust are responsible of approximately 26 % of total melting. Zhang et al. (2018) do not report the effect on melt rates but only the impact on seasonal snow cover duration, and hence the results are not easy to compare with ours. Painter et al. (2013) used a glacier mass balance model similar to ours, but introducing temperature anomalies (due to BC radiative forcing) to estimate mass balance changes.
They used several approximations to postulate BC concentrations over the glaciers based on limited ice cores. Their results are difficult to compare to ours due to the different approach, they analyze general mass balance trends over two centuries.





Flanner et al. (2007) and Ménégoz et al. (2014) used emission inventories and general circulation models to study deposition of BC (and mineral dust, in the latter work) and its radiative forcing. The spatial resolution of their simulations make difficult the comparison with field PM concentration measurements, and hinder the accuracy of quantitative mass balance calculations

(Ménégoz et al., 2014; Qian et al., 2015). Young et al. (2014) used modeled ash deposition, SNICAR and a restricted degree-day radiation balance. They found melt rates between 140 % and 320 % higher than for pure snow, although the low spatial resolution of the simulations ($\approx 18\,\mathrm{km}$) may affect the precision of the results. Vionnet et al. (2012) used the detailed snow model CROCUS implemented on the soil model SURFEX to study the snowpack on the Grandes Rousses mountain range in the French Alps . They used a high resolution DEM ($150\,\mathrm{m}$) together with meteorological forcing from interpolation of

SAFRAN atmospheric reanalysis. They main weakness is that at that moment CROCUS did not explicitly treated PM in snow (it was only implicitly included in the parametrization of snow albedo changes with snow aging).

There are also some examples on literature that studied the coupling of meteorological models with glacier or snowpack models. Different authors studied climate feedback effects on Karakoram glaciers (Collier et al., 2013) and in the Svalbard glaciers (Aas et al., 2016), and the snowpack in Antarctica (Vionnet et al., 2012). The authors suggest that the next steps would

be to couple a regional atmospheric model with the ability of prognosis of PM deposition (such as Ménégoz et al. (2014)) with a high resolution glacier mass balance model (such as ours or CROCUS implementation on SURFEX (Vionnet et al., 2012)), and including explicit treatment of PM effect on snow albedo (such as SNICAR or recent CROCUS implementations (Tuzet et al., 2017)).

## 4 Conclusions

Our study combines field observation and modeling activities to analysis the role of PM over the albedo of Alerce glacier in Monte Tronador.

PM content of the samples varied in a wide range, from lowest to highest: fresh snow ($1.1\,\mathrm{mg\,kg^{-1}}$ to $21.9\,\mathrm{mg\,kg^{-1}}$), old winter snow/firn ($4.9\,\mathrm{mg\,kg^{-1}}$ to $51\,\mathrm{mg\,kg^{-1}}$, except from some samples from ablation zone), and thin, darker layers with contribution of local/regional resuspension of dust/ashes ($365\,\mathrm{mg\,kg^{-1}}$ to $410\,\mathrm{mg\,kg^{-1}}$) or with high PM enrichment due to

spring and summer ablation ($339\,\mathrm{mg\,kg^{-1}}$ to $9040\,\mathrm{mg\,kg^{-1}}$, reaching even $12\,250\,\mathrm{mg\,kg^{-1}}$ to $30\,000\,\mathrm{mg\,kg^{-1}}$ in the ablation zone). Microscopical characterization of PM showed that the major component on snow and firn layers after 2014 and also glacier ice surface is volcanic ash, not only from the recent Calbuco eruption (2015), but also from the Cordón Caulle eruption (2011). Minor contributions of mineral dust and Black Carbon were also detected.

The major presence of volcanic ash in all studied samples indicate that the effect of nearby volcanic eruptions are expected

not only inmediately after direct deposition, but also many years later, due to surface enrichment and wind resuspension and redeposition. The spatial and temporal distribution of PM is highly heterogeneous, due both to seasonality and to the combination of glacier topography and prevailing wind direction. These facts need to be accounted for when studying the effect of snow albedo on glacier mass balance.





The measured snow albedo also varied in a wide range (0.26 to 0.81), similar to that of other glaciers with dust of vol-
canic concentrations in same order of magnitude. We found that rapid changes in cloudiness hinder the repeatability of albedo
measurements and may difficult the comparison with modeled albedo. Nevertheless, comparison of measured and modeled
snow albedo showed a good match, and illustrates the effect of PM content and composition (i.e., BC versus dust or volcanic
ash), snow grain size, layer thickness, and cloudiness on snow albedo. To evaluate the latter, we updated the SNICAR snow
albedo model, to accurately represent the effect of cloudiness on direct and diffuse solar spectra (SNICARv2.1). This update
improved considerably the match of measured and modeled albedo for partially cloudy sky conditions. The effect of uncertain-
ties of field measurements of snow properties was evaluated for different types of samples (lower or higher PM content, grain
size, layer thickness, snow density, etc.), suggesting strategies to reduce uncertainty in snow albedo modeling or retrieval of
snow properties from measured albedo.

We showed that glacier-wide mass balance is highly sensitive to firn or old snow albedo changes. We find a glacier-wide
albedo change sensitivity of around of $-0.6 \, \mathrm{mwe/yr}$, mostly due to a higher ablation during spring and summer. Finally, we
suggest that the effect of volcanic ashes in Alerce glacier can be as high as a $1.25 \, \mathrm{mwe}$ decrease in the glacier annual mass
balance or a 34 % of increase in the melt during the ablation season. Nevertheless, a more accurate calculation of volcanic ash
impact would take into account the amount of other regional or local sources of PM present on the glacier in absence of such
volcanic eruptions, which cannot be estimated with the results of the field campaigns reported in this article.

To the best of our knowledge, this work is the first study of PM content and snow albedo on Argentinian glaciers. Our results
highlight the need of considering appropriately the effect of volcanic eruptions on snow albedo and glacier mass balance even
years after the eruption events. We suggest possible future steps to improve prognosis ability and mass balance accuracy, using
a combination of measurements and modeling.

*Code and data availability.* The complete set of field measurements are available from the corresponding author on reasonable request. The
code of SNICAR v2.1 is available online at https://github.com/EarthSciCode/SNICARv2.git

*Author contributions.* JGC, LD and LR designed the field campaigns. HB designed and fabricated the supports for albedo measurents. JGC
and FB collected and filtered the snow samples and performed the albedo measurements, with assistance of LR and his team. JGC performed
the gravimetry of filters and corrected albedo measurements. JGC updated SNICAR code in collaboration with CH. JGC performed modeling
related to solar incident spectra and SNICAR snow albedo. VO and GV characterized PM on the filters by binocular microscopy, SEM and
EDS. LR performed the modelling of glacier mass balance. JGC, LD and LR prepared the manuscript, with contributions of VO, GV and
CH.

*Competing interests.* The authors declare that they have no conflict of interest.



*Acknowledgements.* JGC, LR, VO and GV are members of Consejo Nacional de Investigaciones Científicas y Técnicas (CONICET). JGC, LD, FB and HB acknowledge funding from the Agencia Nacional de Promoción Científica and Tecnológica through project PICT 2016-
3590. This material is based upon work supported by the National Center for Atmospheric Research, which is a major facility sponsored by the National Science Foundation under Cooperative Agreement No. 1852977. The authors thank Claudio Bolzi and team (DES, CAC-CNEA) for sharing their equipment for albedo measurements. The authors thank the valuable field knowledge and collaboration provided by Hernan Gargantini, Mariano Castro, Ernesto Corvalán, Valentina Zorzut, and Inés Dussaillant. The authors acknowledge Dario R. Gómez for their valuable comments and contributions to the manuscript. The authors thank Adriana Dominguez and Lucia Marzocca (Laboratorio de Microscopía Electrónica, G. Materiales, CAC-CNEA) for SEM images and EDS analysis. Administración de Parques Nacionales kindly provided permission and logistical assistance to work at Cerro Tronador inside Parque Nacional Nahuel Huapi.



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



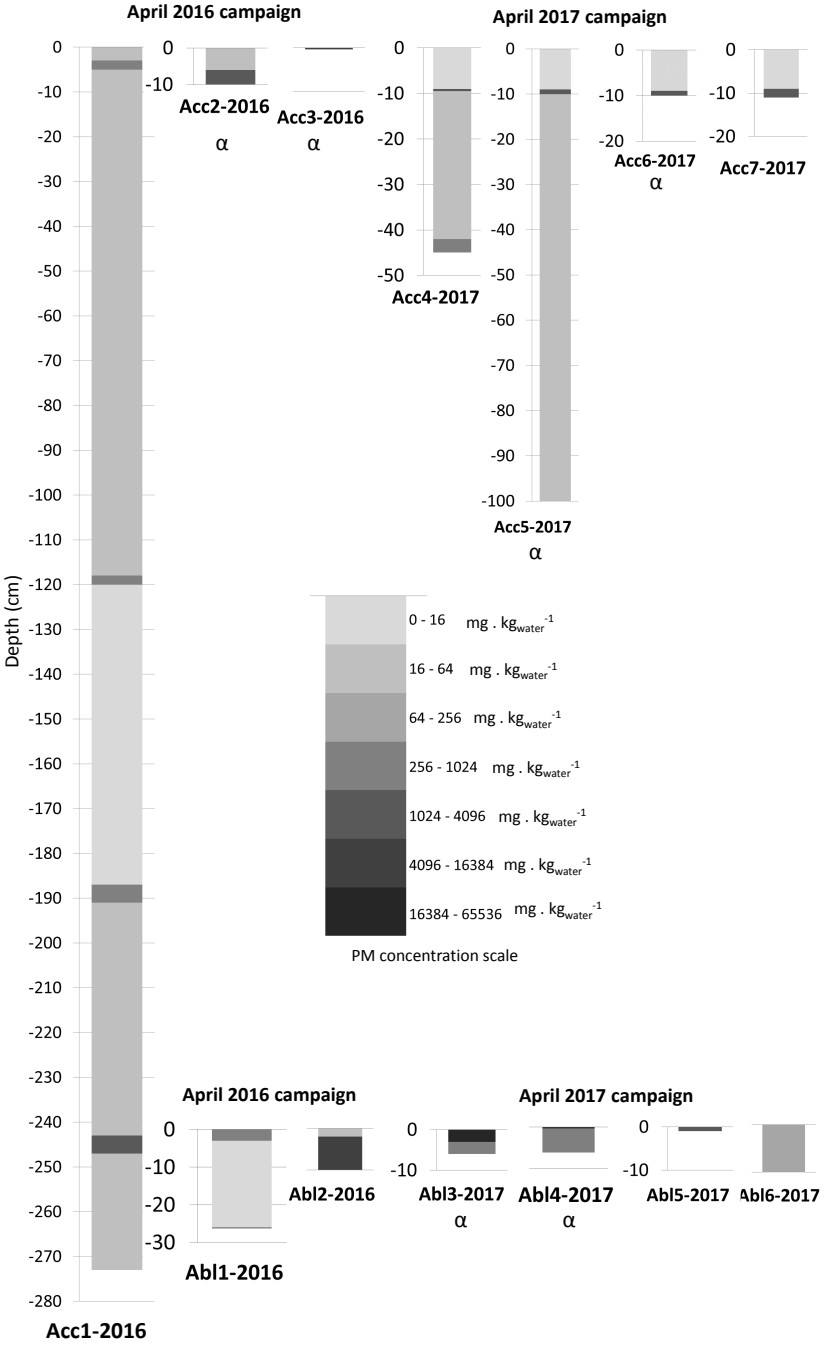

**Figure 2.** PM concentration (grayscale) as a function of pit depth for different sampling sites. Top panel: accumulation zone. Bottom panel: ablation zone. $\alpha$ symbol is used to highlight sites with concurrent albedo measurements. In sample *Abl2-2016*, the top rectangle corresponds to the average PM content of the first two layers (fresh snow and end-of-summer dark layer).




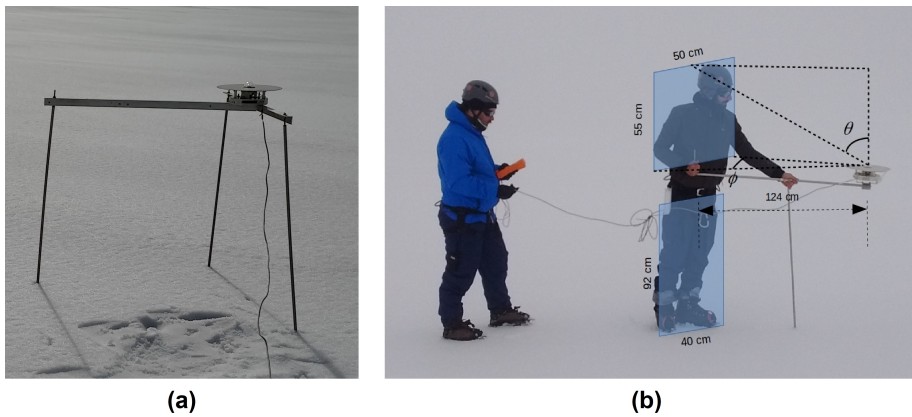

**Figure 3.** Albedo measurement equipment. (a) support used in the 2016 campaign. (b) support used in the 2017 campaign. The presence of the support and the observer is taken into account to correct the albedo measurement through the angles $\theta$ and $\phi$ and Eq. (S1) and (S2) in the Supplement.

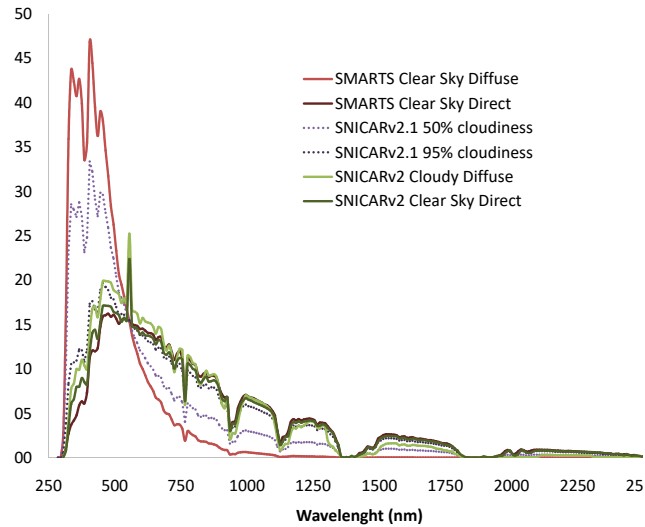

**Figure 4.** Different spectral distributions of sun radiation for SNICAR snow albedo model. SNICARv2 included two spectra for mid-latitude locations: one for overcast conditions (light green line), and one for clear sky conditions (dark green line). SMARTS diffuse (light red line) and direct (dark red line) clear sky spectra for one of our sampling sites are represented for comparison. Dotted lines represent spectra for partly cloudy conditions (SNICARv2.1).





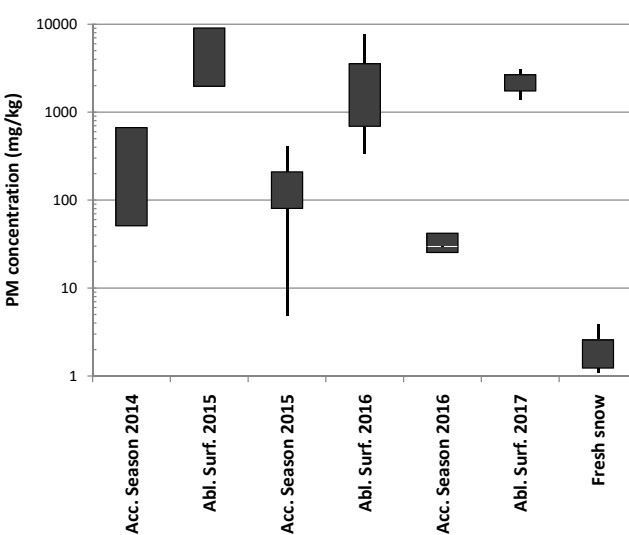

**Figure 5.** Seasonal range of PM concentration found on snow/firn samples. For accumulation season, the values represent the mean PM concentration in thick, low PM layers of snow/firn. For ablation season the values represents the surface PM concentration at the end of the season. The box encompasses one standard deviation of data, and whiskers represent minimum and maximum values (when $N > 2$). The plot includes data from both field campaigns, and excludes ablation ice samples, which cannot be assigned to an specific year/season. Fresh snow represent snow fallen a few days before field campaigns of 2016 or 2017.





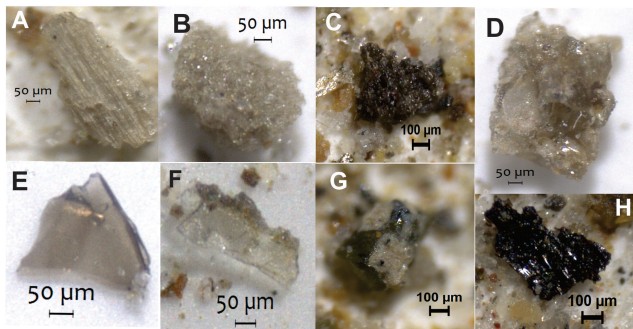

**Figure 6.** Stereo microscope images of juvenile glass fragments from ash fall events identified in the filters. Different morphologies are shown: A: Colourless glass fragment with elongate, thin, pipe-shaped vesicles (2017 end-of-summer dark layer, site *Acc7-2017*); B: Colourless pumice (surface ablation ice, site *Abl6-2017*). C: Dark brown fragment of vesicular glass (2017 end-of-summer dark layer, site *Acc7-2017*). D: Glass fragments with smooth, round surfaces formed by surface tension within still-molten, vesiculating droplets suggesting highly vesicular interior (2017 end-of-summer dark layer, site *Acc7-2017*). E and F: Two flat, tan glass shards derived from broken vesicle walls. Left: Y-shaped fragment formed where three bubbles were in close proximity (surface ablation ice, site *Abl6-2017*). Right: flat glass plate formed by the fragmentation of walls that enclosed large elongated, flattened vesicles as those shown above (fresh snow on top of ablation ice, site *Abl5-2017*). G: Pyroxene crystal with two patches of colourless glass with tiny dots of magnetite (2016 end-of-summer dark layer, site *Acc4-2017*). H: planar piece of charcoal with subtle striated surface texture and brilliant luster.



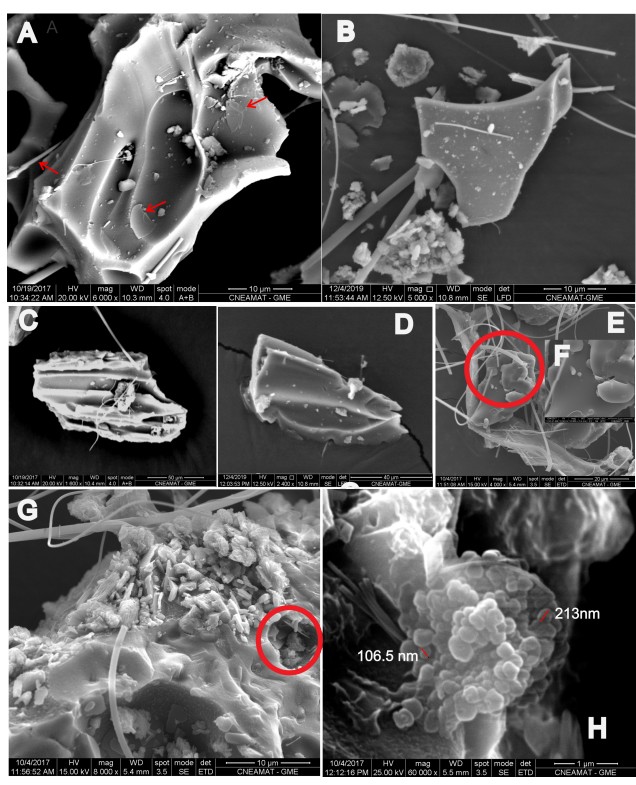

**Figure 7.** Scanning Electron Microscopy images of samples collected on Alerce glacier. A: irregular glass fragment with low vesicularity, evidence of bubble coalescence, and small, flat, platy, very thin glass shards indicated by red arrows, loosely adhering to the grain surface. These tiny fragments are remnants of burst vesicle walls. B: glass fragment with smooth surface. C: glass fragment, with remnant of parallel pipe vesicles, notice the thin vesicle walls. D: Y-shaped glass fragment, remnant of a partially broken pumiceous pyroclast with elongated parallel bubbles. E: glass fragment with smooth surface. F: closeup of the glass fragment in E, showing in detail the smooth surface. G: portion of a vitric pyroclast with loose material on its surface (adhering dust), mostly tiny glass fragments, and a vesicle indicated by a red circle which contains small particles. H: closeup of the vesicle filling in G, showing an aggregate of carbon spherules of 100 nm to 200 nm corresponding to Black Carbon (BC) particles.

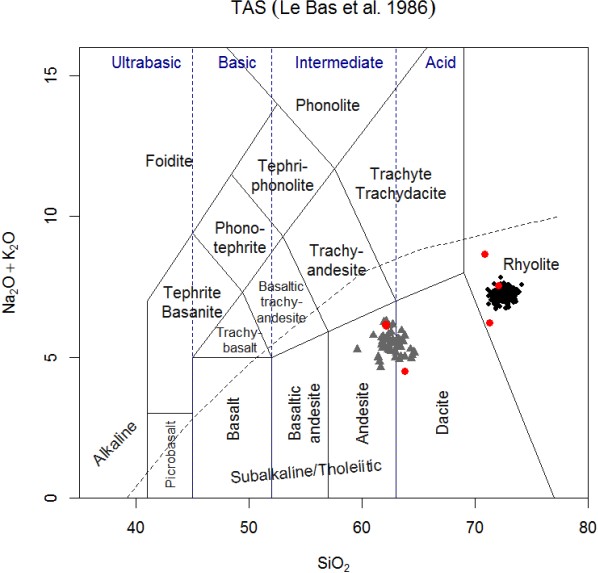

**Figure 8.** Classification diagram TAS (Le Bas et al., 1986). Major element compositions of glass shards from the AD 2015 Calbuco eruption acquired by electron microprobe analyses (LAMARX, Córdoba, Argentina) from samples collected during direct ashfall events in Junín de los Andes and Paso Cardenal Samoré, Argentina (Villarosa et al., 2016) and from the AD 2011 Puyehue-Cordón Caulle eruption acquired by electron microprobe (EMP) analysis, samples collected in San Carlos de Bariloche, Villa La Angostura and Paso Cardenal Samoré, Argentina (Alloway et al., 2015). Red circles: EDS analyses from PM samples from the studied area. Glass shards derived from Puyehue-Cordón Caulle (black circles) are rhyolitic in composition while glass from Calbuco eruption (grey triangles) is andesitic to dacitic in composition.

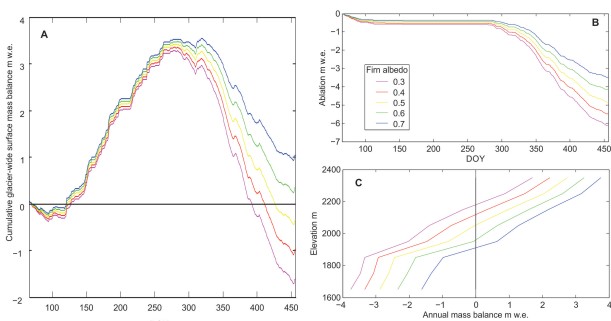

**Figure 9.** Sensitivity of Alerce glacier-wide surface mass balance to change in albedo of old snow or firn. A) cumulative glacier-wide surface mass balance, B) cumulative melt and C) mass balance gradient of Alerce glacier for the different albedo.