# Peer review of "Measurements and modeling of snow albedo at Alerce Glacier, Argentina: effects of volcanic ash, snow grain size and cloudiness"

_The Cryosphere, 2020_

## Referee Comment (RC1) · Anonymous Referee #1 · 2 Jun 2020

This paper gives a thorough account of April (2016 and 2017) field measurements conducted on the Alerce Glacier in the Northern Patagonian Andes. Combined with an updated Snow, Ice, and Aerosol Radiative (SNICAR) model that accounts for partly cloudy conditions, the measurements are used to estimate the glacier's April 2016 – April 2017 surface mass balance. Representing the first particulate matter concentration, albedo, and grain size measurements conducted on the Alerce Glacier, these results are a valuable contribution to the community and therefore warrant consideration for publication in *The Cryosphere*. Before acceptance, however, there are specific concerns, provided below, followed by a list of technical corrections that I recommend the authors consider in a minor revision.

Throughout the manuscript, the authors refer to an average snow grain radius value that they claim (in Sect. 2.2) to be precise. Average radii values were obtained using two methods: from visual inspection against a crystal grid, which is outdated, and from ImageJ software, which, to my knowledge, is not a standard method for obtaining snow grain radius. Although these methods provide one estimate of snow grain size (e.g., the length of maximum dimension), they will not yield a precise optically equivalent snow grain radius (nor specific surface area) that is the relevant quantity in two-stream snow radiative transfer algorithms like the SNICAR model. To reduce a potential source of error regarding the SNICAR modeling results, I suggest placing a greater emphasis on the other measured quantities used as inputs into the SNICAR model, especially the light absorbing particle (LAP) concentrations.

Regarding the use of terminology, a reader would benefit from a brief description of the distinction, if any, between LAPs and particulate matter (PM). The abstract begins by stating the relevance of light absorbing impurities in snow studies, however, the results and discussion most frequently refer to PM. Because "LAP" is a well known acronym, I suggest either maintaining the convention used in the literature, or defining PM while also elucidating the reason for the use of "PM" to describe these particular measurements.

Although I found Sect. 3 to be well written, I recommend the following technical corrections regarding mostly the other sections and figures:

1. *Abstract (lines 1–4):* Background could be refined, perhaps by moving one or two of the sentences into Sect. 1, to quickly introduce the present work.

2. *Abstract (line 6):* "during ablation" → "during the ablation"

3. *Abstract (line 9):* "from recent...eruption, with minor" → "from the recent...eruptions, with a minor"

4. *Abstract (lines 11–12):* "SNICAR model has been updated to model snow albedo taking into account" → "We updated the SNICAR model to account for"

5. *Abstract (line 14):* This part seems like an important component of this study, yet, it took me two or three times to understand the meaning of this sentence. Perhaps "which field measurements precision" can be rephrased to improve the readability.

6. *Abstract (line 17):* "m we" → "m snow water equivalent (SWE)"

7. *Sect. 1 (line 20):* I like this opening, but the first sentence needs to begin with "Since" or "Because."

8. *Sect. 1 (line 29):* It's probably better to use the term "light-absorbing particles (LAP)" (Skiles et al., 2018).

9. *Sect. 1 (line 38):* What is the distinction between LAP and atmospheric particulate matter?

10. *Sect. 1 (line 44):* "there has been found" → "it has been shown"

11. *Sect. 2 (lines 88-89):* "the hydrological year is defined from the 1-April to the 31-March of the next year. The accumulation season last from 1-April to 31-October and the ablation season from 31-October to the 31-March of the next year." → "the hydrological year begins on April 1st with the accumulation season. The accumulation season lasts until October 31st, which marks the beginning of the ablation season."

12. *Fig. 1 (caption):* It might be good practice to include the term "true color" in the description to indicate that the image is intended to reproduce a natural color rendition.

13. *Fig. 2 (caption):* It might be good practice to indicate that the grayscale used is logarithmic.

14. *Sect. 2.2 (line 125):* Please provide additional details of the "in-house developed supports" in order to improve the reproducibility of results.

15. *Sect. 2.2 (line 127):* "In the 2017 a" → "In the 2017 campaign, a"

16. *Sect. 2.2 (subsection headings):* Are these subsection headings supposed to be numbered (i.e., 2.2.1, 2.2.2, and 2.2.3)?

17. *Sect. 2.2 (line 153):* Equation (S1) has now been referred to twice. Should it be included in the main text?

18. *Sect. 2.3 (line 189):* Is $I_{glob} = I_{dir} + I_{diff}$, or something else? Perhaps this can be more clearly described in Sect. 2.2.

19. *Fig. 4*: If the vertical axis represents a normalized, dimensionless quantity, please indicate so. Otherwise, please provide the meaning of the vertical dimension. Also, the right-most part of the figure (horizontal axis) appears clipped.

20. *Fig. 5*: The box-and-whisker plot demonstrates the distribution of measurements nicely when $N > 2$. Does this mean that boxes represent standard deviations even when $N = 2$? If this is the case, perhaps a bar chart displaying the minimum and maximum values would be a more consistent portrayal of seasonal ranges, since standard deviations are better for estimating the variance of a distribution with a larger number of samples.

21. *Sect. 3.3 (line 368):* Although Cuffey and Paterson (2010) have written a standard text-book for glaciology, it would be nice to include a more accessible, primary reference that demonstrates this phenomenon.

22. *Sect. 3.3 (line 403):* "In the other hand" → "On the other hand"

23. *Sect. 3.4 (AAR):* Definition of accumulation area ratio? If it is the accumulation area to ablation area ratio, why are the values in m?

24. *Fig. 9:* The prefix "glacier-wide" is technically redundant, as"surface mass balance" is considered a surface area-integrated quantity. When referring to it as a local quantity, however, it can be stated as "specific surface mass balance." Also, the units on the axes labels should be in parentheses.

25. *Sect. 4 (line 510)* "observation and modeling activities to analysis" → "measurements and modeling to analyze"

26. *Sect. 4 (line 512):* No need for a paragraph break here.

27. *Sect. 4 (line 526):* "may difficult" → "may degrade"

28. *Sect. 4 (line 529):* Remove the comma.

29. *Sect. 4 (line 534):* "glacier-wide" → "surface" (see comment 24)

---

## Referee Comment (RC2) · Marius Schaefer (Referee) · 15 Jun 2020

**Review of "Measurements and modeling of snow albedo at Alerce Glacier,**
**Argentina: effects of volcanic ash, snow grain size and cloudiness" by Gelman et al**

**by Marius Schaefer (mschaefer@uach.cl)**

**General Comments:**

The manuscript presents punctual albedo measurements over snow surfaces on different parts of a small glacier in the Northern Patagonian Andes in two consecutive years, together with measurements of physical parameters which could mostly explain the measured albedo variations ( like grain size and form and particulate matter content). Then the authors try to reproduce the measured albedos, using a model, which is improved to account for partly cloudy conditions (which were present at least at one of the field days). In a last step the possible influence of the ash content, caused by eruptions of nearby volcanoes, on the total glacier surface mass balance is estimated using a simplified energy balance/ mass balance model.

To my point of view the study is original and novel and fits well into the scope of the journal. I think that the significance of the study could be significantly increased by adding some additional data and analysis, which should not be too difficult to obtain and which would allow to better interpret the presented field data and model results:

1) Measured surface mass data at stakes: I think that the surface mass balance data measured at stakes were somehow used to interpret the sample obtained form the snow pits (section 3.1, Figure 2) but the detailed data are not indicated. Also in section 2.4 it is stated: "The model was calibrated by surface mass balance measurements performed on a seasonal to annual basis through the year 2016 over Alerce glacier". I would like to know more details about this calibration process. How well could the model reproduce the observed melt and accumulation of snow? Which alpha_firn values fitted best to the observations? The time series of measured surface mass balance could also be helpful for quantifying the impact of the volcanic eruptions on the glacier's surface mass balance.

2) I am surprised by the big influence  of alpha_firn on the modeled surface mass balance of the glacier. In a "normal" year I would expect to have no firn in the ablation area and the firn of the accumulation area being buried by snow most of the year. How did you initialize the model (regarding presence of snow, firn, ice). Was 2016 a typical year? Probably not since autumn 2016 was exceptionally dry in the region.  I would propose to run the model with a few years of "typical" meteorologic data (mean value of several years)  and standard firn albedo for model initialization and then start to study the influence of different firn albedos. I think it should be much lower on average.

3) Since the albedo measurements are very punctual in time and space, and, as your repeating in the text several times that particulate matter concentration is very variable in time and space, it would be great to get an idea about the significance of your punctual albedo measurements by analyzing for example optical reflectance in satellite images. Images obtained at dates near to your field campaigns could be used for calibration. By this means you could also easily go back until the 2011 Cordon Caulle eruption.  Would be great to see how the reflectance of the glacier changed from summer 2011 to 2012. Or from summer  2015 to 2016.

**Technical Comments:**

Your abstract is 350 words which is too long (instructions form the journal's web page copied below). Try to reduce! For example you have three introducing sentences. One should be enough!

**Research articles** report substantial and original scientific results within the journal's scope. Generally, these are expected to be within 12 journal pages, have appropriate figures and/or tables, a maximum of 80 references, and an abstract of 150–250 words.

**Detailed Comments:**

**Page2**

**Line 26:** Patagonian Andes or Wet Andes instead of Southern Andes ? ( to be more precise).
**Line 27:** you mean net shortwave ? Albedo is not influencing the oncoming shortwave radiation. I would say summer, since in spring glaciers are mostly snow covered and exhibit high albedos
**Line 29 – until Page3 Line72:** in this section you discuss the influence of light-absorbing impurities on snow albedo. You mention particulate matter, mineral dust,volcanic ash and black carbon). Are all particulate matter light-absorbing impurities? Are mineral dust,volcanic ash and black carbon both particulate matter and light-absorbing impurities? Perhaps order these definitions in an introducing sentence and avoid synonyms ( particulate matter = light-absorbing impurities?)
**Line 31:** produced → producing
**Line 32:** "the growth of snow grains is accelerated" explain when and why.
**Line 38:** "as well as several positive feedbacks" which one?
**Line 42:** do not understand the sentence. What is a particle metric distribution?
**Line 45:** explain broadband albedo
**Line 5o:** what is "online coupling"?

**Page3**

**Lines 67-68:** do not understand the sentence starting with "For example ..." Reformulate!

**Page4**
**Line94:** I think the mass balance model is not mentioned in Ruiz et al 2017

**Page5:**
**Line124:** … "with a" … → … with one ...
**Line126:** How much W/m² is 0.1mv?

**Page6**
**Line 166:** "High-resolution pictures" … Would be great if you could show them in the supplementary material

**Page7**
**Line173/174** "are decribed in detail in section 3.2" → (Section 3.2)
**Line 180:** for → of

**Page8**

**Line 221&228 I** could not open the links indicated for the weather stations! Please indicate distance from glacier and elevation for both stations!

**Page9**

**Line 251/252:** on the base of what is this interpretation?

**Page10**
**Line 262:** Abl2-2016 → Abl1-2016?
**Line 264:** "These sites ..." which one? Abl3 and Abl4 ? In Abl2 and Abl5 PM content also seems to be quite high!
**Line 268: "** firn layer from 2015 winter" – how do you know?
**Line 290/291:** "low seasonal humidity" – do you mean variations?

**Page11**
**Line 328:** "it was dated as winter snow from 2014" – how?

**Page12:**
**Line 349:** "a single measurement" - what does that mean? One voltage reading? How stable is the voltage in time?
**Line 259: SNOW RADIUS!!!**

**Table1:**
Why are there two values for the measured albedo in Abl4?
Why do you present the measured albedo in different lines? Should be always next to the modelled W.Aver?
**Last column:**
could you describe in the methods how you obtain these sensitivities? Are they really always symmetric? I do not understand the uncertainty associated to the concentration of BC? Why is is sometimes 100microgranms/kg and sometimes 20mg/kg.
These numbers have many zeros! Could you better indicate the percentual sensitivity and mark the most important contributor?

**Page14.**
**Line 399: non-additive → non-linear?**

**Page 15**

**Line414/415:** revise sentence starting with: "Volcanic ash ..."
**Line 419:** what is a thin layer? Give number!

**Page 16**
**Line442** Albedo and glacier mass balance **model:** up to now only modeled mass balance is analyzed
**Line443 "…** glacier wide **modeled** annual and winter **..."**

**Page 18**
**Line510:** delete "PM over"
**Line 519:** delete "major"
**Line 523/524:** please propose how to take account for that
**Page 19**
**Line525: "We found that rapid changes ..."** this is only a problem for your specific set-up. If you are able to measure upwelling and downwelling radiation simultaneously, this is not a problem.
**Line530:** "… suggesting strategies ..."  which strategies are you suggesting? Which were the most important uncertainty?
**Line534/535:** glacier-wide albedo change sensitivity : explain this sensitivities with words or indicate where it was defined.
**Line536:** how high concentration of volcanic ash do you need for this reduction in SMB?

**Figure 1:**
could you please show the outline of Alerce glacier in the map and contour of terrain elevation?  Would also be nice to have another more zoom-out map to better see the glaciolocial context of Alerce Glacier.

**Figure 2:**
what meaning has a white column color?
What do you think: why did you not find the dark layer at 45cm in Acc4 in Acc5?

**Figure 4:**
what are the units of the Y-Axis?
Diffuse radiation should be less intense than the direct one!

---

## Author Comment (AC1) · 24 Jun 2020

**Author's response to Anonymous Referee #1**

**Referee comment**

This paper gives a thorough account of April (2016 and 2017) field measurements conducted on the Alerce Glacier in the Northern Patagonian Andes. Combined with an updated Snow, Ice, and Aerosol Radiative (SNICAR) model that accounts for partly cloudy conditions, the measurements are used to estimate the glacier's April 2016 – April 2017 surface mass balance. Representing the first particulate matter concentration, albedo, and grain size measurements conducted on the Alerce Glacier, these results are a valuable contribution to the community and therefore warrant consideration for publication in *The Cryosphere*. Before acceptance, however, there are specific concerns, provided below, followed by a list of technical corrections that I recommend the authors consider in a minor revision.

*Author's response*

*We deeply appreciate the referee for the thorough and useful comments to improve the manuscript.*

**Referee comment**

Throughout the manuscript, the authors refer to an average snow grain radius value that they claim (in Sect. 2.2) to be precise. Average radii values were obtained using two methods: from visual inspection against a crystal grid, which is outdated, and from ImageJ software, which, to my knowledge, is not a standard method for obtaining snow grain radius. Although these methods provide one estimate of snow grain size (e.g., the length of maximum dimension), they will not yield a precise optically equivalent snow grain radius (nor specific surface area) that is the relevant quantity in two-stream snow radiative transfer algorithms like the SNICAR model. To reduce a potential source of error regarding the SNICAR modeling results, I suggest placing a greater emphasis on the other measured quantities used as inputs into the SNICAR model, especially the light absorbing particle (LAP) concentrations.

*Author's response*

*We appreciate the referee's comment. Regarding the method in the first field campaign, we do agree is outdated, but it was the only method available for the first, exploratory campaign. Regarding the improved method we used in the second field campaign, which averages the maximum and minimum axes of equivalent ellipses that fit the snow grains in the pictures, we believe that it gives a reasonable estimate of the particles dimensions. We want to clarify that we do not claim that it is "precise", but only "more precise" than the previous method. The main evidence in support of our grain size results is that the differences among measured albedo values for fine and coarse snow can be explained using these grain size values in SNICARv2.1 model.*

*Nevertheless, we do agree that the snow grain size measurement method could be further improved. Pirazzini et al. (2015, cited in the manuscript) also use 2D photos, but with a different metric. They suggest that their metric is a proxy for "half the width of the shortest particle dimension", which they claim is a better approximation of the optically equivalent snow grain radius. If that is the case, our results would overestimate the optically equivalent snow grain radius. Pirazzini et al. determined 11% uncertainty in the 2D photos metrics (due to the subjectivity of the software operators). Although we did not determine such kind of uncertainty in our measurements, we report the estimated effect of the dispersion of the grain size for each sample, through sensitivity studies on SNICARv2.1 model. Even though the dispersions are large (probably larger than the uncertainty of the method), the effect on the modeled albedo are lower than 4.5% (for clean snow) or lower than 0.8% (for dirty snow). We believe that this explains the fact that we can reproduce the measured albedo using the estimated grain size together with other snow properties (especially LAP content), even though our grain size estimate might not be as accurate as that obtained by other methods. Spectral albedo (not available in our field campaigns) would be a complementary approach to validate separately the effect of snow grain size and LAP content on our albedo results. For instances, Carmagnola et al. (2013, cited in the manuscript) measured snow SSA (indirectly,*

*through an IR optical method) independently of LAP content, which mostly affects UV-vis albedo (lines 33-35 of the manuscript).*

*We modified the manuscript to include the limitations of our snow grain size measurement method, and we also modified the discussion of the results to remark that snow grain size results might not be as accurate as that from other measurements.*

**Manuscript Changes**

**Sect. 2.2, lines 165-168**

In the 2017 campaign, a similar in-house developed grid was used (with two scales: 1 mm and 0.5 mm) in combination with a macro lens and a mobile phone digital camera. High-resolution pictures where analyzed later with ImageJ software (Schneider et al., 2012). Snow grains were manually fitted with ellipses; the metric choice was the average of the minor and major axes of the ellipse. The new equipment and methodology introduced in the 2017 campaign allows a more detailed description of the snow samples and a more precise average radius value.

**Sect. 3.3, lines 358-361**

Regarding snow grain sizes, it is relevant to notice the range of observed average radius. In fresh snow samples from the accumulation zone (sites Acc5-2017 and Acc6-2017) we found an average radius of $(151 \pm 41)$ µm, whereas in samples of older firn in the ablation zone (or sub-surface snow/firn in the accumulation zone) we measured values usually around $(1000 \pm 200)$ µm. Pirazzini et al. (2015) also used 2D photos, but with a different metric. They suggest that SSK (shortest skeleton branch) is a proxy for "half the width of the shortest particle dimension", which they claim is a better approximation of the optically equivalent snow grain radius. Our metric (see Sect. 2.2) would probably give higher results than SSK, and hence we might have overestimated the optically equivalent snow grain radius. Nevertheless, as we show below in this section, our grain size measurements seem to be good enough to reproduce the measured albedo for fine and coarse snow in SNICARv2.1 snow albedo model.

**Sect. 3.3, lines 403-406**

In the other hand, comparison between sites with low PM content shows that snow grain size has a remarkable effect, as previously reported (Wiscombe and Warren, 1980; Hadley and Kirchstetter, 2012). Fresh snow with small grain size presents $\alpha_{meas} \approx 0.8$ (sites Acc5-2017 and Acc6-2017), but snow with similar PM content that has aged a few days presents $\alpha_{meas} \approx 0.6$ (site Acc2-2016). Spectral albedo measurements (not available in our field campaigns) would allow to study separately the effect of grain size and LAP content (see for instances measurements of snow specific surface area, SSA, in Carmagnola et al., 2013), to confirm that our grain size measurements are a good estimate of the optically equivalent grain radius.

**Sect. 3.3, lines 411-414**

Concerning grain size uncertainty (the standard deviation of snow grain radii in each sample), it is clear that the impact on albedo is much larger when PM content is low (sites Acc2-2016, Acc5-2017 and Acc6-2017). For low PM content sites, the effect is comparable to experimental uncertainty, and is relevant both for sites with finer and coarser grain sizes snow. For sites with high content of PM the uncertainty of grain size do not have an appreciable effect. Pirazzini et al. (2015) determined 11% uncertainty in the grain size measurements from 2D photos (due to the subjectivity of the software operators). Although we did not determine such kind of uncertainty in our measurements, we suggest that the reported standard deviation (between 16% and 26% of the average value) is probably larger than the uncertainty of the method. The sensitivity studies showed that the effect on the modeled albedo is lower than 4.5% for clean snow and lower than 0.8% for dirty snow. We believe that this explains the fact that we can reproduce the measured albedo using the estimated grain size together with other snow properties (especially LAP content), even though our grain size estimate might not be as accurate as that obtained by other methods.

**Referee comment**

Regarding the use of terminology, a reader would benefit from a brief description of the distinction, if any, between LAPs and particulate matter (PM). The abstract begins by stating the relevance of light absorbing impurities in snow studies, however, the results and discussion most frequently refer to PM. Because "LAP" is a well known acronym, I suggest either maintaining the convention used in the literature, or defining PM while also elucidating the reason for the use of "PM" to describe these particular measurements.

**Author's response**

*We agree with the referee that we should stress the difference between both expressions. The gravimetric measurements presented on this manuscript must be attributed to PM deposited on the glacier, because we don't know precisely the fraction of LAP among total PM. Qualitative observations also reported here (field stratigraphies and microscopy observations) suggest that most of collected PM can be attributed to volcanic ash. Quantifying the fraction of ash in collected PM (and/or measure the contributions from 2011 Cordon-Caulle and 2015 Calbuco eruptions) was not amenable. Routine stereo microscope inspection, even at high magnification (up to 80x), did not*

*allow quantification; only estimates of percentage of dark components was possible due to the fine-grained components. Na₂O, K₂O and SiO₂ content (SEM-EDS) from individual particles helped to distinguish volcanic ash from both eruptions, as exemplified in the manuscript (Fig. 8). In addition, CaO and FeO contents also proved useful to distinguish Cordón Caulle volcanic ash from ash derived from the 2015 Calbuco eruption (not included in the manuscript), but measuring a representative number of particles through SEM is not feasible. Hence, at this moment we can only suggest that most of PM on these samples correspond to volcanic ash, and that is the reason why we used SNICAR's built-in volcanic ash optical properties without further tuning. We know this is not exact, since optical microscopy and SEM microscopy have shown evidence of a minor fraction of mineral dust and black carbon, but we believe our results show that this assumption is a good first order approach to understand snow albedo on the glacier surface. A follow up article will include further chemical characterization of these samples, which has been delayed due to several reasons. We modified the manuscript to introduce a clear distinction between PM and LAP (LAI in our previous version of the manuscript), and we checked that those terms were used consistently through the manuscript.*

***Manuscript Changes***

- *We replaced the acronym "LAI" by "LAP" throughout the text.*
- *We replaced the acronym "PM" by "LAP" in some paragraphs, to clarify that we refer to the effect of particles that absorb light:*

***Sect. 1, lines 73-82:***

[revised manuscript text omitted]

**Referee comment**

Although I found Sect. 3 to be well written, I recommend the following technical corrections regarding mostly the other sections and figures:

1. *Abstract (lines 1–4):* Background could be refined, perhaps by moving one or two of the sentences into Sect. 1, to quickly introduce the present work.

*Manuscript Changes*

*Abstract, lines 1-4:*

The effect of volcanic ash in snow albedo (and its impact in seasonal snow and glacier mass balance) has been much less studied than that of other light absorbing impurities such as carbonaceous particles and mineral dust.  We present here the first field measurements on Argentinian Andes, combined with albedo and mass balance modeling activities.

**Referee comment**

2. *Abstract (line 6):* "during ablation" → "during the ablation"

3. *Abstract (line 9):* "from recent...eruption, with minor" → "from the recent...eruptions, with a minor"

4. *Abstract (lines 11–12):* "SNICAR model has been updated to model snow albedo taking into account" → "We updated the SNICAR model to account for"

*Author's response*

*We thank the referee for the useful grammar/phrasing suggestions.*

*Manuscript Changes*

*We adopt all changes suggested by the referee.*

**Referee comment**

5. *Abstract (line 14):* This part seems like an important component of this study, yet, it took me two or three times to understand the meaning of this sentence. Perhaps "which field measurements precision" can be rephrased to improve the readability.

*Author's response*

*We agree with the referee regarding the readability of the sentence, we rephrased it.*

*Manuscript Changes*

*Abstract, line 14:*

We also ran sensitivity studies considering the uncertainty of the main measured parameters (impurities content and composition, snow grain size, layer thickness, etc)to detect the field measurements that should be improved to facilitate the validation of the snow albedo model.

**Referee comment**

6. *Abstract (line 17):* "m we" → "m snow water equivalent (SWE)"

*Author's response*

*We thank the referee's suggestion, but we believe that both abbreviations for the snow water equivalence are widely used. We do agree that it needs to be defined in the abstract, and also in the main text.*

*Manuscript Changes*

*Abstract, line 17*

1.25 meter water equivalent decrease

*Sect. 3.4, line 447*

−0.6 meter water equivalent per year (m w.e./yr )

***Referee comment***

7. *Sect. 1 (line 20):* I like this opening, but the first sentence needs to begin with "Since" or "Because."

***Author's response***

*We agree with the referee's suggestion.*

***Manuscript Changes***

*Abstract, line 20*

Since laciers are highly sensitive to climate fluctuations, their unprecedented retreating rates observed during the last decades represent one of the most unambiguous signals of climate change

***Referee comments***

8. *Sect. 1 (line 29):* It's probably better to use the term "light-absorbing particles (LAP)" (Skiles et al., 2018).

9. *Sect. 1 (line 38):* What is the distinction between LAP and atmospheric particulate matter?

***Author's response***

*See response and changes in the discussion regarding PM and LAP above in this file.*

***Referee comments***

10. *Sect. 1 (line 44):* "there has been found" → "it has been shown"

11. *Sect. 2 (lines 88-89):* "the hydrological year is defined from the 1-April to the 31-March of the next year. The accumulation season last from 1-April to 31-October and the ablation season from 31-October to the 31-March of the next year." → "the hydrological year begins on April 1st with the accumulation season. The accumulation season lasts until October 31st, which marks the beginning of the ablation season."

***Author's response***

*We thank the referee for the useful grammar/phrasing suggestions.*

***Manuscript Changes***

*We adopt all changes suggested by the referee.*

***Referee comments***

12. *Fig. 1 (caption):* It might be good practice to include the term "true color" in the description to indicate that the image is intended to reproduce a natural color rendition.

13. *Fig. 2 (caption):* It might be good practice to indicate that the grayscale used is logarithmic.

***Author's response***

*We thank the referee for the useful suggestions.*

***Manuscript Changes***

*Fig. 1 caption:*

**Figure 1.** True color satellite image of Alerce Glacier. Sampling points are represented as blue markers (2017 campaign) and red markers (2016 campaign). Green marker represents Otto Meiling mountain hut. Copyright: © Google Earth, 2020, CNES/Airbus

*Fig. 2 caption:*

**Figure 2.** PM concentration (grayscale) as a function of pit depth for different sampling sites. Notice that the grayscale is logarithmic. Top panel: accumulation zone. Bottom panel: ablation zone. $\alpha$ symbol is used to highlight sites with concurrent albedo measurements. In sample *Abl2-2016*, the top rectangle corresponds to the average PM content of the first two layers (fresh snow and end-of-summer dark layer).

***Referee comment***

14. *Sect. 2.2 (line 125):* Please provide additional details of the "in-house developed supports" in order to improve the reproducibility of results.

***Author's response***

*We added a Figure at the Supplement to provide additional construction details on the supports.*

***Manuscript Changes***

***Sect 2.2, line 150***

Additional details on the supports are given in Fig. S1 in Supplement.

*Supplement, Fig. S1*

[Figure]

*Figure S1. Details of the pyranometer supports in Fig. 3 of the main text. (a) Side view and (b) top view of the support used in 2016 field campaign. (c) Side view of the support used in 2017 field campaign.*

**Referee comments**
15. *Sect. 2.2 (line 127):* "In the 2017 a" → "In the 2017 campaign, a"
16. *Sect. 2.2 (subsection headings):* Are these subsection headings supposed to be numbered (i.e., 2.2.1, 2.2.2, and 2.2.3)?
**Author's response**
*We agree with the referee's suggestions.*
**Manuscript Changes**
*We adopt the referee's suggestions.*

**Referee comment**
17. *Sect. 2.2 (line 153):* Equation (S1) has now been referred to twice. Should it be included in the main text?
**Author's response**
*We believe that the details of the albedo measurements corrections (including Eq. S1) are not needed in the main text.*
**Manuscript Changes**
*We did not find the need to introduce any changes.*

**Referee comment**
18. *Sect. 2.3 (line 189):* Is $I_{glob} = I_{dir} + I_{diff}$, or something else? Perhaps this can be more clearly described in Sect. 2.2.
**Author's response**
*Yes, $I_{glob} = I_{dir} + I_{diff}$, as usually defined, but we accept the suggestion to remark that in the manuscript.*
**Manuscript Changes**
**Sect. 2.3 (line 189)**
*$I_{dir}$, $I_{diff}$ , and $I_{glob}$ are clear-sky  direct, diffuse and global solar irradiance (where $I_{glob} = I_{dir} + I_{diff}$,), as calculated from SMARTS model.*

**Referee comment**
19. *Fig. 4*: If the vertical axis represents a normalized, dimensionless quantity, please indicate so. Otherwise, please provide the meaning of the vertical dimension. Also, the right-most part of the figure (horizontal axis) appears clipped.
**Author's response**
*We thank the referee for the comment. The plotted distributions are indeed normalized, we modified the caption to make that clear. We corrected the clipping of the image to show the last horizontal axis label correctly.*

*Manuscript Changes*
- *We corrected the clipping of the image.*
- *Fig. 4 caption:*

**Figure 4.** Different normalized spectral distributions of sun radiation for SNICAR snow albedo model. SNICARv2 included two spectra for mid-latitude locations: one for overcast conditions (light green line), and one for clear sky conditions (dark green line). SMARTS diffuse (light red line) and direct (dark red line) clear sky spectra for one of our sampling sites are represented for comparison. Dotted lines represent spectra for partly cloudy conditions (SNICARv2.1).

*Referee comment*

20. *Fig. 5*: The box-and-whisker plot demonstrates the distribution of measurements nicely when $N > 2$. Does this mean that boxes represent standard deviations even when $N = 2$? If this is the case, perhaps a bar chart displaying the minimum and maximum values would be a more consistent portrayal of seasonal ranges, since standard deviations are better for estimating the variance of a distribution with a larger number of samples.

*Author's response*

*We thank the referee for drawing the attention on this plot. We agree that standard deviation is more relevant for $N >> 2$, but even so we believe in this case box-and-whiskers plot gives more information than a bar chart. For cases with $N > 2$ (four of the plotted seasonal ranges) the plot allows showing the range where most data fall, together with the extreme values (which in some cases are far away form standard deviation, e.g. Acc. season 2015). For $N=2$ (the three remaining seasonal ranges), the standard deviation is equal to half the separation between the minimum and maximum value, and hence the plot shows the minimum and maximum values. We modified the figure's caption to stress this fact, so the plot can be easily interpreted.*

*Manuscript Changes*
- *Fig. 5 caption:*

**Figure 5.** Seasonal range of PM concentration found on snow/firn samples. For accumulation season, the values represent the mean PM concentration in thick, low PM layers of snow/firn. For ablation season the values represents the surface PM concentration at the end of the season. The box encompasses one standard deviation of data, and whiskers represent minimum and maximum values (when $N > 2$). Notice that for seasonal layers with only two measurements, the box represents those two values (coincident with the definition of standard deviation for $N = 2$). The plot includes data from both field campaigns, and excludes ablation ice samples, which cannot be assigned to a specific year/season. Fresh snow represent snow fallen a few days before field campaigns of 2016 or 2017.

*Referee comment*

21. *Sect. 3.3 (line 368):* Although Cuffey and Paterson (2010) have written a standard textbook for glaciology, it would be nice to include a more accessible, primary reference that demonstrates this phenomenon.

*Author's response*

*We took the referee suggestion and found a different reference regarding the phenomenon.*

*Manuscript Changes*

Flanner, M. G., and Zender, C. S.: Linking snowpack microphysics and albedo evolution, Journal of Geophysical Research, 111, D12208. https://doi.org/10.1029/2005JD006834, 2006

*Referee comment*

22. *Sect. 3.3 (line 403):* "In the other hand" → "On the other hand"

*Author's response*

*We thank the referee for the useful grammar suggestion.*

*Manuscript Changes*

*We adopt the change suggested by the referee.*

*Referee comment*

23. *Sect. 3.4 (AAR):* Definition of accumulation area ratio? If it is the accumulation area to ablation area ratio, why are the values in m?

*Author's response*

*We thank the referee for noticing the mistake in the units of the Accumulation Area Ratio (the ratio of the glacier's accumulation area to its total area). The numbers are correct but they are a percentage.*

*Manuscript Changes*

*Table 2 header*

| $\alpha_{ice}$ | $\alpha_{firn}$ | $\alpha_{max}$ | Wint. MB (m w.e.) | Annu. MB (m w.e.) | ELA (m) | AAR (%) |
|---|---|---|---|---|---|---|

*Referee comments*

24. *Fig. 9:* The prefix "glacier-wide" is technically redundant, as "surface mass balance" is considered a surface area-integrated quantity. When referring to it as a local quantity, however, it can be stated as "specific surface mass balance." Also, the units on the axes labels should be in parentheses.

25. *Sect. 4 (line 510)* "observation and modeling activities to analysis" → "measurements and modeling to analyze"

26. *Sect. 4 (line 512):* No need for a paragraph break here.

27. *Sect. 4 (line 526):* "may difficult" → "may degrade"

28. *Sect. 4 (line 529):* Remove the comma.

29. *Sect. 4 (line 534):* "glacier-wide" → "surface" (see comment 24)

*Author's response*

*We thank the referee for the useful grammar/phrasing suggestions.*

*Manuscript Changes*

*We adopt the changes suggested by the referee.*

---

## Author Comment (AC2) · 23 Jul 2020

**Author's response to Marius Schaefer's review**

**Referee comment**

**General Comments:**

The manuscript presents punctual albedo measurements over snow surfaces on different parts of a small glacier in the Northern Patagonian Andes in two consecutive years, together with measurements of physical parameters which could mostly explain the measured albedo variations (like grain size and form and particulate matter content). Then the authors try to reproduce the measured albedos, using a model, which is improved to account for partly cloudy conditions (which were present at least at one of the field days). In a last step the possible influence of the ash content, caused by eruptions of nearby volcanoes, on the total glacier surface mass balance is estimated using a simplified energy balance/mass balance model.

To my point of view the study is original and novel and fits well into the scope of the journal. I think that the significance of the study could be significantly increased by adding some additional data and analysis, which should not be too difficult to obtain and which would allow to better interpret the presented field data and model results:

**Author's response**

We appreciate the referee's thorough and useful comments to improve the manuscript. Although the suggested additions would increase the significance of the article, some of them are outside the focus of this manuscript. The manuscript already deals with field measurements and models. Including the use of remote sensing data would make it excessively long. We discuss the suggested additions point by point next.

**Referee** comment**

1) Measured surface mass data at stakes: I think that the surface mass balance data measured at stakes were somehow used to interpret the sample obtained form the snow pits (section 3.1, Figure 2) but the detailed data are not indicated. Also in section 2.4 it is stated: "The model was calibrated by surface mass balance measurements performed on a seasonal to annual basis through the year 2016 over Alerce glacier". I would like to know more details about this calibration process. How well could the model reproduce the observed melt and accumulation of snow? Which alpha\_firn values fitted best to the observations? The time series of measured surface mass balance could also be helpful for quantifying the impact of the volcanic eruptions on the glacier's surface mass balance.

**Author's response**

A comprehensive evaluation of the mass balance of Alerce glacier is beyond the scope of this work and it is core of an ongoing manuscript by one of the members of the author team (Lucas Ruiz). We included in Fig. 1 the location of ablation stakes, and in the Supplement (Fig. S4) the location of snow thickness measurements. Detail regarding the process of calibration of the surface mass balance model (SMB model) was added in Sect. 2.4 together with two new figures in the Supplement (Fig S5 and S6) which shows the agreement between modeled and measurements used to calibrate the SMB model and the fitting of the model for two of the ablation stakes close to the albedo sampling locations.

For the hydrological years 2015 and 2016 (during and after the Calbuco eruption) best agreement between measurements and model was achieved using minimum snow albedo values of 0.42-0.38. The range express the difficulty to achieve a straightforward calibration of the different parameters used in enhanced degree-day models. Some parameters counteract each other and minimum RMSEs could be achieved with a variety of parameter combination. Thus, it is also necessary considering surface characteristics at the stakes locations and their distribution across the glaciers, like transient snow lines or extra mass balance measurements through the year.

**Manuscript Changes**

**Lines 228-234:**

After calibration of the model,  $c_0 = -50 \text{ W m}^{-2}$  and  $c_1 = 12 \text{ W m}^{-2} \cdot \mathbb{C}^{-1}$ . Potential direct solar radiation for all grid cells and days was calculated following Hock (1999). The local surface albedo  $\alpha(x, y, t)$  was taken to be constant for bare-ice

surfaces ( $\alpha_{ice} = 0.34$ ), using most commonly applied literature value (Oerlemans and Knap, 1998; Cuffey and Paterson, 2010), for snow surfaces,  $\alpha_{snow}$  was calculated based on the snow aging function proposed by Oerlemans and Knap (1998) with a maximum snow albedo ( $\alpha_{max}$ ) of 0.8 and a variable-minimum snow albedo ( $\alpha_{min}$ ) adjusted during the calibration procedure. ( $\alpha_{firn}$ , table 2).

The model was calibrated in two steps using surface mass balance measurements of year 2016 in Alerce glacier (Supplement, Fig. S4). First, the model is run over the winter period with an initial set of constants ( $c_0$  and  $c_1$ ) and a guess for the precipitation correction factor  $C_{pre}$ . As melt is of minor importance in winter, this run is used to calibrate  $C_{pre}$ , that scales  $D_s$  for every snow fall event. After a good agreement of measured and calculated winter accumulation is obtained, the model is run over the entire year and the remaining constants are calibrated so that the root-mean-square error between modelled and observed point annual balances is minimized and the average misfit is close to zero (Supplement, Fig. S5 and S6). A random set of snow accumulation and ablation stakes measurements performed through the year and not used to calibrate the model are left apart to validate the results of the surface mass balance model.

We studied Gglacier-wide mass balance changes for between different values of  $\alpha_{min afree}$  (Table 2), which are indicative of the sensitivity of glacier mass balance to a change in albedo that might occur in response to the darkening of the glacier surface.

Supplement, Fig. S4, S5 and S6 (see at the end of this file).

**Referee comment**

2) I am surprised by the big influence of alpha\_firn on the modeled surface mass balance of the glacier. In a "normal" year I would expect to have no firn in the ablation area and the firn of the accumulation area being buried by snow most of the year. How did you initialize the model (regarding presence of snow, firn, ice). Was 2016 a typical year? Probably not since autumn 2016 was exceptionally dry in the region. I would propose to run the model with a few years of "typical" meteorologic data (mean value of several years) and standard firn albedo for model initialization and then start to study the influence of different firn albedos. I think it should be much lower on average.

**Author's response**

We acknowledge that the use of  $\alpha_{firn}$  as a synonymous of minimum snow albedo was not a good choice and give place to confusions. As we stated in Section 2.4,  $\alpha_{firn}$  is the minimum albedo that snow could reach using the snow aging function of Oerlemans and Knap (1998). We replaced  $\alpha_{firn}$ for  $\alpha_{min}$  to avoid any confusion. We agree that if we had only changed the albedo of the firn (the snow accumulated after more than year, for instances), the effect on the surface mass balance would have been much lower.

The model is initiated with a guess snow and firn lines and run for a few days before the evaluated period, which is observational period. to stabilize the surface mass balance to the input data. We have tested different initiation scenarios, to check the sensitivity of the model to initial conditions, and under realistic scenarios, the sensitivity is rather low.

Finally, we agree with the reviewer, 2016 was the driest year since we start the monitoring of the Alerce glacier in 2013.

**Manuscript Changes**

We replaced  $\alpha_{firn}$  for  $\alpha_{min}$  throughout the manuscript.

**Referee** comment**

3) Since the albedo measurements are very punctual in time and space, and, as your repeating in the text several times that particulate matter concentration is very variable in time and space, it would be great to get an idea about the significance of your punctual albedo measurements by analyzing for example optical reflectance in satellite images. Images obtained at dates near to your field campaigns could be used for calibration. By this means you could also easily go back until the 2011 Cordon Caulle eruption. Would be great to see how the reflectance of the glacier changed from summer 2011 to 2012. Or from summer 2015 to 2016.

**Author's response**

Satellite observations are relevant, and we have already look at MODIS products and other remote data for a following article. Although satellite snow reflectance data could be used to evaluate the significance of our punctual surface measurements (albedo measurements, particles content and

snow grain size), Landsat and Sentinel images close to the timing of our measurements are totally or partially cloud covered for Monte Tronador. As we stated in the manuscript cloudiness conditions were challenging and we needed to update SNICAR model to deal with it. Regarding the use of MODIS, although the time resolution allows us to have more images without excessive cloud cover, it spatial resolutions challenges the evaluation against punctual surface measurements. Nevertheless, our preliminary evaluation of MODIS albedo time series of Monte Tronador, shown a decrease in late summer albedo after the Cordon Caulle and Calbuco eruption, with a minimum during the late summer of 2017 (both a combination of the ashes and less snow fallen over the glacier). Nevertheless, as we mention above, these additional analysis would require a considerable amount of space, hence we decided to keep them for another manuscript where we can deal properly with it.

**Referee comment**

**Technical Comments:**

Your abstract is 350 words which is too long (instructions form the journal's web page copied below). Try to reduce! For example you have three introducing sentences. One should be enough! **Research articles** report substantial and original scientific results within the journal's scope. Generally, these are expected to be within 12 journal pages, have appropriate figures and/or tables, a maximum of 80 references, and an abstract of 150–250 words.

**Author's response**

We thank the referee for the suggestion. We have already reduced the length of the abstract following a suggestion of the Anonymous Referee #1. We present here a further effort of making the abstract more concise.

**Manuscript Changes**

**Abstract**

The impact of volcanic ash on seasonal snow and glacier mass balance has been much less studied than that of carbonaceous particles and mineral dust. We present here the first field measurements on Argentinian Andes, combined with snow albedo and glacier mass balance modeling. Measured impurities content (1.1 mg kg-1 to 30 000 mg kg-1) varied abruptly in snow pits and snow/firn cores, due to high surface enrichment during the ablation season and possibly local/regional wind driven resuspension and redeposition of dust and volcanic ash. In addition, we observed a high spatial heterogeneity, due to glacier topography and prevailing wind direction. Microscopical characterization showed that the major component was ash from recent Calbuco (2015) and Cordón Caulle (2011) volcanic eruption, with a minor presence of mineral dust and Black Carbon. We also found a wide range of measured snow albedo (0.26 to 0.81), which reflected mainly the impurities content and the snow/firn grain size (due to aging). We updated the SNICAR snow albedo model to account for the effect of cloudiness on incident radiation spectra, improving the match of modeled and measured values. We also ran sensitivity studies considering the uncertainty of the main measured parameters (impurities content and composition, snow grain size, layer thickness, etc) to detect the field measurements that should be improved to facilitate the validation of the snow albedo model. Finally, we studied the impact of these albedo reductions in Alerce glacier using a spatially distributed surface mass-balance model. We found a large impact of albedo changes in glacier mass balance, and we estimated that the effect of observed ash concentrations can be as high as a 1.25 meter water equivalent decrease in the glacier-wide annual mass balance (due to a 34 % of increase in the melt during the ablation season).

**Referee comment**

Detailed Comments: Page2

**Line 26:** Patagonian Andes or Wet Andes instead of Southern Andes ? ( to be more precise). *Author's response*

We agree with the referee that the suggested terms are more precise, we rephrased. Manuscript Changes

**Lines 25-26:**

Along the SouthernWet Andes (below 35° S latitude), melt is driven by shortwave radiation and sensible turbulent flux (Schaefer et al., 2019).

**Referee comment**

Line 27: you mean net shortwave ? Albedo is not influencing the oncoming shortwave radiation. I

would say summer, since in spring glaciers are mostly snow covered and exhibit high albedos *Author's response*

We thank the referee for the comments. Regarding the first comment, we rephrased the sentence in order to make sure the meaning of the sentence is transparent. Regarding the second comment, we agree that the exposure of low albedo layers is much more significant in summer. Manuscript Changes

Lines 25-29:

The effect of incoming shortwave radiation absorption increases significantly is enhanced during spring and summer, due to the exposure of low albedo areas in their ablation zones, which causes strong, positive feedback that enhances surface melt significantly and shapes the spatial ablation pattern (Brock et al., 2000).

**Referee comment**

**Line 29 – until Page3 Line72:** in this section you discuss the influence of light-absorbing impurities on snow albedo. You mention particulate matter, mineral dust, volcanic ash and black carbon). Are all particulate matter light-absorbing impurities? Are mineral dust, volcanic ash and black carbon both particulate matter and light-absorbing impurities? Perhaps order these definitions in an introducing sentence and avoid synonyms ( particulate matter = light-absorbing impurities?) *Author's response*

We agree with the referee that the original manuscript was not clear enough regarding these definitions, as was also pointed out by Anonymous Referee #1.

**Manuscript Changes**

We introduced several changes that are detailed in the Author's Response to Anonymous Referee #1, pages 2-4.

**Referee comment**

**Line 31:** produced  $\rightarrow$  producing *Author's response* We thank the referee for the useful phrasina su

We thank the referee for the useful phrasing suggestion. *Manuscript Changes*

**Lines 29-31*:**

Furthermore, deposition of light-absorbing impurities (LAP; mineral dust, volcanic ash, and black carbon) have a fundamental impact on the melting of glacier and snow-covered areas by increasing the absorption of solar radiation and producinges a regional land-atmosphere feedback

**Referee** comment**

Line 32: "the growth of snow grains is accelerated" explain when and why.

**Author's response**

We accept the referee's suggestion to further explain this effect. We rephrased two sentences to better explain the direct and indirect effects of LAP on snow.

**Manuscript Changes**

**Lines 29-33*:**

Furthermore, deposition of light-absorbing impurities (LAP; mineral dust, volcanic ash, and black carbon) have a fundamental impact on the melting of glacier and snow-covered areas by increasing the absorption of solarradiation and produces a regional land-atmosphere feedback (Warren and Wiscombe, 1980; Bond et al., 2013; Molina et al., 2015). LAP decrease snow albedo, increasing solar radiation absorption and thus producing a direct effect on snowmelting. But, in addition, the snowpack temperature increase due to the direct effect\_Along with the enhanced melting due to the darkening of the snow or ice surface; accelerates the growth of snow grains is accelerated, which furtherreinforces snowmelt rates due toproduces a further albedo decrease (and thus an additional, indirect impact on snowmelting) (Bond et al., 2013; Flanner et al., 2007).

**Referee comment**

**Line 38:** "as well as several positive feedbacks" which one? *Author's response*

The thorough review by Bond et al. (2013) describes in detail the multiple rapid changes in snow due to LAP deposition (see Fig. 29 of the reference). We added in the text two of the more important feedback processes and refer the reader to the reference.

**Manuscript Changes**

**Lines 37-40*:**

Different snow albedo models have been developed to include the direct effect of Black Carbon (BC) and other LAP atmospheric particulate matter (PM) as well as several positive feedbacks (Flanner et al., 2007; Koch et al., 2009; Krinner et al., 2006), such as the increase in surface concentration of impurities due to enhanced snow melting, or the albedo reduction due to snow grains growth by accelerated snow aging (Bond et al., 2013). More recently, models have included the effects of non-spherical snow grains (Libois et al., 2013; He et al., 2017), and external/internal mixing of impurities with snow grains (He et al., 2018).

**Referee comment**

**Line 42: do not understand the sentence. What is a particle metric distribution?**

**Author's response**

We agree with the referee that sentence needs rephrasing. We hope that this new phrasing gives a better, concise description of the main results of the references, and help the reader to find further details therein.

**Manuscript Changes**

**Lines 42-43:**

More than just one particle metric distribution is necessary to reproduce the spectral snow albedo at all optical wavelengths, especially wWhen the snow has been undergoing heavy metamorphosis processes, a single snow grain size distribution is not enough to reproduce the snow spectral albedo, due to the fact that the largest particles and the thinnest protrusions of the irregular crystals have contributions to the snow reflectance that depend on the wavelength (Carmagnola et al., 2013; Pirazzini et al., 2015)

**Referee comment**

**Line 45: explain broadband albedo**

**Author's response**

We thank the referee's question. We rephrased the sentence to explain more clearly the results in Zhang et al., 2018.

**Manuscript Changes**

Notably, there has been found that taking into account the amount of LAPI in the snow reduces the difference between simulated and measured broadband albedos, specially in the visible range (Zhang et al., 2018).

**Referee comment**

**Line 50: what is "online coupling"?**

**Author's response**

We agree with the referee that the phrase might not be clear for some readers. We use the term "online coupling" to imply that the two models (snow albedo model and atmospheric chemistry model) are run simultaneously and allowing two-way feedback. Other studies use offline coupling, where one of the models (usually, the atmospheric chemistry model) is run first, and the results are used as input for the other model (snowpack model or glacier mass balance model).

**Manuscript Changes**

**Lines* 50-52:**

"Online" coupling of snow albedo models in global or regional atmospheric chemistry models (where both models are run simultaneously allowing two-way feedback) have been applied to study snow and glaciers interaction with the climate around the globe (Hansen et al., 2005; Flanner, 2013; Ménégoz et al., 2014).

**Referee comment**

Page3 Lines 67-68: do not understand the sentence starting with "For example ..." Reformulate! *Author's response*

We rephrased the sentence. Manuscript Changes Lines 67-68: For example, the albedo reduction for spherical snow grains radii of 100 µm due to BC alone in the north was estimated to beis only about 43 % of that for all light-absorbing impurities (assuming spherical 100 µm radii snow grains).

**Referee comment* Page4**

Line 94: I think the mass balance model is not mentioned in Ruiz et al 2017

**Author's response**

We thank the referee for noticing the mistake, we corrected the position of the references regarding the Alerce glacier monitoring and we added a new one regarding the mass balance model. *Manuscript Changes*

**Lines 91-94:**

Since 2013 it has been the focus of a glacier mass balance monitoring program by the IANIGLA (Instituto Argentino de Nivología, Glaciología y Ciencias Ambientales; Ruiz et al., 2015, 2017). Seasonal mass balance has been studied every year using the traditional glaciological method of stakes, and snow pits. An enhanced temperature index mass balance model has been developed (Huss et al., 2008; Huss, 2010)(<del>Ruiz et al., 2015, 2017</del>) to study the surface mass balance of the glacier.

**Referee comment**

**Page 5: Line 124:** ... "with a"  $\dots \rightarrow \dots$  with one ... *Author's response We thank the referee for the useful suggestion. Manuscript Changes*

Upwelling (reflected) and downwelling (direct + diffuse) radiation were measured with aone CM5 Kipp & Zonen pyranometer (wavelength range 0.3  $\mu$ m to 2.8  $\mu$ m), using two different in-house developed supports in 2016 and 2017 campaigns, logged with a handheld voltmeter.

**Referee comment**

Line 126: How much W/m2 is 0.1mv?

**Author's response**

For reference, 0.1 mV represents approximately 9.5  $W/m^2$  for our pyranometer. We did not find relevant to include the conversion factor in the article since we do not report solar irradiances, but only measured albedos (the conversion factor is not relevant for the radiation ratios).

**Referee comment**

**Line 166:** "High-resolution pictures" ... Would be great if you could show them in the supplementary material

Author's response

We added a figure in the Supplement (Fig. S3).

**Manuscript Changes**

Lines 166-167:

High-resolution pictures (Fig. S3, Supplement) where analyzed later with ImageJ software (Schneider et al., 2012). *Supplement, Fig. S3 (see at the end of this file)*

**Referee comment Page 7 Line 173/174** "are decribed in detail in section 3.2"  $\rightarrow$  (Section 3.2) **Line 180:** for  $\rightarrow$  of **Author's response** We thank the referee for the useful suggestion. **Manuscript Changes** We adopted the suggested changes.

**Referee comment**

**Page 8**

**Line 221&228 I** could not open the links indicated for the weather stations! Please indicate distance from glacier and elevation for both stations!

**Author's response**

We thank the referee for noticing the mistake, we corrected the links and added the altitude of the stations.

**Manuscript Changes**

*Line 221:*

*P(t)* was the daily precipitation at Tepual weather station (90 m altitude, ID = 857990; http://www7.nede.noaa.gov/https://www.ncei.noaa.gov/access/search/data-search/global-summary-of-the-day) *Lines 277-228:*

T(t) was taken from the air surface temperature at Bariloche airport weather station (846 m altitude, ID = 877650; http://www7.ncde.noaa.gov/https://www.ncei.noaa.gov/access/search/data-search/global-summary-of-the-day).

**Referee comment**

Line 251/252: on the base of what is this interpretation?

**Author's response**

The interpretation of the snow/firn layers is based on the observed stratigraphy of the snow column. Snow pits walls and cores were described following common glaciological practices, in terms of layering, grain size and shape, content of PM, density and hardness. Dating of layers or attribution of time windows for each layer was based on the stratigraphic relations between layers and its characteristics. In this case, the layer (242 cm to247 cm) had a high PM concentration, was below a thick, relative low PM content, soft snow layer (interpreted as the snow accumulated during the accumulation season of the hydrological year 2015-2016) and above a harder, coarser grained firn layer (interpreted as the snow of the accumulation season of 2014-2015).

**Manuscript Changes**

The deepest (242 cm to 247 cm deep) thin, high PM concentration layer (( $1970 \pm 200$ ) mg kg-1) was interpreted as the surface at end of the ablation season of the hydrological year 2014-15, based on the abrupt change of the density, hardness and grain size of the snow above this layer and the firn found below.

Referee comment Page 10 Line 262: Abl2-2016  $\rightarrow$  Abl1-2016? Author's response We thank the referee for noticing the mistake Manuscript Changes We corrected the mistake in the sampling site name.

**Referee comment**

Line 264: "These sites ..." which one? Abl3 and Abl4 ? In Abl2 and Abl5 PM content also seems to be quite high!

**Author's response**

The sentence refers to the sites Abl3-2017 and Abl4-2017, mentioned in the previous sentence, but we changed the sentence to avoid any misunderstanding. The PM content on the surface layer of those sites,  $(30000 \pm 5000) \text{ mg kg}^{-1}$  and  $(12000 \pm 2000) \text{ mg kg}^{-1}$  respectively, is much higher than that of any other site, due to the reasons explained in the manuscript and in the new section S2 of the Supplement (see response to the next comment). Sites Abl2-2016 and Abl5-2017 had a surface layer of recent snow. Below the surface layer, the PM content of the summer surface layer of site Abl2-2016 was (4400 ± 800) mg kg-1. Site Abl5-2017 presented glacier ice below the surface layer (which was not sampled).

Manuscript Changes Lines 264-265: These sSites Abl3-2017 and Abl4-2017 had a negative net balance during hydrological year 2016-17, consequently the surface layer presented the highest PM content observed in both campaigns

**Referee** comment**

**Line 268:** " firn layer from 2015 winter" – how do you know? *Author's response**

The layers from sites Abl3-2017 and Abl4-2017 (placed close to each other in the same accumulation pocket, see new Fig. S2 at the end of this file) were identified based on stratigraphic relationships. The dark surface at site Abl4-2017 was the topmost layer of the pocket, but based on the grain size ( $738 \pm 167 \mu m$ ), density and hardness, we interpreted that all accumulation from 2016 winter had melted. The high PM concentration ( $12000 \pm 2000$ ) mg kg-1 was also consistent with the surface enrichment due to melting of snow deposited in more than one hydrological year. The firn below this layer was then identified as the accumulation layer from 2015 winter. In site Abl3-2017, towards the border of the accumulation pocket, the topmost layers described for site Abl4-2017 had also disappeared. Hence, we interpreted that all accumulation from 2015 winter had also melted in this site, and this darkest, surface layer contained most of PM deposited in 2016 and 2015. The firn layer below was interpreted as the accumulation layer from 2014 winter.

**Manuscript Changes**

**Lines 267-270:**

In-situ stratigraphy revealed that in *Abl4-2017* site, the high concentration layer was on top of relatively low concentration, firn layer from 2015 winter, which means that, during the 2016-2017 ablation season, all the snow accumulated during 2016 winter was melted. Site *Abl3-2017* presented an even lower net balance, revealing older firm (winter 2014) below the surface high concentration layer. See Sect. S2 in Supplement for additional details on the attribution of layers in sites Abl3-2017 and Abl4-2017.

**Supplement, line 34:**

S2 Dating of snow/firn layers

Most snow/firn layers sampled during both field campaigns were easily dated, considering that the topmost layer contains the most recent snow and attributing layers below based on PM content, density, hardness and grain size. Topmost layers were identified as:

(1) fresh snow from a recent deposition events, on the accumulation zone, (sites Acc1-2016, Acc2-2016, Acc4-2017, Acc5-2017, Acc6-2017 and Acc7-2017, Fig S2(a)), on an accumulation pocket (site Ab11-2016), or on top of ablation ice (sites Ab12-2016 and Ab15-2017),

(2) end-of-ablation season surface, with high enrichment of PM content (Acc3-2016, Fig. S2 (b)), or (3) ablation ice (site Abl6-2017).

The only exception were sites Abl3-2017 and Abl4-2017, placed in an accumulation pocket in the ablation zone of the glacier. As can be seen in Fig. S2 (c), site Abl4-2017 corresponded to the topmost layer of the pocket (which

disappeared toward the borders of the pocket, site Abl3-2017). However, based on the hardness, density, coarse grain size (738  $\pm$  167 µm) and high surface enrichment (PM content as high as (12000  $\pm$  2000) mg kg-1), we interpreted that this was a firn layer due to negative net accumulation during 2016-2017 hydrological year. The sub-surface firn layer of site Abl4-2017, with a low PM content, was attributed to firn accumulated during 2015 winter. Since those two layers have disappeared in site Abl3-2017, this area was identified as an area with even lower specific mass balance, where all accumulation from 2015-2016 hydrological year had also melted. The PM content, (30000  $\pm$  5000) mg kg-1, is consistent with the expected higher surface enrichment. The sub-surface firn layer was then attributed to accumulation during 2014 winter.

Supplement, Fig. S2 (see at the end of this file)

**Referee comment**

**Line 290/291: "low seasonal humidity" – do you mean variations?**

**Author's response**

We thank the referee for suggesting to clarify this sentence. During summer, snow melting exposes volcanic ash (and mineral dust) deposited in previous years in Monte Tronador and surrounding mountains. During the summer, when humidity is particularly low (such as in 2016 summer), mobility of ash and soil is higher, producing more relevant resuspension events.

**Manuscript Changes**

Lines 290-294:

The magnitude of resuspension events in Andean Patagonia, a region with strong, persistent westerlies and a dry season with low seasonal relative humidity, is well known. These aeolian remobilization events may produce huge ash clouds that may be even confused with true volcanic plumes, they can remobilize ash tenths of kilometers away (Toyos et al., 2017). In particular, the deposits of volcanic ash that are covered by snow during the winter in the high mountain usually become exposed to remobilization during the summer, travelling through the atmosphere and redepositing over different surfaces due to decrease of wind competence or by adherence of particles on humid surfaces, even at considerably high altitudes.

**Referee comment**

Line 328: "it was dated as winter snow from 2014" – how?

**Author's response**

*The interpretation was based in stratigraphic relationships as discussed for Line 268 comment (above).*

**Manuscript Changes**

One of the samples described under microscope, corresponds to a sub-surface sample from site *Abl3-2017*, it-which was dated interpreted as winter snow from 2014, previous to 2015 Calbuco eruption, and approximately 75 % of the observed particles correspond to fine-grained colourless pumiceous ash.

**Referee comment**

**Page 12:**

**Line 349:** "a single measurement" - what does that mean? One voltage reading? How stable is the voltage in time?

**Author's response**

The sentence means that in 2016 campaign the pyranometer was placed once towards incoming solar radiation and once towards radiation reflected by the snowpack. The voltage was stable during reading (up to the 0.1 mV resolution of the voltmeter), and hence we used the voltmeter resolution as the instrumental uncertainty. In 2017, the higher resolution voltmeter allowed to see changes in voltage readings. As we explain in the manuscript, we believe that this was due both to the higher resolution of the voltmeter and to faster changes in cloudiness.

**Manuscript Changes**

Lines 349-350:

For the 2016 campaign, the reported measured albedo is a single measurement (registered after voltage reached a stable value) and is informed together with its instrumental uncertainty.

**Referee comment Line 259: SNOW RADIUS!!!**

*Author's response* We thank the referee for the suggestion. *Manuscript Changes*

**Lines 359-361:**

In fresh snow samples from the accumulation zone (sites *Acc5-2017* and *Acc6-2017*) we found an average snow grain radius of  $(151 \pm 41) \mu m$ , whereas in samples 360 of older firm in the ablation zone (or sub-surface snow/firm in the accumulation zone) we measured values usually around  $(1000 \pm 200) \mu m$ .

**Referee comment**

**Table1:**

Why are there two values for the measured albedo in Abl4?

Why do you present the measured albedo in different lines? Should be always next to the modelled W.Aver?

**Author's response**

We thank the referee for the comments. For site Abl4-2017, we decided to register two sets of measurements, instead of one single set, due to the observed rapid movements of clouds. The irradiance values were significantly different in both sets, and so were the average albedo values. The second value is similar to the one measured in site Abl3-2017, and both are similar to the modeled value. The coincidence with the modeled value suggests that the sky pictures (taken after both sets of measurements) and cloud cover estimate represent better the sky conditions of the

**second set of measurements. Regarding the second comment, we do agree that the measured albedo should be always placed next to the weighted average modeled albedo.**

**Manuscript Changes**

**See modified Table 1 at the end of this file *Lines 376-379:**

For overcast conditions (*Acc3-2016, Abl3-2017* and *Abl4-2017*), the pure diffuse albedo from both models is also similar, and weighted average albedo from SNICARv2.1 is coincident with the pure diffuse albedo. For both models, the diffuse radiation spectrum for overcast conditions is coincident with global solar radiation spectrum (see Fig. 4), which explains the similar results. It must be noticed that for site Abl4-2017, we observed rapid cloud movements, and we decided to register two sets of albedo measurements, The average albedo of the second set is similar to the modeled weighted average albedo and to the measurement for site Abl3-2017. We suggest that this coincidence means that the pictures of the sky above the site (taken after the two sets of measurements) and the estimate of cloud cover based on those pictures represent more accurately the sky conditions during the second set of measurements.

**Referee comment**

**Last column:**

could you describe in the methods how you obtain these sensitivities? Are they really always symmetric? I do not understand the uncertainty associated to the concentration of BC? Why is is sometimes 100micrograns/kg and sometimes 20mg/kg.

These numbers have many zeros! Could you better indicate the percentual sensitivity and mark the most important contributor?

**Author's response**

We thank the referee for the comment. The sensitivity studies were performed modifying one parameter at a time in SINCARv2.1 calculations: for parameter "A", we calculated the albedo values  $\alpha(A+\Delta A)$ ,  $\alpha(A)$  and  $\alpha(A-\Delta A)$  (where  $\Delta A$  stands for the parameter uncertainty reported in the Table 1), keeping all other parameters unchanged. The sensitivities calculated in this way are not always symmetric: we expressed them as single range to make the table easier to read, but we accept the referee suggestion to show that asymmetry. However, we prefer to keep the expression of the observed albedo change (instead of percentage change) to better appreciate which significant figures of the modeled albedo are affected by each estimated sensitivity.

Regarding BC, we were not able to measure (yet) the carbon content of the samples, due to difficulties of equipment availability. We introduced a sensitivity study on BC content since one of the possible limitations of our simulations is the uncertainty regarding other LAP present in the samples aside from volcanic ash. The example value of 100  $\mu$ g/kg was chosen since is compatible with BC concentrations usually found on glacier surfaces (e.g., Ginot et al. 2014). For sites with higher LAP concentration, 100  $\mu$ g/kg of BC did not modify the modeled albedo, hence we decided to also calculate the impact of a higher amount of BC (20 mg/kg) to show how high it would need to be to have a similar impact in the albedo.

**Manuscript Changes**

**Table 1:**

We corrected the expression of the sensitivities in the last column to show that they are not symmetrical with respect to the parameters uncertainties. We highlighted the most important contributors for each site. See modified Table 1 at the end of this file.

Lines 407-410:

The last column in Table 1 reports the results of sensitivity studies to evaluate the impact on the calculated albedo of the uncertainty in key input parameters. We define the sensitivities as the modeled albedo changes increasing or decreasing one parameter in the same magnitude of its reported uncertainty (identified in Table 1 with a "+" or a "-" sign, respectively), while keeping all other parameters unchanged. The parameters have been modified in ranges allowed by the uncertainty of the input parameters. For each site, we studied PM content and grain size impact, together with other parameters that could be relevant at each site. We highlighted (with bold characters) the higher sensitivities for each site.

**Referee comment Page14.** Line 399: non-additive  $\rightarrow$  non-linear? *Author's response*  We thank the referee for the suggestion. We believe that in this context both phrases express almost the same meaning, but we prefer the expression "non-additive" since it remarks the fact that we are talking about the effect on albedo of two separate fractions of LAP. **Manuscript Changes** No changes were introduced.

**Referee comment* Page 15**

Line414/415: revise sentence starting with: "Volcanic ash ..."

**Author's response**

We thank the referee for the suggestion.

**Manuscript Changes**

Lines 414-415:

The uncertainty of  $\forall \underline{v}$  olcanic ash content uncertainty does not have a relevant impact for any of the sites, although it is larger for site *Abl4-2017*.

**Referee comment**

Line 419: what is a thin layer? Give number!

**Author's response**

We thank the referee for the suggestion. We added a reference to specific samples/sites and their thicknesses to clarify the affirmation.

**Manuscript Changes**

**Lines 419-421:**

The impact is maximum for very thin layers, especially when the underlying layer has a significantly different albedo (i.e., PM content)site *Abl4-2017*, 0.1 cm thick), and its minimum for the thicker layers (sites *Acc5-2017* or *Acc6-2017*, 9 cm thick), or for intermediate thicknesses with high PM content (i.e., low penetration of incident light, site *Abl3-2017*, 0.3 cm thick).

**Referee** comments**

**Page 16**

Line 442 Albedo and glacier mass balance **model:** up to now only modeled mass balance is analyzed

Line 443 "... glacier wide modeled annual and winter ..."

**Author's response**

We thank the referee for the suggestions. For the section title, we suggest a different phrasing that we find represents better the content of the section.

**Manuscript Changes**

Lines 442-444:

**3.4 Albedo and modeled impact on glacier mass balance**

Table 2 shows the glacier-wide modeled annual and winter mass balance, Equilibrium Line Altitude (ELA) and Accumulation Area Ratio (AAR) for different values of old snow albedo ( $\alpha_{firn}$ ).

**Referee comments**

Page 18 Line 510: delete "PM over" Line 519: delete "major"

**Author's response**

We thank the referee for the suggestions. Regarding the first comment, we do not agree: our manuscript focus on the impact of PM or LAP on albedo. Hence, we prefer not to delete the phrase. Regarding the second comment, we suggest an additional change that reflects better the intended meaning: the fact that volcanic ash are not only present, but that they represent the major fraction of the collected PM. Manuscript Changes Lines 519-521:

The major presence of fact that volcanic ash represents the largest fraction of the collected PM in all studied samples indicates that the effect of nearby volcanic eruptions are expected not only immediately after direct deposition, but also many years later, due to surface enrichment and wind resuspension and redeposition.

**Referee comment**

Line 523/524: please propose how to take account for that

**Author's response**

We thank the referee for the suggestion. While we do propose how to take account for the spatial heterogeneity of PM distribution at the end of the previous section, we agree that is appropriate to summarize that in the Conclusions as well.

**Manuscript Changes**

Lines 522-523:

These facts need to be accounted for when studying the effect of snow albedo on glacier mass balance. While the albedo parametrization used in the mass balance model partially accounts for the spatial heterogeneity of PM surface concentration (implicitly), we suggest that in the future it would be useful to couple our mass balance model with an atmospheric model which provides prognosis of PM content and a snow albedo model that includes LAP effect explicitly.

**Referee** comment**

**Page 19**

**Line 525: "We found that rapid changes …"** this is only a problem for your specific set-up. If you are able to measure upwelling and downwelling radiation simultaneously, this is not a problem. *Author's response*

We thank the referee for noticing the phrasing mistake. Indeed, we are not describing an inherent problem of albedo measurements but a limitation of our set-up. Using two pyranometers has other instrumental limitations that need to be aknowledged (specially, the need to account for the different sensitivities of the upward and downward sensor; Pirazzini, R., J.Geophys.Res., 109, D20118, 2004).

**Manuscript Changes**

Lines 525-526:

We found that for our set-up (where the pyranometer must be inverted sequentially to measure upwelling and downwelling radiation) rapid changes in cloudiness hinder the repeatability of albedo measurements and may degrade the comparison with modeled albedo.

**Referee** comment**

**Line 530:** "... suggesting strategies ..." which strategies are you suggesting? Which were the most important uncertainty?

**Author's response**

We thank the referee for the suggestion.

**Manuscript Changes**

**Lines 530-533:**

The effect of uncertainties of field measurements of snow properties was evaluated for different types of samples (lower or higher LAPPM content, grain size, layer thickness, snow density, etc.), suggesting strategies to reduce uncertainty in snow albedo modeling or retrieval of snow properties from measured albedo. We found that snow grain size must be measured more carefully in samples with low volcanic ash content and that the accuracy of layer thickness can be relevant not only for very thin layers (0.1 cm) but also for thicker layers (6 cm) with low ash content. The accuracy of ash content was found to be good enough for reproducing our albedo measurements. However, it was remarked that the presence of small amounts of BC can affect the albedo significantly in samples with low ash content.

**Referee comment**

**Line 534/535:** glacier-wide albedo change sensitivity : explain this sensitivities with words or indicate where it was defined.

**Author's response**

The glacier mass balance sensitivity to albedo change is defined at lines 445-447.

**Referee comment**

**Line 536:** how high concentration of volcanic ash do you need for this reduction in SMB? *Author's response**

We thank the referee for the question. The mentioned impact on the glacier mass balance was estimated with the minimum snow albedo value of 0.4 (see lines 459-468), which was based on the modeled daily average for site Abl4-2017, with an estimated volcanic ash content of (12000  $\pm$  2000) mg kg-1. However, we have calculated that the modeled albedo for site Acc3-2016 varies only 3.8% for ash contents between 4500 mg kg-1 and 10500 mg kg-1. Hence, the 0.4 albedo value can represent a range of sites with high volcanic ash content.

**Manuscript Changes**

Finally, we suggest that the effect of volcanic ashes in Alerce glacier can be as high as a 1.25 mwe decrease in the glacier annual mass balance or a 34 % of increase in the melt during the ablation season, considering a surface volcanic ash content compatible with that measured in sites Acc3-2016, Abl3-2017 and Abl4-2017.

**Referee comment**

**Figure 1:**

could you please show the outline of Alerce glacier in the map and contour of terrain elevation? Would also be nice to have another more zoom-out map to better see the glaciolocial context of Alerce Glacier.

Author's response

We thank the referee for the suggestion. We modified Fig. 1 to include the glacier outline and an inset with a zoom-out. Manuscript Changes

See modified Fig. 1 at the end of this file.

**Referee comment**

**Figure 2:**

what meaning has a white column color?

What do you think: why did you not find the dark layer at 45cm in Acc4 in Acc5?

Author's response

Regarding the first question, white color was not used in the concentration gray-scale, hence white color appears only at the depth where sampling ends (for instance, below 10 cm for site Acc6-2017).

Regarding the second question, we regret that weather conditions did not allow us to continue the snow spit in site Acc5-2017. We believe that the dark layer corresponding to the 2016 summer surface layer was not too far below. This area of the accumulation zone of the glacier has a high specific accumulation variation in very short surface distances.

**Referee comment**

**Figure 4:**

what are the units of the Y-Axis? Diffuse radiation should be less intense than the direct one!

**Author's response**

We thank the referee for the comment. The spectra shown here are normalized to highlight the difference in their wavelength dependence, hence the Y-Axis has arbitrary units. We have corrected the caption of Fig. 4 as a response to a similar question by Anonymous Referee #1. Manuscript Changes

We corrected the caption of Fig. 4, see Author's Response to Anonymous Referee #1, pages 6-7.

measurement and is informed together with its instrumental uncertainty. For 2017 campaign, we report the average and the standard error of the average for several Layer thickness: (+) +0.000 0001 (-) -0.000 002 Layer thickness: (+) +0.010 (-) -0.013 Layer thickness: (+) -0.001 (-) +0.007 Layer thickness: (+) -0.008 (-) +0.031 Layer thickness: (+) +0.001 (-) -0.002 Layer thickness: (+) +0.002 (-) -0.002 Effective angle: (+) +0.001 (-) -0.001 Effective angle: (+) + 0.001 (-) - 0.001Effective angle: (+) +0.001 (-) -0.001% Diff. Rad.: +0.001 (89 % diff. rad.) Snow density: (+) -0.001 (-) +0.003 % Diff. Rad.: -0.007 (89 % diff. rad.) Snow density: (+) -0.005 (-) +0.007 % Diff. Rad.: (+) +0.002 (-) -0.002% Diff. Rad.: (+) +0.005 (-) -0.002% Diff. Rad.: (+) +0.004 (-) -0.002PM content: (+) +0.001 (-) -0.001 PM content: (+) -0.004 (-) +0.006 Grain size: (+) -0.012 (-) +0.015 Grain size: (+) -0.013 (-) +0.015 PM content: (+) -0.001 (-) +0.001 PM content: (+) -0.001 (-) +0.002 PM content: (+) -0.001 (-) +0.001 PM content: (+) -0.001 (-) +0.001 Grain size: (+) -0.024 (-) +0.028 Grain size: (+) -0.001 (-) +0.002 Grain size: (+) +0.001 (-) -0.001 Grain size: (+) -0.001 (-) +0.002 100 μgkg-1 BC: -0.022 100 μgkg-1 BC: -0.021 100 μgkg-1 BC: -0.017  $20 \text{ mg kg}^{-1}$  BC: -0.015 $20 \text{ mg kg}^{-1}$  BC: -0.049 $20 \,\mathrm{mg\,kg^{-1}\,BC}: -0.050$ αSNICARv2 αSNICARv2.1 αSNICARv2.1 Sens. epetitions. For modeled albedo, sensitivity to different input parameters is reported, as an estimation of albedo uncertainty. W.Aver.: 0.359 W.Aver.: 0.356 W.Aver.: 0.825 W.Aver.: 0.368 Diffuse: 0.748 W.Aver.: 0.590 Diffuse: 0.364 Diffuse: 0.359 Diffuse: 0.354 Diffuse: 0.368 Diffuse: 0.860 Diffuse: 0.910 W.Aver.:0.828 Diffuse: 0.905 Direct: 0.583 Direct: 0.573 Direct: 0.445 Direct: 0.381 Direct: 0.384 Direct: 0.778 Direct: 0.776 Diffuse: 0.856 Diffuse: 0.655 Diffuse: 0.360 Diffuse:0.375 Direct: 0.435 Direct: 0.374 Direct: 0.379 Direct: 0.788 Direct: 0.786  $0.626 \pm 0.011$  $0.257 \pm 0.041$  $0.266 \pm 0.008$  $0.376 \pm 0.015$  $0.814 \pm 0.013$  $0.757 \pm 0.026$  $0.371 \pm 0.011$  $\alpha_{meas}$ PM:  $(30\,000\pm5000)$  mg kg-1 PM:  $(12\,250\pm2050)\,\mathrm{mg\,kg^{-1}}$  $\rm PM:\,(7800\pm1500)\,mg\,kg^{-1}$ Grain size:  $(1000 \pm 200) \,\mu m$ Grain size:  $(1000 \pm 200) \, \mu m$ Grain size:  $(1020 \pm 160) \, \mu m$ PM: (1.28  $\pm$  0.03) mg kg^{-1} PM:  $(22.0 \pm 0.6) \text{ mgkg}^{-1}$ Grain size:  $(740 \pm 170) \, \mu m$ Snow density:  $500 \, \mathrm{kg \, m^{-3}}$ Snow density:  $500 \text{ kg m}^{-3}$ Snow density:  $500 \, \rm kg \, m^{-3}$ Snow density:  $300 \, \mathrm{kg \, m^{-3}}$ Snow density:  $300 \, \text{kg m}^{-3}$ Grain size:  $(150 \pm 40) \, \mu m$ Snow density:  $300 \text{ kg m}^{-3}$ Grain size:  $(150 \pm 40) \, \mu m$  $\rm PM:\,(3.9\pm0.2)\,mg\,kg^{-1}$ Layer:  $(0.10 \pm 0.05) \text{ cm}$ Layer:  $(0.3 \pm 0.1) \text{ cm}$ Layer:  $(0.3 \pm 0.1) \text{ cm}$ Dirty summer snow Layer:  $(9 \pm 1) \text{ cm}$ Layer:  $(6 \pm 1) \text{ cm}$ Layer:  $(9 \pm 1)$  cm Recent snow Recent snow Recent snow Slope: 11° Dirty snow Slope: 15° Dirty snow Slope: 15° Slope: 5° Slope: 5° Slope: 0° Surface Overcast sky (approx. 100 % diffuse rad.) Overcast sky (approx. 100 % diffuse rad.) Abl3-2017 (accum. pocket on abl. zone) Overcast sky (89 % to 95 % diffuse rad.) Abl4-2017 (accum. pocket on abl. zone) Cloudy sky (34 % to 48 % diffuse rad.) Cloudy sky (34 % to 48 % diffuse rad.) Effective angle: 68.7° to 73.6° Effective angle: 56.5° to 59.9° Effective angle: 57.1° to 60.1° Effective angle: 49.4° to 52.3° Effective angle: 51.0° to 53.9° Acc2-2016 (accum. zone) Acc3-2016 (accum. zone) Acc5-2017 (accum. zone) Acc6-2017 (accum. zone) Effective angle: 51.98° 16:55 (UTC - 3) 14:47 (UTC - 3) 14:26 (UTC - 3) 14:48 (UTC - 3) 13:11 (UTC - 3) 13:30 (UTC - 3) April 12th 2016 April 12th 2016 Zenith: 51.98° Zenith: 66.04° April 3rd 2017 April 3rd 2017 April 5th 2017 April 5th 2017 Zenith: 47.57° Zenith: 46.95° Zenith: 48.20° Zenith: 49.35° Clear Sky Site

[able 1. Measured and modeled snow albedo for six sites (two in 2016 campaign and four in 2017 campaign). For 2016 campaign the measured albedo is a single

---

## Author Response (AR3)

**Author's response to Benjamin Smith's Editor Decision of 06 September 2020**

***Editor comment***
Comments to the Author:
The manuscript looks good. I have a small list of edits that I saw during a quick read-through, and more may be caught during typesetting. Please make these corrections and we can get the manuscript off to the next stage!

Overall: I don't think 'firn' needs to be in italics.
***Author's response***
*We followed the Editor's suggestion and removed italics from the word "firn" throughout the manuscript.*

***Editor comments***
Line (in the change-tracked manuscript) – comment
18: detect -> identify
23: retreating -> retreat (or 'rates of retreat')
30: delete 'the'
***Author's response***
*We agree with the suggestions. In line 23, we prefer "rates of retreat".*

***Editor comment***
33 "light-absorbing impurities" -> "light-absorbing particles"
***Author's response***
*We agree. We fixed the same mistake in two other paragraphs (lines 83 and 87).*

***Editor comments***
44 "PM" should be treated as singular (e.g. "PM reflects")
45 relevant -> significant
345: have shown -> have been shown
346: "in a similar way as in" -> "similar to the effect of debris in "
***Author's response***
*We agree with the suggestions.*

***Editor comment***
372: "One of the samples described under microscope" : Not clear what this means.
***Author's response***
*We rephrased the sentence.*
*Manuscript changes:*
  a sub-surface sample from site Abl3-2017, which was interpreted as winter snow from 2014, previous to 2015 Cal eruption,  we found that approximately 75 % of the observed particles correspond to fine-grained colourless pumiceous ash.

***Editor comments***
467: "would allow to study" -> "would allow us to study"
481: "such kind of " -> such
485 that-> those
501 "has shown" -> "has been shown"
***Author's response***
*We agree with the suggestions.*

**Author's response to Benjamin Smith's Editor Decision**

***Editor comment***

Editor Decision: Publish subject to minor revisions (review by editor) (07 Aug 2020) by Benjamin Smith
Comments to the Author:
There is some sort of a problem with the TC system, and I am not able to see the authors' revised manuscript today. As I'll be out of town next week, I wanted to provide some feedback based on what I can see in the responses to the referees.

*Author's response*

*We thank the Editor's decision to publish the revised manuscript. We answer the specific comments and suggestions below.*

***Editor comment***

It looks to me like the authors have done a good job of responding to the referees' comments, and I think the major scientific disagreements have been dealt with well. At the same time, based on the samples of the revised text that are available in the response documents, I suspect that the final manuscript will need some revisions for English and grammar. In particular:
--the response to lines 228-234 has a description of methods in present tense, while all methods should be in past tense

*Author's response*

*We thank the Editor's grammar suggestion. We changed all verbs in the mentioned paragraph to the past tense.*

***Editor comments***

--In the abstract, "microscopical" should be "microscopic"
--Lines 29-33: temperature increase -> temperature increases
--Lines 37-44 : "snow grains growth" -> "snow grains' growth" or "growth of snow grains"
--Lines 50-52: should be "online coupling...has"
In the block of supplement text starting with line 34: "placed" should be "located"
... and so on.

*Author's response*

*We appreciate the Editor's grammar suggestions. We accepted them all, with exception of the second one. The word "increase" is not used here as a verb but as a noun.*

***Editor comment***

I'm afraid I don't have time for a thorough proof-read of the manuscript, but I'd ask that the authors look carefully at their text with all revisions applied and check the grammar throughout before submitting a final draft. The manuscript is very close!

*Author's response*

*We thank the Editor's suggestion. We made several additional grammar or spelling corrections, which are included in the marked-up version of the manuscript at the end of this file. We also improved some of the figure captions to follow more closely the journal guidelines.*

*Following this response, we include the responses to both reviewers (already posted as Author Comments in the interactive discussion) and a marked-up version of the manuscript.*

**Author's response to Marius Schaefer's review**

*Referee comment*
**General Comments:**
The manuscript presents punctual albedo measurements over snow surfaces on different parts of a small glacier in the Northern Patagonian Andes in two consecutive years, together with measurements of physical parameters which could mostly explain the measured albedo variations (like grain size and form and particulate matter content). Then the authors try to reproduce the measured albedos, using a model, which is improved to account for partly cloudy conditions (which were present at least at one of the field days). In a last step the possible influence of the ash content, caused by eruptions of nearby volcanoes, on the total glacier surface mass balance is estimated using a simplified energy balance/mass balance model.
To my point of view the study is original and novel and fits well into the scope of the journal. I think that the significance of the study could be significantly increased by adding some additional data and analysis, which should not be too difficult to obtain and which would allow to better interpret the presented field data and model results:

*Author's response*
*We appreciate the referee's thorough and useful comments to improve the manuscript. Although the suggested additions would increase the significance of the article, some of them are outside the focus of this manuscript. The manuscript already deals with field measurements and models. Including the use of remote sensing data would make it excessively long. We discuss the suggested additions point by point next.*

*Referee comment*
1) Measured surface mass data at stakes: I think that the surface mass balance data measured at stakes were somehow used to interpret the sample obtained form the snow pits (section 3.1, Figure 2) but the detailed data are not indicated. Also in section 2.4 it is stated: "The model was calibrated by surface mass balance measurements performed on a seasonal to annual basis through the year 2016 over Alerce glacier". I would like to know more details about this calibration process. How well could the model reproduce the observed melt and accumulation of snow? Which alpha_firn values fitted best to the observations? The time series of measured surface mass balance could also be helpful for quantifying the impact of the volcanic eruptions on the glacier's surface mass balance.

*Author's response*
*A comprehensive evaluation of the mass balance of Alerce glacier is beyond the scope of this work and it is core of an ongoing manuscript by one of the members of the author team (Lucas Ruiz). We included in Fig. 1 the location of ablation stakes, and in the Supplement (Fig. S4) the location of snow thickness measurements. Detail regarding the process of calibration of the surface mass balance model (SMB model) was added in Sect. 2.4 together with two new figures in the Supplement (Fig S5 and S6) which shows the agreement between modeled and measurements used to calibrate the SMB model and the fitting of the model for two of the ablation stakes close to the albedo sampling locations.*
*For the hydrological years 2015 and 2016 (during and after the Calbuco eruption) best agreement between measurements and model was achieved using minimum snow albedo values of 0.42-0.38. The range express the difficulty to achieve a straightforward calibration of the different parameters used in enhanced degree-day models. Some parameters counteract each other and minimum RMSEs could be achieved with a variety of parameter combination. Thus, it is also necessary considering surface characteristics at the stakes locations and their distribution across the glaciers, like transient snow lines or extra mass balance measurements through the year.*

*Manuscript Changes*
*Lines 228-234:*
 Potential direct solar radiation for all grid cells and days was calculated following Hock (1999). The local surface albedo $\alpha(x,y,t)$ was taken to be constant for bare-ice

surfaces ($\alpha_{ice}$ = 0.34), using most commonly applied literature value (Oerlemans and Knap, 1998; Cuffey and Paterson, 2010), for snow surfaces, $\alpha_{snow}$ was calculated based on the snow aging function proposed by Oerlemans and Knap (1998) with a maximum snow albedo ($\alpha_{max}$) of 0.8 and a  minimum snow albedo ($\alpha_{min}$) adjusted during the calibration procedure. .

The model was calibrated in two steps using surface mass balance measurements of year 2016 in Alerce glacier (Supplement, Fig. S4). First, the model is run over the winter period with an initial set of constants ($c_0$ and $c_1$) and a guess for the precipitation correction factor $C_{pre}$. As melt is of minor importance in winter, this run is used to calibrate $C_{pre}$, that scales $D_s$ for every snow fall event. After a good agreement of measured and calculated winter accumulation is obtained, the model is run over the entire year and the remaining constants are calibrated so that the root-mean-square error between modelled and observed point annual balances is minimized and the average misfit is close to zero (Supplement, Fig. S5 and S6). A random set of snow accumulation and ablation stakes measurements performed through the year and not used to calibrate the model are left apart to validate the results of the surface mass balance model.

We studied glacier-wide mass balance changes for different values of $\alpha_{min}$  (Table 2), which are indicative of the sensitivity of glacier mass balance to a change in albedo that might occur in response to the darkening of the glacier surface.

*Supplement, Fig. S4, S5 and S6 (see at the end of this file).*

**Referee comment**

2) I am surprised by the big influence of alpha_firn on the modeled surface mass balance of the glacier. In a "normal" year I would expect to have no firn in the ablation area and the firn of the accumulation area being buried by snow most of the year. How did you initialize the model (regarding presence of snow, firn, ice). Was 2016 a typical year? Probably not since autumn 2016 was exceptionally dry in the region. I would propose to run the model with a few years of "typical" meteorologic data (mean value of several years) and standard firn albedo for model initialization and then start to study the influence of different firn albedos. I think it should be much lower on average.

**Author's response**

*We acknowledge that the use of $\alpha_{firn}$ as a synonymous of minimum snow albedo was not a good choice and give place to confusions. As we stated in Section 2.4, $\alpha_{firn}$ is the minimum albedo that snow could reach using the snow aging function of Oerlemans and Knap (1998). We replaced $\alpha_{firn}$ for $\alpha_{min}$ to avoid any confusion. We agree that if we had only changed the albedo of the firn (the snow accumulated after more than year, for instances), the effect on the surface mass balance would have been much lower.*

*The model is initiated with a guess snow and firn lines and run for a few days before the evaluated period, which is observational period. to stabilize the surface mass balance to the input data. We have tested different initiation scenarios, to check the sensitivity of the model to initial conditions, and under realistic scenarios, the sensitivity is rather low.*

*Finally, we agree with the reviewer, 2016 was the driest year since we start the monitoring of the Alerce glacier in 2013.*

**Manuscript Changes**

*We replaced $\alpha_{firn}$ for $\alpha_{min}$ throughout the manuscript.*

**Referee comment**

3) Since the albedo measurements are very punctual in time and space, and, as your repeating in the text several times that particulate matter concentration is very variable in time and space, it would be great to get an idea about the significance of your punctual albedo measurements by analyzing for example optical reflectance in satellite images. Images obtained at dates near to your field campaigns could be used for calibration. By this means you could also easily go back until the 2011 Cordon Caulle eruption. Would be great to see how the reflectance of the glacier changed from summer 2011 to 2012. Or from summer 2015 to 2016.

**Author's response**

*Satellite observations are relevant, and we have already look at MODIS products and other remote data for a following article. Although satellite snow reflectance data could be used to evaluate the significance of our punctual surface measurements (albedo measurements, particles content and*

*snow grain size), Landsat and Sentinel images close to the timing of our measurements are totally or partially cloud covered for Monte Tronador. As we stated in the manuscript cloudiness conditions were challenging and we needed to update SNICAR model to deal with it. Regarding the use of MODIS, although the time resolution allows us to have more images without excessive cloud cover, it spatial resolutions challenges the evaluation against punctual surface measurements. Nevertheless, our preliminary evaluation of MODIS albedo time series of Monte Tronador, shown a decrease in late summer albedo after the Cordon Caulle and Calbuco eruption, with a minimum during the late summer of 2017 (both a combination of the ashes and less snow fallen over the glacier). Nevertheless, as we mention above, these additional analysis would require a considerable amount of space, hence we decided to keep them for another manuscript where we can deal properly with it.*

**Referee comment**
**Technical Comments:**
Your abstract is 350 words which is too long (instructions form the journal's web page copied below). Try to reduce! For example you have three introducing sentences. One should be enough!
**Research articles** report substantial and original scientific results within the journal's scope. Generally, these are expected to be within 12 journal pages, have appropriate figures and/or tables, a maximum of 80 references, and an abstract of 150–250 words.

*Author's response*
*We thank the referee for the suggestion. We have already reduced the length of the abstract following a suggestion of the Anonymous Referee #1. We present here a further effort of making the abstract more concise.*

**Manuscript Changes**
*Abstract*

The impact of volcanic ash on seasonal snow and glacier mass balance has been much less studied than that of carbonaceous particles and mineral dust. We present here the first field measurements on Argentinian Andes, combined with snow albedo and glacier mass balance modeling. Measured impurities content (1.1 mg kg−1 to 30 000 mg kg−1) varied abruptly in snow pits and snow/firn cores, due to high surface enrichment during the ablation season and possibly local/regional wind driven resuspension and redeposition of dust and volcanic ash. In addition, we observed a high spatial heterogeneity, due to glacier topography and prevailing wind direction. Microscopical characterization showed that the major component was ash from recent Calbuco (2015) and Cordón Caulle (2011) volcanic eruption, with a minor presence of mineral dust and Black Carbon. We also found a wide range of measured snow albedo (0.26 to 0.81), which reflected mainly the impurities content and the snow/firn grain size (due to aging). We updated the SNICAR snow albedo model to account for the effect of cloudiness on incident radiation spectra, improving the match of modeled and measured values. We also ran sensitivity studies considering the uncertainty of the main measured parameters (impurities content and composition, snow grain size, layer thickness, etc) to detect the field measurements that should be improved to facilitate the validation of the snow albedo model. Finally, we studied the impact of these albedo reductions in Alerce glacier using a spatially distributed surface mass-balance model. We found a large impact of albedo changes in glacier mass balance, and we estimated that the effect of observed ash concentrations can be as high as a 1.25 meter water equivalent decrease in the glacier-wide annual mass balance (due to a 34 % of increase in the melt during the ablation season).

**Referee comment**
**Detailed Comments:**
**Page2**
**Line 26:** Patagonian Andes or Wet Andes instead of Southern Andes ? ( to be more precise).

*Author's response*
*We agree with the referee that the suggested terms are more precise, we rephrased.*

**Manuscript Changes**
*Lines 25-26:*

Along the Wet Andes (below 35º S latitude), melt is driven by shortwave radiation and sensible turbulent flux (Schaefer et al., 2019).

**Referee comment**
**Line 27:** you mean net shortwave ? Albedo is not influencing the oncoming shortwave radiation. I

would say summer, since in spring glaciers are mostly snow covered and exhibit high albedos

*Author's response*

*We thank the referee for the comments. Regarding the first comment, we rephrased the sentence in order to make sure the meaning of the sentence is transparent. Regarding the second comment, we agree that the exposure of low albedo layers is much more significant in summer.*

*Manuscript Changes*

*Lines 25-29:*

The  shortwave radiation absorption increases significantly  during  summer, due to the exposure of low albedo areas in their ablation zones, which causes strong, positive feedback that enhances surface melt significantly and shapes the spatial ablation pattern (Brock et al., 2000).

*Referee comment*

**Line 29 – until Page3 Line72:** in this section you discuss the influence of light-absorbing impurities on snow albedo. You mention particulate matter, mineral dust, volcanic ash and black carbon). Are all particulate matter light-absorbing impurities? Are mineral dust, volcanic ash and black carbon both particulate matter and light-absorbing impurities? Perhaps order these definitions in an introducing sentence and avoid synonyms ( particulate matter = light-absorbing impurities?)

*Author's response*

*We agree with the referee that the original manuscript was not clear enough regarding these definitions, as was also pointed out by Anonymous Referee #1.*

*Manuscript Changes*

*We introduced several changes that are detailed in the **Author's Response to Anonymous Referee #1,** pages 2-4.*

*Referee comment*

**Line 31:** produced → producing

*Author's response*

*We thank the referee for the useful phrasing suggestion.*

*Manuscript Changes*

*Lines 29-31:*

Furthermore, deposition of light-absorbing impurities (LAP; mineral dust, volcanic ash, and black carbon) have a fundamental impact on the melting of glacier and snow-covered areas by increasing the absorption of solar radiation and producing a regional land-atmosphere feedback

*Referee comment*

**Line 32:** "the growth of snow grains is accelerated" explain when and why.

*Author's response*

*We accept the referee's suggestion to further explain this effect. We rephrased two sentences to better explain the direct and indirect effects of LAP on snow.*

*Manuscript Changes*

*Lines 29-33:*

Furthermore, deposition of light-absorbing impurities (LAP; mineral dust, volcanic ash, and black carbon) have a fundamental impact on the melting of glacier and snow-covered areas  (Warren and Wiscombe, 1980; Bond et al., 2013; Molina et al., 2015). LAP decrease snow albedo, increasing solar radiation absorption and thus producing a direct effect on snow melting. But, in addition, the snowpack temperature increase due to the direct effect accelerates the growth of snow grains , which produces a further albedo decrease (and thus an additional, indirect impact on snow melting) (Bond et al., 2013; Flanner et al., 2007).

*Referee comment*

**Line 38:** "as well as several positive feedbacks" which one?

*Author's response*

*The thorough review by Bond et al. (2013) describes in detail the multiple rapid changes in snow due to LAP deposition (see Fig. 29 of the reference). We added in the text two of the more important feedback processes and refer the reader to the reference.*

*Manuscript Changes*

*Lines 37-40:*

Different snow albedo models have been developed to include the direct effect of Black Carbon (BC) and other LAP  as well as several positive feedbacks (Flanner et al., 2007; Koch et al., 2009; Krinner et al., 2006), such as the increase in surface concentration of impurities due to enhanced snow melting, or the albedo reduction due to snow grains growth by accelerated snow aging (Bond et al., 2013). More recently, models have included the effects of non-spherical snow grains (Libois et al., 2013; He et al., 2017), and external/internal mixing of impurities with snow grains (He et al., 2018).

**Referee comment**

**Line 42:** do not understand the sentence. What is a particle metric distribution?

*Author's response*

*We agree with the referee that sentence needs rephrasing. We hope that this new phrasing gives a better, concise description of the main results of the references, and help the reader to find further details therein.*

*Manuscript Changes*

*Lines 42-43:*

When the snow has been undergoing heavy metamorphosis processes, a single snow grain size distribution is not enough to reproduce the snow spectral albedo, due to the fact that the largest particles and the thinnest protrusions of the irregular crystals have contributions to the snow reflectance that depend on the wavelength (Carmagnola et al., 2013; Pirazzini et al., 2015)

**Referee comment**

**Line 45:** explain broadband albedo

*Author's response*

*We thank the referee's question. We rephrased the sentence to explain more clearly the results in Zhang et al., 2018.*

*Manuscript Changes*

Notably, there has been found that taking into account the amount of LAP in the snow reduces the difference between simulated and measured  albedos, specially in the visible range (Zhang et al., 2018).

**Referee comment**

**Line 50:** what is "online coupling"?

*Author's response*

*We agree with the referee that the phrase might not be clear for some readers. We use the term "online coupling" to imply that the two models (snow albedo model and atmospheric chemistry model) are run simultaneously and allowing two-way feedback. Other studies use offline coupling, where one of the models (usually, the atmospheric chemistry model) is run first, and the results are used as input for the other model (snowpack model or glacier mass balance model).*

*Manuscript Changes*

*Lines 50-52:*

"Online" coupling of snow albedo models in global or regional atmospheric chemistry models (where both models are run simultaneously allowing two-way feedback) have been applied to study snow and glaciers interaction with the climate around the globe (Hansen et al., 2005; Flanner, 2013; Ménégoz et al., 2014).

**Referee comment**

**Page3**

**Lines 67-68:** do not understand the sentence starting with "For example ..." Reformulate!

*Author's response*

*We rephrased the sentence.*

*Manuscript Changes*

*Lines 67-68:*

For example, the albedo reduction  due to BC alone in the north was estimated to be only about 43 % of that for all light-absorbing impurities (assuming spherical 100 μm radii snow grains).

*Referee comment*
**Page4**
**Line 94:** I think the mass balance model is not mentioned in Ruiz et al 2017
*Author's response*
*We thank the referee for noticing the mistake, we corrected the position of the references regarding the Alerce glacier monitoring and we added a new one regarding the mass balance model.*
*Manuscript Changes*
*Lines 91-94:*
Since 2013 it has been the focus of a glacier mass balance monitoring program by the IANIGLA (Instituto Argentino de Nivología, Glaciología y Ciencias Ambientales; Ruiz et al., 2015, 2017**).** Seasonal mass balance has been studied every year using the traditional glaciological method of stakes, and snow pits. An enhanced temperature index mass balance model has been developed (Huss et al., 2008; Huss, 2010) to study the surface mass balance of the glacier.

*Referee comment*
**Page 5:**
**Line 124:** … "with a" … → … with one ...
*Author's response*
*We thank the referee for the useful suggestion.*
*Manuscript Changes*
Upwelling (reflected) and downwelling (direct + diffuse) radiation were measured with one CM5 Kipp & Zonen pyranometer (wavelength range 0.3 μm to 2.8 μm), using two different in-house developed supports in 2016 and 2017 campaigns, logged with a handheld voltmeter.

*Referee comment*
**Line 126:** How much W/m² is 0.1mv?
*Author's response*
*For reference, 0.1 mV represents approximately 9.5 $W/m^2$ for our pyranometer. We did not find relevant to include the conversion factor in the article since we do not report solar irradiances, but only measured albedos (the conversion factor is not relevant for the radiation ratios).*

*Referee comment*
**Page 6**
**Line 166:** "High-resolution pictures" … Would be great if you could show them in the supplementary material
*Author's response*
*We added a figure in the Supplement (Fig. S3).*
*Manuscript Changes*
*Lines 166-167:*
High-resolution pictures (Fig. S3, Supplement) where analyzed later with ImageJ software (Schneider et al., 2012).
*Supplement, Fig. S3 (see at the end of this file)*

*Referee comment*
**Page 7**
**Line 173/174** "are decribed in detail in section 3.2" → (Section 3.2)
**Line 180:** for → of
*Author's response*
*We thank the referee for the useful suggestion.*
*Manuscript Changes*
*We adopted the suggested changes.*

*Referee comment*

**Page 8**

**Line 221&228 I** could not open the links indicated for the weather stations! Please indicate distance from glacier and elevation for both stations!

*Author's response*

*We thank the referee for noticing the mistake, we corrected the links and added the altitude of the stations.*

*Manuscript Changes*

*Line 221:*

$P(t)$ was the daily precipitation at Tepual weather station (90 m altitude, ID = 857990; https://www.ncei.noaa.gov/access/search/data-search/global-summary-of-the-day)

*Lines 277-228:*

$T(t)$ was taken from the air surface temperature at Bariloche airport weather station (846 m altitude, ID = 877650; https://www.ncei.noaa.gov/access/search/data-search/global-summary-of-the-day).

*Referee comment*
**Page 9**
**Line 251/252:** on the base of what is this interpretation?

*Author's response*

*The interpretation of the snow/firn layers is based on the observed stratigraphy of the snow column. Snow pits walls and cores were described following common glaciological practices, in terms of layering, grain size and shape, content of PM, density and hardness. Dating of layers or attribution of time windows for each layer was based on the stratigraphic relations between layers and its characteristics. In this case, the layer (242 cm to247 cm) had a high PM concentration, was below a thick, relative low PM content, soft snow layer (interpreted as the snow accumulated during the accumulation season of the hydrological year 2015-2016) and above a harder, coarser grained firn layer (interpreted as the snow of the accumulation season of 2014-2015).*

*Manuscript Changes*

The deepest (242 cm to 247 cm deep) thin, high PM concentration layer (($1970 \pm 200$) mg kg$^{-1}$) was interpreted as the surface at end of the ablation season of the hydrological year 2014-15, based on the abrupt change of the density, hardness and grain size of the snow above this layer and the firn found below.

*Referee comment*
**Page 10**
**Line 262:** Abl2-2016 → Abl1-2016?

*Author's response*

*We thank the referee for noticing the mistake*

*Manuscript Changes*

*We corrected the mistake in the sampling site name.*

*Referee comment*
**Line 264:** "These sites ..." which one? Abl3 and Abl4 ? In Abl2 and Abl5 PM content also seems to be quite high!

*Author's response*

*The sentence refers to the sites Abl3-2017 and Abl4-2017, mentioned in the previous sentence, but we changed the sentence to avoid any misunderstanding. The PM content on the surface layer of those sites, ($30000 \pm 5000$) mg kg$^{-1}$ and ($12000 \pm 2000$) mg kg$^{-1}$ respectively, is much higher than that of any other site, due to the reasons explained in the manuscript and in the new section S2 of the Supplement (see response to the next comment). Sites Abl2-2016 and Abl5-2017 had a surface layer of recent snow. Below the surface layer, the PM content of the summer surface layer of site Abl2-2016 was ($4400 \pm 800$) mg kg$^{-1}$. Site Abl5-2017 presented glacier ice below the surface layer (which was not sampled).*

*Manuscript Changes*

*Lines 264-265:*

Sites *Abl3-2017* and *Abl4-2017* had a negative net balance during hydrological year 2016-17, consequently the surface layer presented the highest PM content observed in both campaigns

**Referee comment**
**Line 268: "** firn layer from 2015 winter" – how do you know?
*Author's response*
*The layers from sites Abl3-2017 and Abl4-2017 (placed close to each other in the same accumulation pocket, see new Fig. S2 at the end of this file) were identified based on stratigraphic relationships. The dark surface at site Abl4-2017 was the topmost layer of the pocket, but based on the grain size (738 ± 167 μm), density and hardness, we interpreted that all accumulation from 2016 winter had melted. The high PM concentration (12000 ± 2000) mg kg$^{-1}$ was also consistent with the surface enrichment due to melting of snow deposited in more than one hydrological year. The firn below this layer was then identified as the accumulation layer from 2015 winter. In site Abl3-2017, towards the border of the accumulation pocket, the topmost layers described for site Abl4-2017 had also disappeared. Hence, we interpreted that all accumulation from 2015 winter had also melted in this site, and this darkest, surface layer contained most of PM deposited in 2016 and 2015. The firn layer below was interpreted as the accumulation layer from 2014 winter.*
*Manuscript Changes*
*Lines 267-270:*
In-situ stratigraphy revealed that in *Abl4-2017* site, the high concentration layer was on top of relatively low concentration, firn layer from 2015 winter, which means that, during the 2016-2017 ablation season, all the snow accumulated during 2016 winter was melted. Site *Abl3-2017* presented an even lower net balance, revealing older firn (winter 2014) below the surface high concentration layer. See Sect. S2 in Supplement for additional details on the attribution of layers in sites Abl3-2017 and Abl4-2017.
*Supplement, line 34:*
**S2 Dating of snow/firn layers**
Most snow/firn layers sampled during both field campaigns were easily dated, considering that the topmost layer contains the most recent snow and attributing layers below based on PM content, density, hardness and grain size. Topmost layers were identified as:
(1) fresh snow from a recent deposition events, on the accumulation zone, (sites Acc1-2016, Acc2-2016, Acc4-2017, Acc5-2017, Acc6-2017 and Acc7-2017, Fig S2(a)), on an accumulation pocket (site Abl1-2016), or on top of ablation ice (sites Abl2-2016 and Abl5-2017),
(2) end-of-ablation season surface, with high enrichment of PM content (Acc3-2016, Fig. S2 (b)), or
(3) ablation ice (site Abl6-2017).
The only exception were sites Abl3-2017 and Abl4-2017, placed in an accumulation pocket in the ablation zone of the glacier. As can be seen in Fig. S2 (c), site Abl4-2017 corresponded to the topmost layer of the pocket (which disappeared toward the borders of the pocket, site Abl3-2017). However, based on the hardness, density, coarse grain size (738 ± 167 μm) and high surface enrichment (PM content as high as (12000 ± 2000) mg kg$^{-1}$), we interpreted that this was a firn layer due to negative net accumulation during 2016-2017 hydrological year. The sub-surface firn layer of site Abl4-2017, with a low PM content, was attributed to firn accumulated during 2015 winter. Since those two layers have disappeared in site Abl3-2017, this area was identified as an area with even lower specific mass balance, where all accumulation from 2015-2016 hydrological year had also melted. The PM content, (30000 ± 5000) mg kg$^{-1}$, is consistent with the expected higher surface enrichment. The sub-surface firn layer was then attributed to accumulation during 2014 winter.
*Supplement, Fig. S2 (see at the end of this file)*

**Referee comment**
**Line 290/291:** "low seasonal humidity" – do you mean variations?
*Author's response*
*We thank the referee for suggesting to clarify this sentence. During summer, snow melting exposes volcanic ash (and mineral dust) deposited in previous years in Monte Tronador and surrounding mountains. During the summer, when humidity is particularly low (such as in 2016 summer), mobility of ash and soil is higher, producing more relevant resuspension events.*
*Manuscript Changes*
*Lines 290-294:*
The magnitude of resuspension events in Andean Patagonia, a region with strong, persistent westerlies and a dry season with low  relative humidity, is well known. These aeolian remobilization events may produce huge ash clouds

that may be even confused with true volcanic plumes, they can remobilize ash tenths of kilometers away (Toyos et al., 2017). In particular,  deposits of volcanic ash that are covered by snow during the winter in the high mountain usually become exposed to remobilization during the summer, travelling through the atmosphere and redepositing over different surfaces due to decrease of wind competence or by adherence of particles on humid surfaces, even at considerably high altitudes.

*Referee comment*
**Page 11**
**Line 328:** "it was dated as winter snow from 2014" – how?
*Author's response*
*The interpretation was based in stratigraphic relationships as discussed for Line 268 comment (above).*
*Manuscript Changes*
One of the samples described under microscope, corresponds to a sub-surface sample from site *Abl3-2017*,  which was  interpreted as winter snow from 2014, previous to 2015 Calbuco eruption, and approximately 75 % of the observed particles correspond to fine-grained colourless pumiceous ash.

*Referee comment*
**Page 12:**
**Line 349:** "a single measurement" - what does that mean? One voltage reading? How stable is the voltage in time?
*Author's response*
*The sentence means that in 2016 campaign the pyranometer was placed once towards incoming solar radiation and once towards radiation reflected by the snowpack. The voltage was stable during reading (up to the 0.1 mV resolution of the voltmeter), and hence we used the voltmeter resolution as the instrumental uncertainty. In 2017, the higher resolution voltmeter allowed to see changes in voltage readings. As we explain in the manuscript, we believe that this was due both to the higher resolution of the voltmeter and to faster changes in cloudiness.*
*Manuscript Changes*
*Lines 349-350:*
For the 2016 campaign, the reported measured albedo is a single measurement (registered after voltage reached a stable value) and is informed together with its instrumental uncertainty.

*Referee comment*
**Line 259: SNOW RADIUS!!!**
*Author's response*
*We thank the referee for the suggestion.*
*Manuscript Changes*
*Lines 359-361:*
In fresh snow samples from the accumulation zone (sites *Acc5-2017* and *Acc6-2017*) we found an average snow grain radius of $(151 \pm 41)$ μm, whereas in samples 360 of older firn in the ablation zone (or sub-surface snow/firn in the accumulation zone) we measured values usually around $(1000 \pm 200)$ μm.

*Referee comment*
**Table1:**
Why are there two values for the measured albedo in Abl4?
Why do you present the measured albedo in different lines? Should be always next to the modelled W.Aver?
*Author's response*
*We thank the referee for the comments. For site Abl4-2017, we decided to register two sets of measurements, instead of one single set, due to the observed rapid movements of clouds. The irradiance values were significantly different in both sets, and so were the average albedo values. The second value is similar to the one measured in site Abl3-2017, and both are similar to the modeled value. The coincidence with the modeled value suggests that the sky pictures (taken after both sets of measurements) and cloud cover estimate represent better the sky conditions of the*

*second set of measurements. Regarding the second comment, we do agree that the measured albedo should be always placed next to the weighted average modeled albedo.*

*Manuscript Changes*

*See modified Table 1 at the end of this file*

*Lines 376-379:*

For overcast conditions (*Acc3-2016*, *Abl3-2017* and *Abl4-2017*), the pure diffuse albedo from both models is also similar, and weighted average albedo from SNICARv2.1 is coincident with the pure diffuse albedo. For both models, the diffuse radiation spectrum for overcast conditions is coincident with global solar radiation spectrum (see Fig. 4), which explains the similar results. It must be noticed that for site Abl4-2017, we observed rapid cloud movements, and we decided to register two sets of albedo measurements, The average albedo of the second set is similar to the modeled weighted average albedo and to the measurement for site Abl3-2017. We suggest that this coincidence means that the pictures of the sky above the site (taken after the two sets of measurements) and the estimate of cloud cover based on those pictures represent more accurately the sky conditions during the second set of measurements.

*Referee comment*

**Last column:**

could you describe in the methods how you obtain these sensitivities? Are they really always symmetric? I do not understand the uncertainty associated to the concentration of BC? Why is is sometimes 100micrograns/kg and sometimes 20mg/kg.

These numbers have many zeros! Could you better indicate the percentual sensitivity and mark the most important contributor?

*Author's response*

*We thank the referee for the comment. The sensitivity studies were performed modifying one parameter at a time in SINCARv2.1 calculations: for parameter "A", we calculated the albedo values α(A+ΔA), α(A) and α(A-ΔA) (where ΔA stands for the parameter uncertainty reported in the Table 1), keeping all other parameters unchanged. The sensitivities calculated in this way are not always symmetric: we expressed them as single range to make the table easier to read, but we accept the referee suggestion to show that asymmetry. However, we prefer to keep the expression of the observed albedo change (instead of percentage change) to better appreciate which significant figures of the modeled albedo are affected by each estimated sensitivity.*

*Regarding BC, we were not able to measure (yet) the carbon content of the samples, due to difficulties of equipment availability. We introduced a sensitivity study on BC content since one of the possible limitations of our simulations is the uncertainty regarding other LAP present in the samples aside from volcanic ash. The example value of 100 μg/kg was chosen since is compatible with BC concentrations usually found on glacier surfaces (e.g., Ginot et al. 2014). For sites with higher LAP concentration, 100 μg/kg of BC did not modify the modeled albedo, hence we decided to also calculate the impact of a higher amount of BC (20 mg/kg) to show how high it would need to be to have a similar impact in the albedo.*

*Manuscript Changes*

*Table 1:*

*We corrected the expression of the sensitivities in the last column to show that they are not symmetrical with respect to the parameters uncertainties. We highlighted the most important contributors for each site. See modified Table 1 at the end of this file.*

*Lines 407-410:*

The last column in Table 1 reports the results of sensitivity studies to evaluate the impact on the calculated albedo of the uncertainty in key input parameters. We define the sensitivities as the modeled albedo changes increasing or decreasing one parameter in the same magnitude of its reported uncertainty (identified in Table 1 with a "+" or a "–" sign, respectively), while keeping all other parameters unchanged.  For each site, we studied PM content and grain size impact, together with other parameters that could be relevant at each site. We highlighted (with bold characters) the higher sensitivities for each site.

*Referee comment*

**Page14.**

**Line 399: non-additive → non-linear?**

*Author's response*

*We thank the referee for the suggestion. We believe that in this context both phrases express almost the same meaning, but we prefer the expression "non-additive" since it remarks the fact that we are talking about the effect on albedo of two separate fractions of LAP.*

*Manuscript Changes*

*No changes were introduced.*

*Referee comment*

**Page 15**

**Line414/415:** revise sentence starting with: "Volcanic ash ..."

*Author's response*

*We thank the referee for the suggestion.*

*Manuscript Changes*

*Lines 414-415:*

The uncertainty of volcanic ash content  does not have a relevant impact for any of the sites, although it is larger for site *Abl4-2017*.

*Referee comment*

**Line 419:** what is a thin layer? Give number!

*Author's response*

*We thank the referee for the suggestion. We added a reference to specific samples/sites and their thicknesses to clarify the affirmation.*

*Manuscript Changes*

*Lines 419-421:*

The impact is maximum for very thin layers, especially when the underlying layer has a significantly different albedo (site *Abl4-2017*, 0.1 cm thick), and its minimum for the thicker layers (sites *Acc5-2017* or *Acc6-2017*, 9 cm thick), or for intermediate thicknesses with high PM content (i.e., low penetration of incident light, site *Abl3-2017*, 0.3 cm thick).

*Referee comments*

**Page 16**

**Line 442** Albedo and glacier mass balance **model:** up to now only modeled mass balance is analyzed

**Line 443 "…** glacier wide **modeled** annual and winter **…"**

*Author's response*

*We thank the referee for the suggestions. For the section title, we suggest a different phrasing that we find represents better the content of the section.*

*Manuscript Changes*

*Lines 442-444:*

**3.4 Albedo and modeled impact on glacier mass balance**

Table 2 shows the glacier-wide modeled annual and winter mass balance, Equilibrium Line Altitude (ELA) and Accumulation Area Ratio (AAR) for different values of old snow albedo ($\alpha_{firn}$).

*Referee comments*

**Page 18**

**Line 510:** delete "PM over"

**Line 519:** delete "major"

*Author's response*

*We thank the referee for the suggestions. Regarding the first comment, we do not agree: our manuscript focus on the impact of PM or LAP on albedo. Hence, we prefer not to delete the phrase. Regarding the second comment, we suggest an additional change that reflects better the intended meaning: the fact that volcanic ash are not only present, but that they represent the major fraction of the collected PM.*

*Manuscript Changes*

*Lines 519-521:*

The fact that volcanic ash represents the largest fraction of the collected PM in all studied samples indicates that the effect of nearby volcanic eruptions are expected not only immediately after direct deposition, but also many years later, due to surface enrichment and wind resuspension and redeposition.

*Referee comment*
**Line 523/524:** please propose how to take account for that
*Author's response*
*We thank the referee for the suggestion. While we do propose how to take account for the spatial heterogeneity of PM distribution at the end of the previous section, we agree that is appropriate to summarize that in the Conclusions as well.*
*Manuscript Changes*
*Lines 522-523:*

These facts need to be accounted for when studying the effect of snow albedo on glacier mass balance. While the albedo parametrization used in the mass balance model partially accounts for the spatial heterogeneity of PM surface concentration (implicitly), we suggest that in the future it would be useful to couple our mass balance model with an atmospheric model which provides prognosis of PM content and a snow albedo model that includes LAP effect explicitly.

*Referee comment*
**Page 19**
**Line 525: "We found that rapid changes ..."** this is only a problem for your specific set-up. If you are able to measure upwelling and downwelling radiation simultaneously, this is not a problem.
*Author's response*
*We thank the referee for noticing the phrasing mistake. Indeed, we are not describing an inherent problem of albedo measurements but a limitation of our set-up. Using two pyranometers has other instrumental limitations that need to be aknowledged (specially, the need to account for the different sensitivities of the upward and downward sensor; Pirazzini, R., J.Geophys.Res., 109, D20118, 2004).*
*Manuscript Changes*
*Lines 525-526:*

We found that for our set-up (where the pyranometer must be inverted sequentially to measure upwelling and downwelling radiation) rapid changes in cloudiness hinder the repeatability of albedo measurements and may degrade the comparison with modeled albedo.

*Referee comment*
**Line 530: "… suggesting strategies ..."** which strategies are you suggesting? Which were the most important uncertainty?
*Author's response*
*We thank the referee for the suggestion.*
*Manuscript Changes*
*Lines 530-533:*

The effect of uncertainties of field measurements of snow properties was evaluated for different types of samples (lower or higher LAP content, grain size, layer thickness, snow density, etc.), suggesting strategies to reduce uncertainty in snow albedo modeling or retrieval of snow properties from measured albedo. We found that snow grain size must be measured more carefully in samples with low volcanic ash content and that the accuracy of layer thickness can be relevant not only for very thin layers (0.1 cm) but also for thicker layers (6 cm) with low ash content. The accuracy of ash content was found to be good enough for reproducing our albedo measurements. However, it was remarked that the presence of small amounts of BC can affect the albedo significantly in samples with low ash content.

*Referee comment*
**Line 534/535:** glacier-wide albedo change sensitivity : explain this sensitivities with words or indicate where it was defined.
*Author's response*
*The glacier mass balance sensitivity to albedo change is defined at lines 445-447.*

*Referee comment*

**Line 536:** how high concentration of volcanic ash do you need for this reduction in SMB?

*Author's response*

*We thank the referee for the question. The mentioned impact on the glacier mass balance was estimated with the minimum snow albedo value of 0.4 (see lines 459-468), which was based on the modeled daily average for site Abl4-2017, with an estimated volcanic ash content of $(12000 \pm 2000)$ mg kg$^{-1}$. However, we have calculated that the modeled albedo for site Acc3-2016 varies only 3.8% for ash contents between 4500 mg kg$^{-1}$ and 10500 mg kg$^{-1}$. Hence, the 0.4 albedo value can represent a range of sites with high volcanic ash content.*

*Manuscript Changes*

Finally, we suggest that the effect of volcanic ashes in Alerce glacier can be as high as a 1.25 mwe decrease in the glacier annual mass balance or a 34 % of increase in the melt during the ablation season, considering a surface volcanic ash content compatible with that measured in sites Acc3-2016, Abl3-2017 and Abl4-2017.

*Referee comment*
**Figure 1:**

could you please show the outline of Alerce glacier in the map and contour of terrain elevation? Would also be nice to have another more zoom-out map to better see the glaciolocial context of Alerce Glacier.

*Author's response*

*We thank the referee for the suggestion. We modified Fig. 1 to include the glacier outline and an inset with a zoom-out.*

*Manuscript Changes*

*See modified Fig. 1 at the end of this file.*

*Referee comment*
**Figure 2:**

what meaning has a white column color?
What do you think: why did you not find the dark layer at 45cm in Acc4 in Acc5?

*Author's response*

*Regarding the first question, white color was not used in the concentration gray-scale, hence white color appears only at the depth where sampling ends (for instance, below 10 cm for site Acc6-2017).*

*Regarding the second question, we regret that weather conditions did not allow us to continue the snow spit in site Acc5-2017. We believe that the dark layer corresponding to the 2016 summer surface layer was not too far below. This area of the accumulation zone of the glacier has a high specific accumulation variation in very short surface distances.*

*Referee comment*
**Figure 4:**

what are the units of the Y-Axis?
Diffuse radiation should be less intense than the direct one!

*Author's response*

*We thank the referee for the comment. The spectra shown here are normalized to highlight the difference in their wavelength dependence, hence the Y-Axis has arbitrary units. We have corrected the caption of Fig. 4 as a response to a similar question by Anonymous Referee #1.*

*Manuscript Changes*

*We corrected the caption of Fig. 4, see **Author's Response to Anonymous Referee #1**, pages 6-7.*

**Table 1.** Measured and modeled snow albedo for six sites (two in 2016 campaign and four in 2017 campaign). For 2016 campaign the measured albedo is a single measurement and is informed together with its instrumental uncertainty. For 2017 campaign, we report the average and the standard error of the average for several repetitions. For modeled albedo, sensitivity to different input parameters is reported, as an estimation of albedo uncertainty.

| Site | Surface | $\alpha_{meas}$ | $\alpha_{SNICARv2}$ | $\alpha_{SNICARv2.1}$ | $\alpha_{SNICARv2.1}$ sens. |
|---|---|---|---|---|---|
| **Acc2-2016** (accum. zone)
**April 12th 2016**
14:47 (UTC - 3)
Zenith: 51.98°
Effective angle: 51.98°
Clear Sky | Recent snow
Layer: (6 ± 1) cm
Grain size: (1000 ± 200) μm
Snow density: 300 kg m⁻³
PM: (22.0 ± 0.6) mg kg⁻¹
Slope: 0° | 0.626 ± 0.011 | Direct: 0.583
Diffuse: 0.655 | Direct: 0.573
Diffuse: 0.748
W.Aver.: 0.590 | **Grain size:** (+) −0.024 (-) +0.028
PM content: (+) −0.001 (-) +0.001
100 μg kg⁻¹ **BC:** −0.017
**Layer thickness:** (+) +0.010 (-) −0.013 |
| **Acc3-2016** (accum. zone)
**April 12th 2016**
16:55 (UTC - 3)
Zenith: 66.04°
Effective angle: 68.7° to 73.6°
Overcast sky (approx. 100 % diffuse rad.) | Dirty summer snow
Layer: (0.3 ± 0.1) cm
Grain size: (1000 ± 200) μm
Snow density: 500 kg m⁻³
PM: (7800 ± 1500) mg kg⁻¹
Slope: 11° | 0.257 ± 0.041 | Direct: 0.435
Diffuse: 0.364 | Direct: 0.445
Diffuse: 0.359
W.Aver.: 0.359 | Grain size: (+) −0.001 (-) +0.002
PM content: (+) −0.001 (-) +0.002
20 mg kg⁻¹ BC: −0.049
**Layer thickness:** (+) −0.001 (-) +0.007
Snow density: (+) −0.001 (-) +0.003
% Diff. Rad.: +0.001 (89 % diff. rad.) |
| **Abl3-2017** (accum. pocket on abl. zone)
**April 3rd 2017**
13:11 (UTC - 3)
Zenith: 47.57°
Effective angle: 56.5° to 59.9°
Overcast sky (89 % to 95 % diffuse rad.) | Dirty snow
Layer: (0.3 ± 0.1) cm
Grain size: (1020 ± 160) μm
Snow density: 500 kg m⁻³
PM: (30 000 ± 5000) mg kg⁻¹
Slope: 15° | 0.371 ± 0.011 | Direct: 0.374
Diffuse: 0.360 | Direct: 0.381
Diffuse: 0.354
W.Aver.: 0.356 | Grain size: (+) +0.001 (-) −0.001
PM content: (+) +0.001 (-) −0.001
20 mg kg⁻¹ BC: −0.015
Layer thickness: (+) +0.0000001 (-) −0.000002
% Diff. Rad.: (+) +0.002 (-) −0.002
Effective angle: (+) +0.001 (-) −0.001 |
| **Abl4-2017** (accum. pocket on abl. zone)
**April 3rd 2017**
13:30 (UTC - 3)
Zenith: 46.95°
Effective angle: 57.1° to 60.1°
Overcast sky (approx. 100 % diffuse rad.) | Dirty snow
Layer: (0.10 ± 0.05) cm
Grain size: (740 ± 170) μm
Snow density: 500 kg m⁻³
PM: (12 250 ± 2050) mg kg⁻¹
Slope: 15° | 0.266 ± 0.008
0.376 ± 0.015 | Direct: 0.379
Diffuse:0.375 | Direct: 0.384
Diffuse: 0.368
W.Aver.: 0.368 | Grain size: (+) −0.001 (-) +0.002
PM content: (+) −0.004 (-) +0.006
20 mg kg⁻¹ BC: −0.050
**Layer thickness:** (+) −0.008 (+) +0.031
Snow density: (+) −0.005 (-) +0.007
% Diff. Rad.: −0.007 (89 % diff. rad.) |
| **Acc5-2017** (accum. zone)
**April 5th 2017**
14:26 (UTC - 3)
Zenith: 48.20°
Effective angle: 49.4° to 52.3°
Cloudy sky (34 % to 48 % diffuse rad.) | Recent snow
Layer: (9 ± 1) cm
Grain size: (150 ± 40) μm
Snow density: 300 kg m⁻³
PM: (1.28 ± 0.03) mg kg⁻¹
Slope: 5° | 0.814 ± 0.013 | Direct: 0.788
Diffuse: 0.860 | Direct: 0.778
Diffuse: 0.910
W.Aver.:0.828 | **Grain size:** (+) −0.012 (-) +0.015
PM content: (+) −0.001 (-) +0.001
100 μg kg⁻¹ **BC:** −0.022
Layer thickness: (+) +0.002 (-) −0.002
% Diff. Rad.: (+) +0.005 (-) −0.002
Effective angle: (+) +0.001 (-) −0.001 |
| **Acc6-2017** (accum. zone)
**April 5th 2017**
14:48 (UTC - 3)
Zenith: 49.35°
Effective angle: 51.0° to 53.9°
Cloudy sky (34 % to 48 % diffuse rad.) | Recent snow
Layer: (9 ± 1) cm
Grain size: (150 ± 40) μm
Snow density: 300 kg m⁻³
PM: (3.9 ± 0.2) mg kg⁻¹
Slope: 5° | 0.757 ± 0.026 | Direct: 0.786
Diffuse: 0.856 | Direct: 0.776
Diffuse: 0.905
W.Aver.: 0.825 | **Grain size:** (+) −0.013 (-) +0.015
PM content: (+) −0.001 (-) +0.001
100 μg kg⁻¹ **BC:** −0.021
Layer thickness: (+) +0.001 (-) −0.002
% Diff. Rad.: (+) +0.004 (-) −0.002
Effective angle: (+) +0.001 (-) −0.001 |

[Figure]

**Figure 1.** Alerce Glacier (green line represents the outline of the glacier). Labels of contour lines of terrain elevation are expressed in meters above sea level. Sampling points are represented as blue rhombuses. Red circles represent ablation stakes used for mass balance model calibration (model output for labeled ablation stakes is shown at Figure S6). Otto Meiling mountain hut and inset of the location of Monte Tronador in the context of Southern SouthAmerica are represented for reference. Background image: false-color pan-sharpened Pléiades satellite image, 7 March 2012, PGO, CNES-Airbus D & S (Ruiz and others, 2015).

[Figure]

(a)                                                                                          (b)

[Figure]

**Abl4-2017**

**Abl3-2017**

(c)

Figure S2. Field pictures of sampling sites. (a) General view of the area of the accumulation zone that includes sites Acc1-2016, Acc2-2016, Acc3-2016, Acc4-2017, Acc5-2017, Acc6-2017 and Acc7-2017.
(b) Close view of surface of site Acc3-2016 (darkest layer, bottom of the picture), next to recent fresh snow (top of the picture). (c) Accumulation pocket in the ablation area of sites Abl3-2017 and Abl4-2017.

[Figure]

(a)

(b)

(c)

(d)

**Figure S3.** High resolution macro pictures of snow/firn grains, Alerce glacier, 2017. (a) Surface fresh snow sample, April 2017, site *Acc5-2017*. (b) Surface snow/*firn* sample, attributed approximately to April 2016 (due to negative specific mass balance), site *Abl4-2017*. (c) Sub-surface *firn* sample, attributed to winter 2015, site *Abl4-2017*. (d) Sub-surface *firn* sample, attributed to winter 2014, site *Abl3-2017*. In all pictures the green bar width represents 1 mm.

[Figure]

**Figure S4.** Snow thickness and ablation stakes used for calibration and validation of the mass balance model. Blue rhombuses are albedo and PM sampling points (same as in Fig. 1 of the main article). Background image: false-color pan-sharpened Pléiades satellite image, 7 March 2012, PGO,CNES-Airbus D & S (Ruiz and others, 2015).

[Figure]

**Figure S5.** Calibration of the mass balance model with 2016 measurements. (a) First step of the calibration, with winter thickness measurements. (b) Second step of the calibration, with summer specific mass balance measurements (ablation stakes).

[Figure]

**Figure S6.** Mass balance model fitting with 2016 measurements for two specific ablation stakes. The residuals between measurements and model it is shown for comparison. Location of (a) stake A VI and (b) stake A are represented in Fig. 1 of the main article.

**Author's response to Anonymous Referee #1**

***Referee comment***
This paper gives a thorough account of April (2016 and 2017) field measurements conducted on the Alerce Glacier in the Northern Patagonian Andes. Combined with an updated Snow, Ice, and Aerosol Radiative (SNICAR) model that accounts for partly cloudy conditions, the measurements are used to estimate the glacier's April 2016 – April 2017 surface mass balance. Representing the first particulate matter concentration, albedo, and grain size measurements conducted on the Alerce Glacier, these results are a valuable contribution to the community and therefore warrant consideration for publication in *The Cryosphere*. Before acceptance, however, there are specific concerns, provided below, followed by a list of technical corrections that I recommend the authors consider in a minor revision.

***Author's response***
*We deeply appreciate the referee for the thorough and useful comments to improve the manuscript.*

***Referee comment***
Throughout the manuscript, the authors refer to an average snow grain radius value that they claim (in Sect. 2.2) to be precise. Average radii values were obtained using two methods: from visual inspection against a crystal grid, which is outdated, and from ImageJ software, which, to my knowledge, is not a standard method for obtaining snow grain radius. Although these methods provide one estimate of snow grain size (e.g., the length of maximum dimension), they will not yield a precise optically equivalent snow grain radius (nor specific surface area) that is the relevant quantity in two-stream snow radiative transfer algorithms like the SNICAR model. To reduce a potential source of error regarding the SNICAR modeling results, I suggest placing a greater emphasis on the other measured quantities used as inputs into the SNICAR model, especially the light absorbing particle (LAP) concentrations.

***Author's response***
*We appreciate the referee's comment. Regarding the method in the first field campaign, we do agree is outdated, but it was the only method available for the first, exploratory campaign. Regarding the improved method we used in the second field campaign, which averages the maximum and minimum axes of equivalent ellipses that fit the snow grains in the pictures, we believe that it gives a reasonable estimate of the particles dimensions. We want to clarify that we do not claim that it is "precise", but only "more precise" than the previous method. The main evidence in support of our grain size results is that the differences among measured albedo values for fine and coarse snow can be explained using these grain size values in SNICARv2.1 model.*
*Nevertheless, we do agree that the snow grain size measurement method could be further improved. Pirazzini et al. (2015, cited in the manuscript) also use 2D photos, but with a different metric. They suggest that their metric is a proxy for "half the width of the shortest particle dimension", which they claim is a better approximation of the optically equivalent snow grain radius. If that is the case, our results would overestimate the optically equivalent snow grain radius. Pirazzini et al. determined 11% uncertainty in the 2D photos metrics (due to the subjectivity of the software operators). Although we did not determine such kind of uncertainty in our measurements, we report the estimated effect of the dispersion of the grain size for each sample, through sensitivity studies on SNICARv2.1 model. Even though the dispersions are large (probably larger than the uncertainty of the method), the effect on the modeled albedo are lower than 4.5% (for clean snow) or lower than 0.8% (for dirty snow). We believe that this explains the fact that we can reproduce the measured albedo using the estimated grain size together with other snow properties (especially LAP content), even though our grain size estimate might not be as accurate as that obtained by other methods. Spectral albedo (not available in our field campaigns) would be a complementary approach to validate separately the effect of snow grain size and LAP content on our albedo results. For instances, Carmagnola et al. (2013, cited in the manuscript) measured snow SSA (indirectly,*

*through an IR optical method) independently of LAP content, which mostly affects UV-vis albedo (lines 33-35 of the manuscript).*

*We modified the manuscript to include the limitations of our snow grain size measurement method, and we also modified the discussion of the results to remark that snow grain size results might not be as accurate as that from other measurements.*

**Manuscript Changes**

**Sect. 2.2, lines 165-168**

In the 2017 campaign, a similar in-house developed grid was used (with two scales: 1 mm and 0.5 mm) in combination with a macro lens and a mobile phone digital camera. High-resolution pictures where analyzed later with ImageJ software (Schneider et al., 2012). Snow grains were manually fitted with ellipses; the metric choice was the average of the minor and major axes of the ellipse. The new equipment and methodology introduced in the 2017 campaign allows a more detailed description of the snow samples and a more precise average radius value.

**Sect. 3.3, lines 358-361**

Regarding snow grain sizes, it is relevant to notice the range of observed average radius. In fresh snow samples from the accumulation zone (sites Acc5-2017 and Acc6-2017) we found an average radius of $(151 \pm 41)$ μm, whereas in samples of older firn in the ablation zone (or sub-surface snow/firn in the accumulation zone) we measured values usually around $(1000 \pm 200)$ μm. Pirazzini et al. (2015) also used 2D photos, but with a different metric. They suggest that SSK (shortest skeleton branch) is a proxy for "half the width of the shortest particle dimension", which they claim is a better approximation of the optically equivalent snow grain radius. Our metric (see Sect. 2.2) would probably give higher results than SSK, and hence we might have overestimated the optically equivalent snow grain radius. Nevertheless, as we show below in this section, our grain size measurements seem to be good enough to reproduce the measured albedo for fine and coarse snow in SNICARv2.1 snow albedo model.

**Sect. 3.3, lines 403-406**

In the other hand, comparison between sites with low PM content shows that snow grain size has a remarkable effect, as previously reported (Wiscombe and Warren, 1980; Hadley and Kirchstetter, 2012). Fresh snow with small grain size presents $\alpha_{meas} \approx 0.8$ (sites Acc5-2017 and Acc6-2017), but snow with similar PM content that has aged a few days presents $\alpha_{meas} \approx 0.6$ (site Acc2-2016). Spectral albedo measurements (not available in our field campaigns) would allow to study separately the effect of grain size and LAP content (see for instances measurements of snow specific surface area, SSA, in Carmagnola et al., 2013), to confirm that our grain size measurements are a good estimate of the optically equivalent grain radius.

**Sect. 3.3, lines 411-414**

Concerning grain size uncertainty (the standard deviation of snow grain radii in each sample), it is clear that the impact on albedo is much larger when PM content is low (sites Acc2-2016, Acc5-2017 and Acc6-2017). For low PM content sites, the effect is comparable to experimental uncertainty, and is relevant both for sites with finer and coarser grain sizes snow. For sites with high content of PM the uncertainty of grain size do not have an appreciable effect. Pirazzini et al. (2015) determined 11% uncertainty in the grain size measurements from 2D photos (due to the subjectivity of the software operators). Although we did not determine such kind of uncertainty in our measurements, we suggest that the reported standard deviation (between 16% and 26% of the average value) is probably larger than the uncertainty of the method. The sensitivity studies showed that the effect on the modeled albedo is lower than 4.5% for clean snow and lower than 0.8% for dirty snow. We believe that this explains the fact that we can reproduce the measured albedo using the estimated grain size together with other snow properties (especially LAP content), even though our grain size estimate might not be as accurate as that obtained by other methods.

**Referee comment**

Regarding the use of terminology, a reader would benefit from a brief description of the distinction, if any, between LAPs and particulate matter (PM). The abstract begins by stating the relevance of light absorbing impurities in snow studies, however, the results and discussion most frequently refer to PM. Because "LAP" is a well known acronym, I suggest either maintaining the convention used in the literature, or defining PM while also elucidating the reason for the use of "PM" to describe these particular measurements.

**Author's response**

*We agree with the referee that we should stress the difference between both expressions. The gravimetric measurements presented on this manuscript must be attributed to PM deposited on the glacier, because we don't know precisely the fraction of LAP among total PM. Qualitative observations also reported here (field stratigraphies and microscopy observations) suggest that most of collected PM can be attributed to volcanic ash. Quantifying the fraction of ash in collected PM (and/or measure the contributions from 2011 Cordon-Caulle and 2015 Calbuco eruptions) was not amenable. Routine stereo microscope inspection, even at high magnification (up to 80x), did not*

*allow quantification; only estimates of percentage of dark components was possible due to the fine-grained components. Na₂O, K₂O and SiO₂ content (SEM-EDS) from individual particles helped to distinguish volcanic ash from both eruptions, as exemplified in the manuscript (Fig. 8). In addition, CaO and FeO contents also proved useful to distinguish Cordón Caulle volcanic ash from ash derived from the 2015 Calbuco eruption (not included in the manuscript), but measuring a representative number of particles through SEM is not feasible. Hence, at this moment we can only suggest that most of PM on these samples correspond to volcanic ash, and that is the reason why we used SNICAR's built-in volcanic ash optical properties without further tuning. We know this is not exact, since optical microscopy and SEM microscopy have shown evidence of a minor fraction of mineral dust and black carbon, but we believe our results show that this assumption is a good first order approach to understand snow albedo on the glacier surface. A follow up article will include further chemical characterization of these samples, which has been delayed due to several reasons. We modified the manuscript to introduce a clear distinction between PM and LAP (LAI in our previous version of the manuscript), and we checked that those terms were used consistently through the manuscript.*

***Manuscript Changes***

- *We replaced the acronym "LAI" by "LAP" throughout the text.*
- *We replaced the acronym "PM" by "LAP" in some paragraphs, to clarify that we refer to the effect of particles that absorb light:*

***Sect. 1, lines 73-82:***

[revised manuscript text omitted]

***Referee comment***
Although I found Sect. 3 to be well written, I recommend the following technical corrections regarding mostly the other sections and figures:
1. *Abstract (lines 1–4):* Background could be refined, perhaps by moving one or two of the sentences into Sect. 1, to quickly introduce the present work.
*Manuscript Changes*
*Abstract, lines 1-4:*
The effect of volcanic ash in snow albedo (and its impact in seasonal snow and glacier mass balance) has been much less studied than that of other light absorbing impurities such as carbonaceous particles and mineral dust.  We present here the first field measurements on Argentinian Andes, combined with albedo and mass balance modeling activities.

***Referee comment***
2. *Abstract (line 6):* "during ablation" → "during the ablation"
3. *Abstract (line 9):* "from recent...eruption, with minor" → "from the recent...eruptions, with a minor"
4. *Abstract (lines 11–12):* "SNICAR model has been updated to model snow albedo taking into account" → "We updated the SNICAR model to account for"
*Author's response*
*We thank the referee for the useful grammar/phrasing suggestions.*
*Manuscript Changes*
*We adopt all changes suggested by the referee.*

***Referee comment***
5. *Abstract (line 14):* This part seems like an important component of this study, yet, it took me two or three times to understand the meaning of this sentence. Perhaps "which field measurements precision" can be rephrased to improve the readability.
*Author's response*
*We agree with the referee regarding the readability of the sentence, we rephrased it.*
*Manuscript Changes*
*Abstract, line 14:*
We also ran sensitivity studies considering the uncertainty of the main measured parameters (impurities content and composition, snow grain size, layer thickness, etc)to detect the field measurements that should be improved to facilitate the validation of the snow albedo model.

***Referee comment***
6. *Abstract (line 17):* "m we" → "m snow water equivalent (SWE)"
*Author's response*
*We thank the referee's suggestion, but we believe that both abbreviations for the snow water equivalence are widely used. We do agree that it needs to be defined in the abstract, and also in the main text.*
*Manuscript Changes*
*Abstract, line 17*
1.25 meter water equivalent decrease
*Sect. 3.4, line 447*
−0.6 meter water equivalent per year (m w.e./yr )

***Referee comment***
7. *Sect. 1 (line 20):* I like this opening, but the first sentence needs to begin with "Since" or "Because."

***Author's response***
*We agree with the referee's suggestion.*

***Manuscript Changes***
*Abstract, line 20*

Since laciers are highly sensitive to climate fluctuations, their unprecedented retreating rates observed during the last decades represent one of the most unambiguous signals of climate change

***Referee comments***
8. *Sect. 1 (line 29):* It's probably better to use the term "light-absorbing particles (LAP)" (Skiles et al., 2018).
9. *Sect. 1 (line 38):* What is the distinction between LAP and atmospheric particulate matter?

***Author's response***
*See response and changes in the discussion regarding PM and LAP above in this file.*

***Referee comments***
10. *Sect. 1 (line 44):* "there has been found" → "it has been shown"
11. *Sect. 2 (lines 88-89):* "the hydrological year is defined from the 1-April to the 31-March of the next year. The accumulation season last from 1-April to 31-October and the ablation season from 31-October to the 31-March of the next year." → "the hydrological year begins on April 1st with the accumulation season. The accumulation season lasts until October 31st, which marks the beginning of the ablation season."

***Author's response***
*We thank the referee for the useful grammar/phrasing suggestions.*

***Manuscript Changes***
*We adopt all changes suggested by the referee.*

***Referee comments***
12. *Fig. 1 (caption):* It might be good practice to include the term "true color" in the description to indicate that the image is intended to reproduce a natural color rendition.
13. *Fig. 2 (caption):* It might be good practice to indicate that the grayscale used is logarithmic.

***Author's response***
*We thank the referee for the useful suggestions.*

***Manuscript Changes***
*Fig. 1 caption:*

**Figure 1.** True color satellite image of Alerce Glacier. Sampling points are represented as blue markers (2017 campaign) and red markers (2016 campaign). Green marker represents Otto Meiling mountain hut. Copyright: © Google Earth, 2020, CNES/Airbus

*Fig. 2 caption:*

**Figure 2.** PM concentration (grayscale) as a function of pit depth for different sampling sites. Notice that the grayscale is logarithmic. Top panel: accumulation zone. Bottom panel: ablation zone. $\alpha$ symbol is used to highlight sites with concurrent albedo measurements. In sample *Abl2-2016*, the top rectangle corresponds to the average PM content of the first two layers (fresh snow and end-of-summer dark layer).

***Referee comment***
14. *Sect. 2.2 (line 125):* Please provide additional details of the "in-house developed supports" in order to improve the reproducibility of results.

***Author's response***
*We added a Figure at the Supplement to provide additional construction details on the supports.*

***Manuscript Changes***
**Sect 2.2, line 150**

Additional details on the supports are given in Fig. S1 in Supplement.

*Supplement, Fig. S1*

[Figure]

*Figure S1. Details of the pyranometer supports in Fig. 3 of the main text. (a) Side view and (b) top view of the support used in 2016 field campaign. (c) Side view of the support used in 2017 field campaign.*

**Referee comments**

15. *Sect. 2.2 (line 127):* "In the 2017 a" → "In the 2017 campaign, a"
16. *Sect. 2.2 (subsection headings):* Are these subsection headings supposed to be numbered (i.e., 2.2.1, 2.2.2, and 2.2.3)?

*Author's response*
*We agree with the referee's suggestions.*
**Manuscript Changes**
*We adopt the referee's suggestions.*

**Referee comment**

17. *Sect. 2.2 (line 153):* Equation (S1) has now been referred to twice. Should it be included in the main text?

*Author's response*
*We believe that the details of the albedo measurements corrections (including Eq. S1) are not needed in the main text.*
**Manuscript Changes**
*We did not find the need to introduce any changes.*

**Referee comment**

18. *Sect. 2.3 (line 189):* Is $I_{glob} = I_{dir} + I_{diff}$, or something else? Perhaps this can be more clearly described in Sect. 2.2.

*Author's response*
*Yes, $I_{glob} = I_{dir} + I_{diff}$, as usually defined, but we accept the suggestion to remark that in the manuscript.*
**Manuscript Changes**
**Sect. 2.3 (line 189)**

$I_{dir}$, $I_{diff}$, and $I_{glob}$ are clear-sky direct, diffuse and global solar irradiance (where $I_{glob} = I_{dir} + I_{diff}$), as calculated from SMARTS model.

**Referee comment**

19. *Fig. 4*: If the vertical axis represents a normalized, dimensionless quantity, please indicate so. Otherwise, please provide the meaning of the vertical dimension. Also, the right-most part of the figure (horizontal axis) appears clipped.

*Author's response*
*We thank the referee for the comment. The plotted distributions are indeed normalized, we modified the caption to make that clear. We corrected the clipping of the image to show the last horizontal axis label correctly.*

- *We corrected the clipping of the image.*
- *Fig. 4 caption:*

**Figure 4.** Different normalized spectral distributions of sun radiation for SNICAR snow albedo model. SNICARv2 included two spectra for mid-latitude locations: one for overcast conditions (light green line), and one for clear sky conditions (dark green line). SMARTS diffuse (light red line) and direct (dark red line) clear sky spectra for one of our sampling sites are represented for comparison. Dotted lines represent spectra for partly cloudy conditions (SNICARv2.1).

*Referee comment*

20. *Fig. 5*: The box-and-whisker plot demonstrates the distribution of measurements nicely when $N$ >2. Does this mean that boxes represent standard deviations even when $N$ = 2? If this is the case, perhaps a bar chart displaying the minimum and maximum values would be a more consistent portrayal of seasonal ranges, since standard deviations are better for estimating the variance of a distribution with a larger number of samples.

*Author's response*

*We thank the referee for drawing the attention on this plot. We agree that standard deviation is more relevant for $N >> 2$, but even so we believe in this case box-and-whiskers plot gives more information than a bar chart. For cases with $N > 2$ (four of the plotted seasonal ranges) the plot allows showing the range where most data fall, together with the extreme values (which in some cases are far away form standard deviation, e.g. Acc. season 2015). For $N=2$ (the three remaining seasonal ranges), the standard deviation is equal to half the separation between the minimum and maximum value, and hence the plot shows the minimum and maximum values. We modified the figure's caption to stress this fact, so the plot can be easily interpreted.*

*Manuscript Changes*

- *Fig. 5 caption:*

**Figure 5.** Seasonal range of PM concentration found on snow/firn samples. For accumulation season, the values represent the mean PM concentration in thick, low PM layers of snow/firn. For ablation season the values represents the surface PM concentration at the end of the season. The box encompasses one standard deviation of data, and whiskers represent minimum and maximum values (when N >2). Notice that for seasonal layers with only two measurements, the box represents those two values (coincident with the definition of standard deviation for N = 2). The plot includes data from both field campaigns, and excludes ablation ice samples, which cannot be assigned to a specific year/season. Fresh snow represent snow fallen a few days before field campaigns of 2016 or 2017.

*Referee comment*

21. *Sect. 3.3 (line 368):* Although Cuffey and Paterson (2010) have written a standard textbook for glaciology, it would be nice to include a more accessible, primary reference that demonstrates this phenomenon.

*Author's response*

*We took the referee suggestion and found a different reference regarding the phenomenon.*

*Manuscript Changes*

Flanner, M. G., and Zender, C. S.: Linking snowpack microphysics and albedo evolution, Journal of Geophysical Research, 111, D12208. https://doi.org/10.1029/2005JD006834, 2006

*Referee comment*

22. *Sect. 3.3 (line 403):* "In the other hand" → "On the other hand"

*Author's response*

*We thank the referee for the useful grammar suggestion.*

*Manuscript Changes*

*We adopt the change suggested by the referee.*

*Referee comment*

23. *Sect. 3.4 (AAR):* Definition of accumulation area ratio? If it is the accumulation area to ablation area ratio, why are the values in m?

*Author's response*

*We thank the referee for noticing the mistake in the units of the Accumulation Area Ratio (the ratio of the glacier's accumulation area to its total area). The numbers are correct but they are a percentage.*

*Manuscript Changes*

*Table 2 header*

| $\alpha_{ice}$ | $\alpha_{firn}$ | $\alpha_{max}$ | Wint. MB (m w.e.) | Annu. MB (m w.e.) | ELA (m) | AAR (%) |
|---|---|---|---|---|---|---|
| | | | | | | |

*Referee comments*

24. *Fig. 9:* The prefix "glacier-wide" is technically redundant, as "surface mass balance" is considered a surface area-integrated quantity. When referring to it as a local quantity, however, it can be stated as "specific surface mass balance." Also, the units on the axes labels should be in parentheses.

25. *Sect. 4 (line 510)* "observation and modeling activities to analysis" → "measurements and modeling to analyze"

26. *Sect. 4 (line 512):* No need for a paragraph break here.

27. *Sect. 4 (line 526):* "may difficult" → "may degrade"

28. *Sect. 4 (line 529):* Remove the comma.

29. *Sect. 4 (line 534):* "glacier-wide" → "surface" (see comment 24)

*Author's response*

*We thank the referee for the useful grammar/phrasing suggestions.*

*Manuscript Changes*

*We adopt the changes suggested by the referee.*

[revised manuscript text omitted]

**12 April 2016**
14:47 (UTC - 3)
Zenith: 51.98°
Effective angle: 51.98°
Clear Sky | Recent snow
Layer: (6 ± 1) cm
Grain size: (1000 ± 200) μm
Snow density: 300 kg m⁻³
PM: (22.0 ± 0.6) mg kg⁻¹
Slope: 0° | 0.626 ± 0.011 | Direct: 0.583
Diffuse: 0.655 | Direct: 0.573
Diffuse: 0.748
W.Aver.: 0.590 | **Grain size:** (+) −0.024 (−) +0.028
PM content: (+) −0.001 (−) +0.001
100 μg kg⁻¹ **BC:** −0.017
**Layer thickness:** (+) +0.010 (−) −0.013 |
| **Acc3-2016** (accum. zone)
**12 April 2016**
16:55 (UTC - 3)
Zenith: 66.04°
Effective angle: 68.7° to 73.6°
Overcast sky (approx. 100 % diffuse rad.) | Dirty summer snow
Layer: (0.3 ± 0.1) cm
Grain size: (1000 ± 200) μm
Snow density: 500 kg m⁻³
PM: (7800 ± 1500) mg kg⁻¹
Slope: 11° | 0.257 ± 0.041 | Direct: 0.435
Diffuse: 0.364 | Direct: 0.445
Diffuse: 0.359
W.Aver.: 0.359 | Grain size: (+) −0.001 (−) +0.002
PM content: (+) −0.001 (−) +0.002
20 mg kg⁻¹ **BC:** −0.049
**Layer thickness:** (+) −0.001 (−) +0.007
Snow density: (+) −0.001 (−) +0.003
% Diff. Rad.: +0.001 (89 % diff. rad.) |
| **Abl3-2017** (accum. pocket on abl. zone)
**3 April 2017**
13:11 (UTC - 3)
Zenith: 47.57°
Effective angle: 56.5° to 59.9°
Overcast sky (89 % to 95 % diffuse rad.) | Dirty snow
Layer: (0.3 ± 0.1) cm
Grain size: (1020 ± 160) μm
Snow density: 500 kg m⁻³
PM: (30000 ± 5000) mg kg⁻¹
Slope: 15° | 0.371 ± 0.011 | Direct: 0.374
Diffuse: 0.360 | Direct: 0.381
Diffuse: 0.354
W.Aver.: 0.356 | Grain size: (+) +0.001 (−) −0.001
PM content: (+) +0.001 (−) −0.001
20 mg kg⁻¹ **BC:** −0.015
Layer thickness: (+) +0.0000001 (−) −0.000002
% Diff. Rad.: (+) +0.002 (−) −0.002
Effective angle: (+) +0.001 (−) −0.001 |
| **Abl4-2017** (accum. pocket on abl. zone)
**3 April 2017**
13:30 (UTC - 3)
Zenith: 46.95°
Effective angle: 57.1° to 60.1°
Overcast sky (approx. 100 % diffuse rad.) | Dirty snow
Layer: (0.10 ± 0.05) cm
Grain size: (740 ± 170) μm
Snow density: 500 kg m⁻³
PM: (12250 ± 2050) mg kg⁻¹
Slope: 15° | 0.266 ± 0.008
0.376 ± 0.015 | Direct: 0.379
Diffuse:0.375 | Direct: 0.384
Diffuse: 0.368
W.Aver.: 0.368 | Grain size: (+) −0.001 (−) +0.002
PM content: (+) −0.004 (−) +0.006
20 mg kg⁻¹ **BC:** −0.050
**Layer thickness:** (+) −0.008 (−) +0.031
Snow density: (+) −0.005 (−) +0.007
% Diff. Rad.: −0.007 (89 % diff. rad.) |
| **Acc5-2017** (accum. zone)
**5 April 2017**
14:26 (UTC - 3)
Zenith: 48.20°
Effective angle: 49.4° to 52.3°
Cloudy sky (34 % to 48 % diffuse rad.) | Recent snow
Layer: (9 ± 1) cm
Grain size: (150 ± 40) μm
Snow density: 300 kg m⁻³
PM: (1.28 ± 0.03) mg kg⁻¹
Slope: 5° | 0.814 ± 0.013 | Direct: 0.788
Diffuse: 0.860 | Direct: 0.778
Diffuse: 0.910
W.Aver.:0.828 | **Grain size:** (+) −0.012 (−) +0.015
PM content: (+) −0.001 (−) +0.001
100 μg kg⁻¹ **BC:** −0.022
Layer thickness: (+) +0.002 (−) −0.002
% Diff. Rad.: (+) +0.005 (−) −0.002
Effective angle: (+) +0.001 (−) −0.001 |
| **Acc6-2017** (accum. zone)
**5 April 2017**
14:48 (UTC - 3)
Zenith: 49.35°
Effective angle: 51.0° to 53.9°
Cloudy sky (34 % to 48 % diffuse rad.) | Recent snow
Layer: (9 ± 1) cm
Grain size: (150 ± 40) μm
Snow density: 300 kg m⁻³
PM: (3.9 ± 0.2) mg kg⁻¹
Slope: 5° | 0.757 ± 0.026 | Direct: 0.786
Diffuse: 0.856 | Direct: 0.776
Diffuse: 0.905
W.Aver.: 0.825 | **Grain size:** (+) −0.013 (−) +0.015
PM content: (+) −0.001 (−) +0.001
100 μg kg⁻¹ **BC:** −0.021
Layer thickness: (+) +0.001 (−) −0.002
% Diff. Rad.: (+) +0.004 (−) −0.002
Effective angle: (+) +0.001 (−) −0.001 |

[revised manuscript text omitted]